# Mastering Syntax, Unlocking Semantics: A Mathematically Provable Two-Stage Learning Process in Transformers

## Abstract

Transformers have emerged as a cornerstone across various fields with extensive applications. However, the training dynamics of transformers remain relatively underexplored. In this work, we present a novel perspective on how transformers acquire knowledge during the training dynamics, inspired by the feature learning theory. To this end, we conceptualize each token as embodying two types of knowledge: ***elementary knowledge*** represented by syntactic information, and ***specialized knowledge*** represented by semantic information. Building on this data structure, we rigorously prove that transformers follow a ***syntax-then-semantics*** learning paradigm, *i.e.*, first mastering syntax in the Elementary Stage and then unlocking semantics in the subsequent Specialized Stage. The results are derived from the training dynamics analysis and finite-time convergence within the in-context learning framework for supervised classification. To our best knowledge, this is the ***first*** rigorous result of a two-stage optimization process in transformers from a feature learning perspective. Empirical findings on real-world language datasets support the theoretical results of the two-stage learning process. Moreover, the spectral properties of attention weights, derived from our theoretical framework, align with the experimental observations, providing further validation.

## 1 Introduction

Transformers (Vaswani, 2017) have emerged as foundational architectures with broad applications across multiple research domains, such as natural language processing (Kenton & Toutanova, 2019; Radford et al., 2019; Brown, 2020), computer vision (Liu et al., 2021; He et al., 2022), *etc*. Recently, large language models (LLM) based on decoder-only transformer architectures further demonstrate impressive capabilities, particularly their remarkable in-context learning (ICL) ability (Brown, 2020), where the model solves new tasks based on prompts without further parameter fine-tuning (Black et al., 2022; Rae et al., 2021). The ICL ability has served as the foundation for developing more advanced prompting techniques to tackle complex problems (Huang & Chang, 2022). Recent theoretical studies derive that transformers can mimic the behavior of supervised learning algorithms when training and test prompts are embedded as sequences of labeled training samples and an unlabeled query (Akyürek et al., 2022; Zhang et al., 2023;

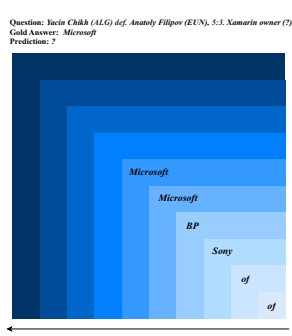

Figure 1: Data information.

Huang et al., 2023; Cheng et al., 2023; Chen et al., 2024). Following this ICL regime, we construct the training prompts and aim to develop the corresponding optimization theory on supervised tasks.

Before theoretically modelling the optimization process, we conducted motivating experiments by setting different rank preservation over attention weights (Details in Section 5). In Figure 1, we have the question 'Yacin Chikh (ALG) def. Anatoly Filipov (EUN), 5:3. Xamarin owner (?)' with the gold answer 'Microsoft'. In this question, the content inside the parentheses, 'ALG' and 'EUN', represents the affiliations of the preceding names. Thus the transformer should induce the affiliation relationship and answer that 'the Xamarin owner is Microsoft'. From right to left in Figure 1, more

small eigenvalues of attention weights are preserved. Using different edited weights, the model's answer improves from syntactically incorrect ('of') to semantically similar ('Sony','BP') to semantically correct ('Microsoft'). We surprisingly induce that smaller eigenvalues preserve syntactic information, while retaining larger eigenvalues enables the model to gradually grasp the semantic meaning of sentences. Motivated by this phenomenon, for transformers on various tasks like classic NLP tasks or supervised learning tasks in ICL regimes, we can disentangle data information (neural representations) into two types: ***elementary knowledge represented by syntactic information, and specialized knowledge represented by semantic information***. Empirically, studies by Bao et al. (2019); Chen et al. (2019a); Huang et al. (2021) consider concrete disentangling processes, such as using paraphrase pairs to experimentally extract two types of information. Theoretically, our research aims to further disentangle the learning process of these two types of data information in a mathematically rigorous manner. This leads to a critical question:

**How do transformers learn the syntactic and semantic information stage wisely?**

To demystify the training dynamics of transformers, the first line of work is beyond ICL regimes (Deora et al., 2023; Li et al., 2023b; Tian et al., 2023a;b). Seminal works by Tian et al. (2023a) and Tian et al. (2023b) analyze how the self-attention mechanism combines input tokens by studying attention maps. However, this line provides few insights into the training dynamics and convergence behaviors. Another closely related line is modelled under the ICL regime like ours (Zhang et al., 2023; Huang et al., 2023; Cheng et al., 2023; Chen et al., 2024). For example, Huang et al. (2023) consider stage wisely learning on the switch of dominant and target features. As a comparison, this paper derives the stage transitions using (equally dominant) syntactic and semantic features. In summary, finite-time training dynamics of transformers remain relatively unexplored, especially when attempting to disentangle the learning process of syntactic and semantic information.

In this paper, we derive a rigorous two-stage learning process where transformers first master syntax and then unlock semantics. Simultaneously, we investigate how transformer weights evolve over time and explore the convergence theory. Our main contributions are summarized as follows.

**(a) Data Modelling with Feature Learning.** Inspired by feature learning theory, we categorize token information into two key feature types: elementary knowledge represented by syntactic information, and specialized knowledge represented by semantic information. Furthermore, we proceed with theoretical abstraction in Section 3, aligning the learning difficulty of foundational knowledge with linearly separable, easy-to-fit data distributions, while associating specialized knowledge with linearly non-separable, hard-to-fit data distributions.

**(b) Mathematical Proof in Two-Stage Learning.** Based on the underlying data structure, to our best knowledge, this is the first paper presenting rigorous proofs for the two-stage learning process in transformers, distinguishing between the initial stage of mastering elementary knowledge and the subsequent stage of acquiring specialized knowledge (Detailed proof in Section F.1 $\sim$ G.2).

**(c) Optimization Trajectory and Convergence Analysis.** We present optimization trajectory and convergence analysis in Section 4, providing deeper insights into the two-stage learning process. Specifically, by adopting feature learning and signal-noise decomposition techniques, we give key propositions and lemmas in Appendix E, carefully discussing the different ReLU activation patterns and the impact of signal or noise weights on network output computations.

**(d) Alignment of Theory and Experiments.** Our theoretical findings are consistent with experimental observations of spectral properties, and experiments on real-world language datasets validate the two-stage learning theory (Experiments are provided in Section 5).

## 2  RELATED WORK

**Optimization Analysis for In-context Learning.** Numerous studies have explored the theoretical properties of transformers for in-context learning (ICL). In theoretical regimes of ICL, a line of work (Zhang et al., 2023; Huang et al., 2023; Cheng et al., 2023; Chen et al., 2024) focuses on optimizing transformers using training prompts structured with input-label pairs, which is similar to this paper. This line of work shows that the global minimum of ICL loss can be reached through gradient flow across different models and tasks (such as models with linear or softmax modules, tasks on linear regression or nonlinear function learning). However, this line usually does not investigate how the model weights are optimized and updated or how the loss evolves throughout training.

Additionally, this line does not address finite-time convergence or the distinct stages of learning various types of information. Among them, Huang et al. (2023) also derive stage-wise learning under linear regression regimes with unbalanced features. Our work differs from them in two aspects: (a) the stage-wise comes from the different types of data in this paper; and (b) this paper focuses on nonlinear classification tasks. This leads to totally different techniques where Huang et al. (2023) emphasize attention maps while we adopt feature learning that accounts for specific data structures to mathematically demonstrate the stages of learning syntactic and semantic information.

**Training Dynamics of Transformers.** Beyond ICL regimes, many studies focus on attention-based models for general learning problems. For example, Deora et al. (2023) investigate optimization and generalization of multi-head attention layer in a binary classification setting, using natural sequences rather than synthetic input-label pairs. Li et al. (2023a) offer a theoretical analysis of training a shallow ViT for classification tasks, characterizing the sample complexity required to achieve zero generalization error. Additionally, Tian et al. (2023a) and Tian et al. (2023b) analyze the SGD training dynamics for one-layer transformers, focusing on how the self-attention layer combines input tokens by studying attention maps.

**Optimization Theory of Neural Networks - Feature Learning.** A line of work studying the convergence of neural networks relies on Neural Tangent Kernel (NTK) technique (Jacot et al., 2018; Li & Liang, 2018; Allen-Zhu et al., 2019; Chen et al., 2019b; Du et al., 2019). It relates the training of over-parameterized (or infinite-width) neural networks to learning over a kernel defined by the network's randomly initialized weights. However, the parameters of practical networks usually do not remain in the lazy training regime and instead move a large distance. Following NTK, a new theoretical branch called feature learning theory in deep learning has emerged (Allen-Zhu & Li, 2020; 2022; Wen & Li, 2021; Li et al., 2023b). Feature learning theory typically assumes specific data generation models, such as Gaussian mixtures. This paper follows this line and utilizes a different syntax-semantics data structure, motivated by empirical observations. This data structure allows us to capture the intrinsic interaction between different features and neural network dynamics.

## 3 PROBLEM SETUP

This section presents the details of the data, model, and training procedure. Concretely, Section 3.1 designs the individual sample structure and constructs training prompts following ICL regimes. Section 3.2 introduces a one-layer attention-based model and two virtual networks. Finally, Section 3.3 describes the corresponding loss function and optimization algorithm used for classification tasks.

**Notations.** Let $\|A\|_F$ be the Frobenius norm for matrix $A$ and $\|x\|_2$ be the 2-norm for vector $x$. For matrix $A$, define $[A]_i$ as the $i$-th row, and $[A]_{ij}$ as the $(i,j)$-th element. For vector $x$, $\text{ReLU}(x) = \max\{x, 0\}$ denotes the standard ReLU activation function, and $\mathbb{1}(x)$ denotes a binary vector which takes entries 1 when $x_i \geq 0$. We use $\odot$ to denote the Hadamard product. For set operators, denote $\cap$, $\cup$, $\oplus$ and $\backslash$ by intersection, union, symmetric difference of two sets, and set difference, respectively. Additionally, throughout the paper, let $U \in \mathbb{R}^{2d \times 2d}$ denote a weight matrix, and $W \in \mathbb{R}^{d \times d}, V \in \mathbb{R}^{d \times d}$ denote the principal submatrices of $U$ which will be defined later. For order analysis, $\text{Poly}(\cdot)$ represents polynomial order, and $f(n) = \mathcal{O}(g(n))$ means that $f(n)$ is asymptotically less than or equal to $g(n)$ in terms of the order of magnitude.

### 3.1 IN-CONTEXT LEARNING FRAMEWORK AND DATA DISTRIBUTION

We adopt the well-established in-context learning (ICL) framework, as introduced by Garg et al. (2022). ICL regime refers to the behavior of models within a specified hypothesis class, where the functions and input samples are drawn respectively from the hypothesis distribution and data distribution. The models operate on sequences, which are known as prompts.

**Training Prompt Structure.** To train a transformer to hold ICL abilities on complex binary classification tasks, the process begins with $N$ random training prompts, which are used to learn a specific classifier in a hypothesis class. As suggested by the general ICL regime, for the $n$-th prompt, input samples $x_1^n, \cdots, x_{L-1}^n$ and query $x_L^n$ are drawn randomly and independently from the same data distribution. The input-label pairs are stacked to form a training prompt $P^n = (x_1^n, y_1^n, \cdots, x_{L-1}^n, y_{L-1}^n, x_L^n)$, with prompt length $L$. For binary classification tasks, the

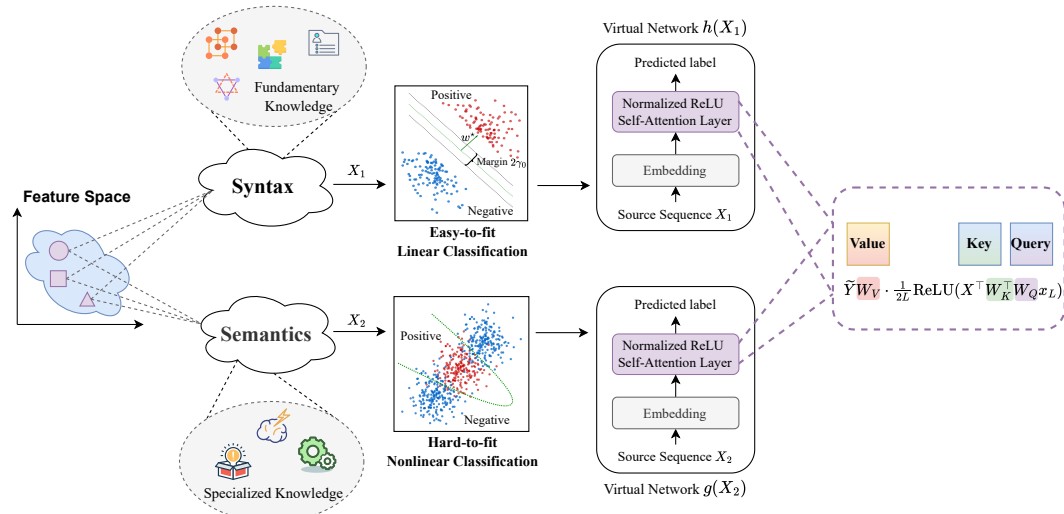

Figure 2: Overview of disentangling syntax and semantics.

ground truth label of $x_i^n$ is denoted by $y_i^n = y(x_i^n) \in \{-1, 1\}$. Especially, for query $x_L^n$, the ground truth label is $y_L^n = y(x_L^n)$. The goal of an in-context learner is to use such prompts to make a prediction $f(x_L^n)$ for the query such that $f(x_L^n) \approx y_L^n$.

**Individual Sample Structure.** For each individual input sample $x_i^n$ in prompt $P^n$, it is composed of two types of components: $\mathcal{P}$ component represents easy-to-fit features, aligning with syntactic information in the corpus, and $\mathcal{Q}$ component represents hard-to-fit features, aligning with semantic information in the corpus.

Specifically, we define $x_i^n = [x_{i,1}^n, x_{i,2}^n]^\top$ where $x_{i,1}^n \in \mathbb{R}^d, x_{i,2}^n \in \mathbb{R}^d$ and $x_i^n \in \mathbb{R}^{2d}$. Then we design the concrete structure of $\mathcal{P}$ and $\mathcal{Q}$ for the sample $x_i^n$ as follows, drawing inspirations from Li et al. (2019). Let $x_{i,1}^n \sim \mathcal{P}_{y_i^n}$ and $x_{i,2}^n \sim \mathcal{Q}_{y_i^n}$. For distribution $\mathcal{P}$, if noise $e$ and optimal classifier $w^\star$ satisfy $\langle w^\star, e \rangle \geq 0$, then we construct the positive sample based on $x_{i,1}^n = \gamma_0 w^\star + e$. Conversely, if $\langle w^\star, e \rangle \leq 0$, we construct the negative sample based on $x_{i,1}^n = -\gamma_0 w^\star + e$. It is natural to find that $x_{i,1}^n$ has easy-to-fit features and it could be easily classified by a linear classifier $\text{sign}(w^\star x_{i,1}^n)$ with a margin of $2\gamma_0$. To simplify, let $w^\star \in \mathbb{R}^d$ be a unit vector *i.e.* $\|w^\star\|_2 = 1$, margin $\gamma_0 = \frac{1}{\sqrt{d}}$, and noise $e \sim \mathcal{N}\left(0, \frac{I_{d \times d}}{d}\right)$. For distribution $\mathcal{Q}$, $x_{i,2}^n = \alpha z$ belongs to the positive class, while $x_{i,2}^n \in \{\alpha(z - \zeta), \alpha(z + \zeta)\}$ belongs to the negative class. Obviously, $z$ is not linearly separable with extremely small bias $\zeta$ and thus $x_{i,2}^n$ contains hard-to-fit features. To simplify, let $\alpha = 1$, $\|z\|_2 = u$, $\|\zeta\|_2 = r \ll u$ and $\langle z, \zeta \rangle = 0$.

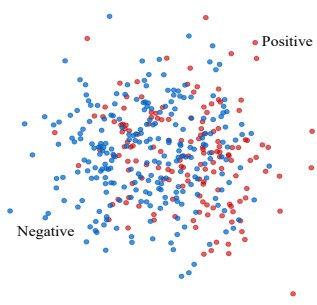

Figure 3: Composite nonlinear classification.

Overall, in Figure 2, we utilize two-dimensional data to intuitively illustrate the roles of two components $\mathcal{P}$ and $\mathcal{Q}$ based on the distribution, in learning both linear and nonlinear classifiers. By concatenating these two components, sample $x_i^n$ is employed to tackle a more complex composite nonlinear classification task, as shown in Figure 3. Despite the data composition, the task's difficulty is significantly increased rather than being a simple combination.

### 3.2 ONE-LAYER TRANSFORMER ARCHITECTURE

**Embeddings.** Given the prompt $P^n = (x_1^n, y_1^n, \cdots, x_{L-1}^n, y_{L-1}^n, x_L^n)$, we construct the embedding matrix by stacking $x_i^n$ or $y_i^n$. Let $X_1^n$ and $X_2^n$ denote the matrices of the two types of features

$x_{i,1}^n$ and $x_{i,2}^n$. Concretely,

$$X_1^n = \begin{bmatrix} x_{1,1}^n & x_{2,1}^n & \cdots & x_{L,1}^n \end{bmatrix} \in \mathbb{R}^{d \times L}, \ X_2^n = \begin{bmatrix} x_{1,2}^n & x_{2,2}^n & \cdots & x_{L,2}^n \end{bmatrix} \in \mathbb{R}^{d \times L}.$$

To ensure the model output is linearly decomposable, we combine $X_1$ and $X_2$ to form the complete feature embedding matrix as $X^n = \begin{bmatrix} X_1^n & 0 \\ 0 & X_2^n \end{bmatrix} \in \mathbb{R}^{2d \times 2L}$. Similarly, we define

$$Y_1^n = Y_2^n \triangleq Y^n = [y_1^n \ \ y_2^n \ \ \cdots \ \ 0] \in \mathbb{R}^{1 \times L},$$

and the complete label matrix as $\widetilde{Y}^n = [Y^n \ \ Y^n] \in \mathbb{R}^{1 \times 2L}$.

**Normalized ReLU Self-Attention Layer.** A self-attention layer (Vaswani, 2017) in the single-head case includes parameters $\theta$: key, query and value matrices $W_K, W_Q \in \mathbb{R}^{2d \times 2d}$, $W_V \in \mathbb{R}^{2L \times 2L}$. Given the feature embedding matrix $X \in \mathbb{R}^{2d \times 2L}$, we use a normalized ReLU activation in place of standard softmax activation as Bai et al. (2024). Then the prediction for query $x_L$ using a one-layer transformer is given by

$$f(U; X, \widetilde{Y}) = \widetilde{Y} W_V \cdot \frac{1}{2L} \mathrm{ReLU} \left( X^\top W_K^\top W_Q x_L \right) = \widetilde{Y}/2L \cdot \mathrm{ReLU} \left( X^\top U x_L \right), \qquad (1)$$

where $\frac{1}{2L}$ is the normalization factor. To simplify, we reparameterize $W_K^\top W_Q \triangleq U \in \mathbb{R}^{2d \times 2d}$ and assume the value matrix is the identity transformation, *i.e.*, $W_V = I$.

We remark that softmax is computationally expensive due to the challenges posed by exponential calculations and the summation over sequence length. Furthermore, transformers with sequence-length normalized ReLU activations have been experimentally studied in Wortsman et al. (2023); Shen et al. (2023), demonstrating comparable performances to standard softmax activation in many vision and NLP tasks.

**Transformer Weight Structure.** Given that individual samples $x_i^n$ can be characterized by two specific types of features, we abstract the real training network into two virtual networks, with the weight matrix composed of two distinct parts. To simplify our analysis, we here consider the simplest structure of weight matrix $U$ as a block diagonal matrix:

$$U = \begin{bmatrix} W & 0 \\ 0 & V \end{bmatrix} \in \mathbb{R}^{2d \times 2d},$$

where weight $W$ operates only on $X_1$ and $V$ operates only on $X_2$. This structure exhibits a strong property of linear decomposability over the model output, i.e. by decomposition, the two new predictions with features $X_1$ and $X_2$ maintain a similar formulation to the original ones:

$$\underbrace{f(U; X, \widetilde{Y})}_{N_U(U; X, \widetilde{Y})} = 1/2 \cdot \underbrace{Y/L \cdot \mathrm{ReLU} \left( X_1^\top W x_{L,1} \right)}_{N_W(W; X_1, Y) \text{ or } h(X_1)} + 1/2 \cdot \underbrace{Y/L \cdot \mathrm{ReLU} \left( X_2^\top V x_{L,2} \right)}_{N_V(V; X_2, Y) \text{ or } g(X_2)}. \qquad (2)$$

In summary, we naturally abstract two virtual networks: network $h(X_1)$ with parameter $W$ operates on $X_1$ part to learn component $\mathcal{P}$, and network $g(X_2)$ with parameter $V$ operates on $X_2$ part to learn component $\mathcal{Q}$. The overview is shown in Figure 2.

### 3.3 Training Procedure

**Loss Function.** To train the transformer model on binary classification tasks, we consider the regularized empirical loss over $N$ training prompts. Denote the logistic loss for each prompt as $l(f(U; X^n, \widetilde{Y}^n)) = \log(1 + e^{-y_L^n f(U; X^n, \widetilde{Y}^n)})$, then

$$\widehat{L}(U) = \frac{1}{N} \sum_{n=1}^N l \left( f(U; X^n, \widetilde{Y}^n) \right), \qquad (3)$$

and the regularized loss function is denoted as $\widehat{L}_\lambda(U) = \widehat{L}(U) + \frac{\lambda}{2} \|U\|_F^2$, where $\lambda$ denotes the $L_2$ regularization coefficient.

**Optimization Algorithm.** Consider stochastic gradient descent with spherical Gaussian noise, which is a simplification of minibatch SGD. Taking initial weight $[U_0]_{ij} \sim \mathcal{N} \left( 0, \tau_0^2 \right)$ and noise $[\xi_t]_{ij} \sim \mathcal{N} \left( 0, \tau_\xi^2 \right)$, then the update of $U$ with time is represented as

$$U_{t+1} = U_t - \gamma_t \nabla_U (\widehat{L}_\lambda(U_t) + \xi_t) = (1 - \gamma_t \lambda) U_t - \gamma_t \xi_t - \gamma_t \nabla_U \widehat{L}(U_t). \qquad (4)$$

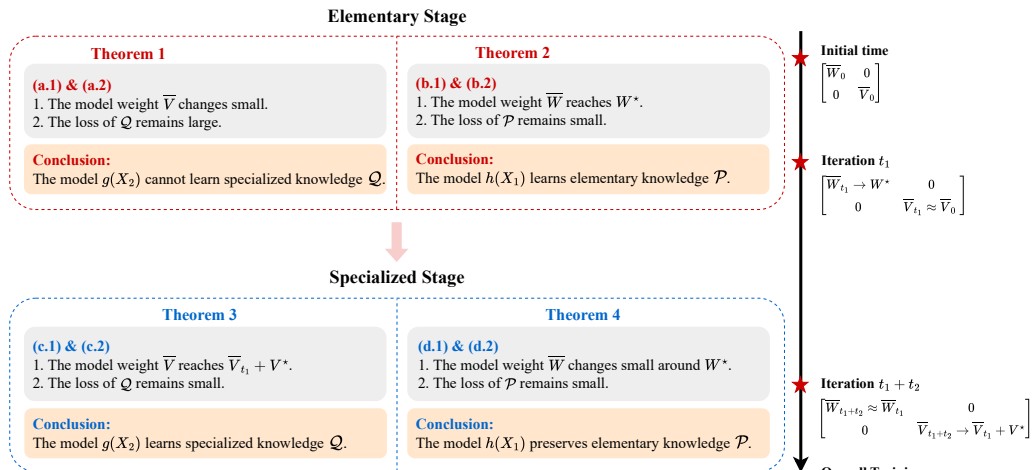

Figure 4: Summary of Two-stage Learning.

**Signal-noise Decomposition.** With noise in SGD optimization, we take signal-noise decomposition for weight $U$, *i.e.*, $U = \overline{U} + \widetilde{U}$ (Allen-Zhu et al., 2019; Li et al., 2019). The signal weight is defined as the weights related to the gradient part, *i.e.*, $\overline{U}_{t+1} \triangleq (1 - \gamma_t \lambda)\overline{U}_t - \gamma_t \nabla_U \widehat{L}(U_t)$. And the noise weight is defined as the weights related to the noise part, *i.e.*, $\widetilde{U}_{t+1} \triangleq (1 - \gamma_t \lambda)\widetilde{U}_t - \gamma_t \xi_t$. Note that due to Equation 4, such decomposition is always valid.

Notably, the noise component $\widetilde{U}$ follows a Gaussian distribution since it is a linear combination of Gaussian random variables. By setting a relatively small variance $\tau_\xi^2$, the signal component always dominates the noise component (Li et al., 2019). Therefore, one can always rewrite the weight $U = \overline{U} + \widetilde{U}$ as a signal part $\overline{U}$ with a small Gaussian random noise $\widetilde{U}$. Based on this observation, we define the training loss $K(\overline{U})$ which depends solely on the signal weight:

$$K(\overline{U}) = \frac{1}{N} \sum_{n=1}^{N} l\left(N_U(\overline{U} + \widetilde{U}; X^n, \widetilde{Y}^n)\right). \tag{5}$$

Based on the above discussions, minimizing Equation 5 is almost equivalent to minimizing Equation 3. Similarly, we take signal-noise decomposition for $W = \overline{W} + \widetilde{W}$ and $V = \overline{V} + \widetilde{V}$, then define the training loss of easy-to-fit component $\mathcal{P}$ over signal weight as $K^1(\overline{W})$, and the training loss of hard-to-fit component $\mathcal{Q}$ over signal weight as $K^2(\overline{V})$:

$$K^1(\overline{W}) = \frac{1}{N} \sum_{n=1}^{N} l\left(N_W(\overline{W} + \widetilde{W}; X_1^n, Y^n)\right), \quad K^2(\overline{V}) = \frac{1}{N} \sum_{n=1}^{N} l\left(N_V(\overline{V} + \widetilde{V}; X_2^n, Y^n)\right). \tag{6}$$

## 4 TWO-STAGE OPTIMIZATION OF TRANSFORMERS

Based on the data characteristics and the different learning complexity of component $\mathcal{P}$ and $\mathcal{Q}$, we split the entire training process into two stages: the Elementary Stage (in Section 4.1, Theorem 1 and Theorem 2), and the Specialized Stage (in Section 4.2, Theorem 3 and Theorem 4). We establish the weight trajectory and analyze the finite-time convergence in the two stages. The main theorems are summarized in Figure 4. Before diving into the details, we introduce the fundamental settings of two stages, including the learning rate and training iterations. Specially,

- **Elementary Stage.** Constant learning rate $\eta_1 = \Theta(1)$; Containing $0 \leq t \leq t_1 \triangleq \frac{1}{\eta_1 \lambda}$ where $\lambda$ denotes the $L_2$ regularization coefficient.
- **Specialized Stage.** Annealing learning rate $\eta_2 = \eta_1 \lambda^2 \epsilon_{V,1}^2 r$ where $\epsilon_{V,1} = \Theta(1/\text{Poly}(d))$ will be introduced later, and $r \triangleq \|\zeta\|_2$ represents the hardness of semantics (See Paragraph 3.1); Containing $t_1 \leq t \leq t_1 + t_2$ where $t_2 \triangleq \frac{\log^2(1/\epsilon_{V,1})}{\eta_2 \lambda \epsilon_{V,1}^2}$.

The annealing learning rate is widely adopted in practical training procedures. Furthermore, we present the same choices of hyperparameters for two stages in Assumption 1.

**Assumption 1.** *Throughout the Theorems, set the variance of initialization parameter* $\tau_0 = \mathcal{O}\left(\frac{1}{\sqrt{\log d}}\right)$, *regularization coefficient* $\frac{1}{\lambda} = \mathcal{O}\left(\sqrt{\log d}\right)$ *and prompt length* $L = \Theta\left(Poly(d)\right)$ *where* $d$ *denotes the input dimension.*

**Discussions on Assumption 1.** We next validate the hyperparameter orders in Assumption 1.

(1) $\tau_0$ denotes the variance of the initialization parameter. The requirement $\tau_0 = \mathcal{O}\left(\frac{1}{\sqrt{\log d}}\right)$ suggests that, as dimension $d$ increases and the data complexity grows, the variance should be adaptively decreased. This aligns with practical training methodologies, as a higher variance might result in a significant shift of the initial weights in high-dimensional spaces, leading to unstable training and potentially impeding convergence.

(2) $\lambda$ denotes the $L_2$ regularization coefficient in the loss function. The requirement $\frac{1}{\lambda} = \mathcal{O}\left(\sqrt{\log d}\right)$ suggests that, as dimension $d$ increases, $\lambda$ should be adjusted to be correspondingly smaller. This is a practical consideration because, in high-dimensional scenarios, a large $\lambda$ may overly constrain the model, potentially causing underfitting. Furthermore, $t_1 \leq \frac{1}{\eta_1 \lambda}$ implies that there might be a longer period during which the model may struggle to effectively learn from the higher-dimensional data $\mathcal{Q}$, which accords with the empirical intuition.

(3) $L$ denotes the prompt length. The requirement $L = \Theta(Poly(d))$ suggests that the model anticipates longer input sequences for learning high-dimensional data, which accords with reality.

## 4.1 ELEMENTARY STAGE

This section aims to analyze the regime with $\eta_1 = \Theta(1)$ and $t \leq t_1 \triangleq \frac{1}{\eta_1 \lambda}$. Our goal is to prove that the weights are optimized from $\overline{U}_0 = \begin{bmatrix} \overline{W}_0 & 0 \\ 0 & \overline{V}_0 \end{bmatrix}$ to $\overline{U}_{t_1} = \begin{bmatrix} \overline{W}_{t_1} \to W^\star & 0 \\ 0 & \overline{V}_{t_1} \approx \overline{V}_0 \end{bmatrix}$. This means that $\overline{W}_{t_1}$ approach the optimal weights $W^\star$, while $\overline{V}_{t_1}$ remains close to $\overline{V}_0$. We split the derivation into two theorems: Theorem 1 demonstrates that the hard-to-fit component $\mathcal{Q}$ (specialized knowledge) is not effectively learned by network $g$, and Theorem 2 demonstrates that the network $h$ successfully learns the easy-to-fit component $\mathcal{P}$ (elementary knowledge). We start from Theorem 1.

**Theorem 1.** *In the elementary stage with* $\eta_1 = \Theta(1)$ *and* $t_1 = \frac{1}{\eta_1 \lambda}$ *where* $\lambda$ *denotes regularization coefficients. With Assumption 1, initial weights* $V_0 \to \mathbf{0}_{d \times d}$ *and* $N$ *training prompts, it holds that*

*(a.1) For the model parameter* $V$ *of network* $g$*, through gradient descent,* $\|\overline{V}_{t_1}\|_F$ *satisfies*

$$\|\overline{V}_{t_1}\|_F = \Theta\left(\frac{1}{Poly(d)}\right).$$

*(a.2) With random and small noise weight, the training loss of hard-to-fit component* $\mathcal{Q}$ *over signal weight (Definition in Equation 6) at iteration* $t_1$ *satisfies*

$$K_{t_1}^2\left(\overline{V}_{t_1}\right) \gtrsim \log 2 - \frac{1}{\sqrt{\log d}} - \sqrt{\frac{\log d}{N}}.$$

*Namely, the hard-to-fit component* $\mathcal{Q}$ *is not efficiently learned by* $g$ *within* $t_1$ *iterations.*

**Messages Behind Theorem 1.** Theorem 1 demonstrates that the hard-to-fit component $\mathcal{Q}$ cannot be effectively learned by the corresponding network $g$ defined in Equation 2. **In (a.1)**, within $t_1$ iterations, the weight $\|\overline{V}_{t_1}\|_F$ is approximately in order $\frac{1}{Poly(d)}$, which implies that the model weight $V$ is almost not optimized since $\|\overline{V}_{t_1}\|_F \approx \|\overline{V}_0\|_F$. **In (a.2)**, we provide the lower bound for the training loss of component $\mathcal{Q}$. The value is close to $\log 2$ with large dimension $d$ and training prompts $N$. Overall, the above discussions exhibit that specialized knowledge like $\mathcal{Q}$ is not effectively learned by the network $g$. We defer the proof to Appendix F.1 and the proof sketch in Remark 6.

**Theorem 2.** *In the elementary stage with* $\eta_1 = \Theta(1)$ *and* $t_1 = \frac{1}{\eta_1 \lambda}$ *where* $\lambda$ *denotes regularization coefficients. With Assumption 1 and initial weights* $W_0 \to \mathbf{0}_{d \times d}$*, it holds that there exist* $\epsilon_{W,1} = \Theta\left(1/Poly(d)\right)$*,* $\epsilon_W = \Theta\left((Poly(d))^{2/3}\right)$ *(See Definition in Equation 16 and 10) such that*

**(b.1)** *The model parameter $W$ of network $h$ is optimized by gradient descent within $t_1$ iterations,*
$$\|\overline{W}_{t_1}\|_F = \Theta\left(d\log(1/\epsilon_{W,1})\right) \gg \|\overline{W}_0\|_F.$$

**(b.2)** *With random and small noise weight, the training loss of easy-to-fit component $\mathcal{P}$ over signal weight (Definition in Equation 6) at iteration $t_1$ satisfies*
$$K_{t_1}^1(\overline{W}_{t_1}) \lesssim \epsilon_{W,1} + \frac{\sqrt{d}\log d}{L}\epsilon_W + \frac{1}{\sqrt{\log d}}.$$
*Namely, the network $h$ learns the easy-to-fit component $\mathcal{P}$ within $t_1$ iterations.*

**Messages Behind Theorem 2.** Theorem 2 describes how the easy-to-fit component $\mathcal{P}$ is learned by the corresponding network $h$ defined in Equation 2. **In (b.1)**, within $t_1$ iterations, $\|\overline{W}\|_F$ significantly grows from the order $\|\overline{W}_0\|_F \approx \sqrt{d}$ to the order $\|\overline{W}_{t_1}\|_F \approx d\log(1/\epsilon_{W,1})$, indicating that the knowledge might be learned. In comparison, $\overline{V}_{t_1}$ for the hard-to-fit component $\mathcal{Q}$ changes small since $\|\overline{V}_{t_1}\|_F \approx \|\overline{V}_0\|_F \approx \frac{1}{\text{Poly}(d)}$ (See Theorem 1 (a.1)). **In (b.2)**, it shows that the loss of easy-to-fit component $\mathcal{P}$ is upper bounded by an $o(1)$ term which converges to zero as the dimension $d$ goes to infinity. Concretely, the upper bound $\epsilon_{W,1} + \frac{\sqrt{d}\log d}{L}\epsilon_W + \frac{1}{\sqrt{\log d}}$, has the order of $\frac{1}{\text{Poly}(d)} + \frac{1}{(\text{Poly}(d))^{1/3}} + \frac{1}{\sqrt{\log d}}$. In comparison, the loss of hard-to-fit component $\mathcal{Q}$ is lower bounded by a constant close to $\log 2$ (See Theorem 1 (a.2)). In summary, the above discussions imply that **the network $h$ learns elementary knowledge like $\mathcal{P}$, marking the so-called elementary stage.** We defer the Proof to Appendix F.2 and the Proof Sketch in Remark 7.

## 4.2 Specialized Stage

This section aims to analyze the regime with $\eta_2 = \eta_1\lambda^2\epsilon_{V,1}^2 r$ and $t_1 \leq t \leq t_1 + t_2$, where $\epsilon_{V,1} = \Theta(1/\text{Poly}(d))$ is defined in Equation 17, $t_1 \triangleq \frac{1}{\eta_1\lambda}$ and $t_2 \triangleq \frac{\log^2(1/\epsilon_{V,1})}{\eta_2\lambda\epsilon_{V,1}^2}$.

Our goal is to prove that the weights are optimized from $\overline{U}_{t_1} = \begin{bmatrix} \overline{W}_{t_1} & 0 \\ 0 & \overline{V}_{t_1} \end{bmatrix}$ to $\overline{U}_{t_1+t_2} = \begin{bmatrix} \overline{W}_{t_1+t_2} \approx \overline{W}_{t_1} & 0 \\ 0 & \overline{V}_{t_1+t_2} \to \overline{V}_{t_1} + V^\star \end{bmatrix}$. In total, we split the derivation into two theorems: Theorem 3 demonstrates that the network $g$ learns specialized knowledge like hard-to-fit component $\mathcal{Q}$, and Theorem 4 demonstrates that the network $h$ continues to preserve the elementary knowledge like easy-to-fit component $\mathcal{P}$. We start from Theorem 3.

**Theorem 3.** *In the specialized stage with annealing learning rate $\eta_2 = \eta_1\lambda^2\epsilon_{V,1}^2 r$ and $t_1 \leq t \leq t_1 + t_2$, where $\epsilon_{V,1} = \Theta(1/Poly(d))$, $t_1 \triangleq \frac{1}{\eta_1\lambda}$, $t_2 \triangleq \frac{\log^2(1/\epsilon_{V,1})}{\eta_2\lambda\epsilon_{V,1}^2}$, $\lambda$ denotes the $L_2$ regularization coefficient and data noise $\|\zeta\|_2 = r$ (See Paragraph 3.1). With Assumption 1, it holds that*

**(c.1)** *The model parameter $V$ of network $g$ is optimized by gradient descent within $t_2$ iterations,*
$$\|\overline{V}_{t_1+t_2}\|_F = \Theta\left(\frac{\log(1/\epsilon_{V,1})}{\epsilon_{V,1}} + \frac{1}{Poly(d)}\right) \gg \|\overline{V}_{t_1}\|_F.$$

**(c.2)** *With random and small noise weight, the training loss of hard-to-fit component $\mathcal{Q}$ over signal weight (Definition in Equation 6) satisfies*
$$K_{t_1+t_2}^2(\overline{V}_{t_1+t_2}) \lesssim \epsilon_{V,1} + \frac{1}{(\log d)^{1/4}} + \frac{1}{\sqrt{\log d}}.$$
*Namely, the network $g$ learns hard-to-fit component $\mathcal{Q}$ within $t_2$ iterations.*

**Messages Behind Theorem 3.** Theorem 3 illustrates the optimization in the specialized stage. **Statement (c.1)** implies that within $t_2$ iterations, $\|\overline{V}\|_F$ grows from the order $\|\overline{V}_{t_1}\|_F \approx \frac{1}{\text{Poly}(d)}$ to the order $\|\overline{V}_{t_1+t_2}\|_F \approx \frac{\log(1/\epsilon_{V,1})}{\epsilon_{V,1}} + \frac{1}{\text{Poly}(d)} \approx \text{Poly}(d)\log\text{Poly}(d) + \frac{1}{\text{Poly}(d)}$ (derivation based on Assumption 1). **Statement (c.2)** implies that the loss is upper bounded by $o(1)$ which converges to zero as $d$ goes to infinity. Notably, the upper bound given by the order $\epsilon_{V,1} + \frac{1}{(\log d)^{1/4}} + \frac{1}{\sqrt{\log d}} \approx \frac{1}{\text{Poly}(d)} + \frac{1}{(\log d)^{1/4}} + \frac{1}{\sqrt{\log d}}$. Compared to Theorem 1 with constant lower bound, we conclude that **with a small learning rate, the network $g$ learns specialized knowledge, marking the so-called specialized stage.** We defer the Proof to Appendix G.1 and the Proof Sketch in Remark 8.

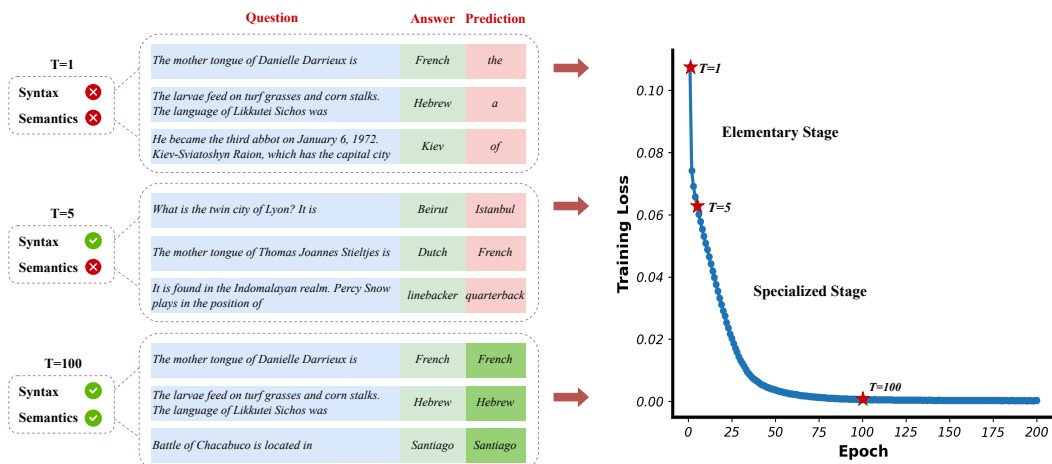

Figure 5: Two-stage learning of syntax and semantics.

**Discussion on Parameter Orders.** We first focus on the learning rate $\eta_2 = \eta_1 \lambda^2 \epsilon_{V,1}^2 r$. Given the choices in Assumption 1, $\eta_2 \approx \mathcal{O}\left(\frac{\log d}{(\text{Poly}(d))^2} \eta_1\right)$. It usually follows that $\eta_2 < \eta_1$, which accords with practical training. Additionally, the current learning process keeps $t_2 = \mathcal{O}\left(\text{Poly}(d)(\log d)^{7/2}/\eta_1\right)$, which is significantly longer than $t_1 = \mathcal{O}\left(\sqrt{\log d}/\eta_1\right)$, coming from the difficulty of learning simple and complex components.

**Theorem 4.** *In the specialized stage with annealing learning rate $\eta_2 = \eta_1 \lambda^2 \epsilon_{V,1}^2 r$ and $t_1 \leq t \leq t_1 + t_2$, where $\epsilon_{V,1} = \Theta(1/Poly(d))$, $t_1 \triangleq \frac{1}{\eta_1 \lambda}$, $t_2 \triangleq \frac{\log^2(1/\epsilon_{V,1})}{\eta_2 \lambda \epsilon_{V,1}^2}$, $\lambda$ denotes the $L_2$ regularization coefficient and data noise $\|\zeta\|_2 = r$ (See Paragraph 3.1). With Assumption 1 and number of training prompts $N = \Theta(Poly(d))$, it holds that*

*(d.1) For the model parameter $W$ of network $h$, through gradient descent optimization from iteration $t_1$ to $t_1 + t_2$, $\|\overline{W}_{t_1+t_2} - \overline{W}_{t_1}\|_F$ satisfies*

$$\left\|\overline{W}_{t_1+t_2} - \overline{W}_{t_1}\right\|_F \lesssim \frac{\epsilon_{V,1}^2}{\log^2\left(1/\epsilon_{V,1}\right)\sqrt{\log d}}.$$

*(d.2) With random and small noise weight, the training loss of easy-to-fit component $\mathcal{P}$ over signal weight (Definition in Equation 2) satisfies*

$$\left|K_{t_1+t_2}^1(\overline{W}_{t_1+t_2}) - K_{t_1}^1(\overline{W}_{t_1}))\right| \lesssim \frac{\epsilon_{V,1}^2}{\log^2\left(1/\epsilon_{V,1}\right)\sqrt{\log d}}.$$

*Namely, the network $h$ continues to preserve the easy-to-fit knowledge like $\mathcal{P}$ within $t_2$ iterations.*

**Messages Behind Theorem 4.** Theorem 4 demonstrates the optimization process on the easy-to-fit part $\mathcal{P}$ in specialized stage, annealing the learning rate from $\eta_1$ to $\eta_2$. **Statement (d.1)** demonstrates that the signal weight $\overline{W}$ does not change significantly in the specialized stage, given the upper bound $o(1)$. Concretely, the upper bound of the weight difference between two moments is $\frac{\epsilon_{V,1}^2}{\log^2(1/\epsilon_{V,1})\sqrt{\log d}}$, with the order of $\frac{1}{(\text{Poly}(d))^2(\log d)^{5/2}}$. **Statement (d.2)** demonstrates that the loss also does not change much from iteration $t_1$ to $t_1 + t_2$, ensuring that the model remains low training loss on easy-to-fit component $\mathcal{P}$. In detail, the small changes in loss have an order of $\frac{1}{(\text{Poly}(d))^2(\log d)^{5/2}}$. In summary, **in the specialized stage, network $h$ continues to preserve the knowledge $\mathcal{P}$ acquired during the elementary knowledge.** Given that both the changes in signal weight $\overline{W}$ and the loss are minimal, we also conclude that the specialized stage is dedicated exclusively to the learning of hard-to-fit component $\mathcal{Q}$. We defer the Proof to Appendix G.2 and the Proof Sketch to Remark 9.

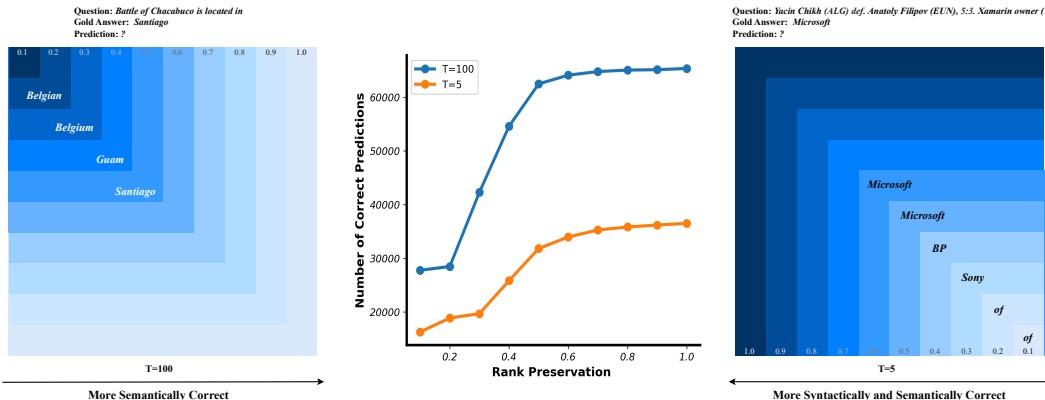

Figure 6: Spectral Characteristics.

## 5 EXPERIMENTS

We conduct motivating experiments on Counterfact dataset containing 65,757 question-answer examples dataset (Meng et al., 2022). All experiments utilize the GPT-2 architecture. Our goal is to verify the two-stage learning of syntax and semantics with this dataset, as well as the spectral characteristics of model attention weights. Additional dataset descriptions and experiments on HotpotQA (Yang et al., 2020) are detailed in Appendix C.

**Verify Two-stage Learning of Syntax and Semantics.** In Figure 5, we present the training loss over 200 epochs, highlighting three key moments with representative samples, including questions, gold answers, and the model's predictions. At the initial time $T = 1$, many predictions are both syntactically and semantically incorrect. By $T = 5$, we observe a significant decrease in training loss; all predictions meet syntactic requirements, but most remain semantically incorrect and inconsistent with the true answers. Thus, the period from $T = 1$ to $T = 5$ corresponds to our theoretical Elementary Stage. By $T = 100$, all predictions are syntactically correct, with most being semantically correct and achieving small training loss. Therefore, the period from $T = 6$ to $T = 100$ represents our theoretical Specialized Stage. Overall, this experiment supports our theory of two-stage learning for syntax and semantics.

**Verify Spectral Characteristics.** There is a direct Corollary 1 (in Appendix B) from Theorems 2 $\sim$ 4, demonstrating that ***relatively small eigenvalues of attention weights store syntax information and large ones store semantics***. We verify this insight empirically in Figure 6 by preserving different eigenvalues and observing the model performances. Concretely, at time $T = 5$ (fully syntactically correct) and $T = 100$ (fully syntactically correct, nearly fully semantically correct), we set the rank preservation $\rho$ ranging from 0.1 to 1.0, to Obtain edited matrices with different eigenvalues using SVD for comparing predictions. For the left figure, we find that the model's predictions become more semantically similar and accurate, as rank preservation $\rho$ increases (maintaining more *large* eigenvalues). For the right figure, we find that the model gradually grasps correct syntax and semantic information as $\rho$ increases (maintaining more *small* eigenvalues). In addition, in the middle figure, the number of correct predictions increases with larger rank preservation, which accords with intuition. We defer the detailed discussion to Appendix C.1.

## 6 CONCLUSION

This paper provides rigorous proof for the two-stage learning process of transformers in ICL tasks. We categorize token information into two feature types: *elementary knowledge* represented by syntactic information, and *specialized knowledge* represented by semantic information. By employing feature learning and signal-noise decomposition techniques, we analyze the optimization trajectory, finite-time convergence, and spectral characteristics under the ICL regime, offering deeper insights into the optimization process. Ultimately, our work aims to provide a new perspective and a theoretical framework for understanding the optimization dynamics of transformers.

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

# Appendix

# A  TABLE OF NOTATIONS

Table 1: Table of Notations.

| Notation | Description |
|---|---|
| $t_1$ | Total iterations of the elementary stage |
| $t_2$ | Total iterations of the specialized stage |
| $N$ | Number of training prompts |
| $L$ | Training prompt length (the last token is a query) |
| $x_i^n = [x_{i,1}^n, x_{i,2}^n]^\top \in \mathbb{R}^{2d}$ | Divide the $i$-th token of $n$-th training prompts into two parts |
| $x_{i,1}^n \sim \mathcal{P} \in \mathbb{R}^d$ | The syntactic information in a token |
| $x_{i,2}^n \sim \mathcal{Q} \in \mathbb{R}^d$ | The semantic information in a token |
| $X_1^n = \begin{bmatrix} x_{1,1}^n & x_{2,1}^n & \cdots & x_{L,1}^n \end{bmatrix} \in \mathbb{R}^{d \times L}$ | Stack of $x_{i,1}^n$ |
| $X_2^n = \begin{bmatrix} x_{1,2}^n & x_{2,2}^n & \cdots & x_{L,2}^n \end{bmatrix} \in \mathbb{R}^{d \times L}$ | Stack of $x_{i,2}^n$ |
| $X^n = \begin{bmatrix} X_1^n & 0 \\ 0 & X_2^n \end{bmatrix} \in \mathbb{R}^{2d \times 2L}$ | Stack of $X_1^n$ and $X_2^n$ |
| $y_i^n \in \{-1, 1\}$ | Binary classification label |
| $Y^n = \begin{bmatrix} y_1^n & y_2^n & \cdots & 0 \end{bmatrix} \in \mathbb{R}^{1 \times L}$ | Stack of $y_i^n$ |
| $\widetilde{Y}^n = \begin{bmatrix} Y^n & Y^n \end{bmatrix} \in \mathbb{R}^{1 \times 2L}$ | Stack of $Y_1^n$ and $Y_2^n$ |
| $f(U; X, \widetilde{Y})$ | Normalized ReLU self-attention output, see in Equation 1 |
| $h(X_1)$ | Virtual network operates on $X_1$, see in Equation 2 |
| $g(X_2)$ | Virtual network operates on $X_2$, see in Equation 2 |
| $U = \begin{bmatrix} W & 0 \\ 0 & V \end{bmatrix} \in \mathbb{R}^{2d \times 2d}$ | Model parameter of normalized ReLU self-attention network |
| $U = \overline{U} + \widetilde{U} \in \mathbb{R}^{2d \times 2d}$ | Signal-noise decomposition of weight $U$ |
| $W = \overline{W} + \widetilde{W} \in \mathbb{R}^{d \times d}$ | Model parameter of virtual network $h$, signal-noise decomposition of weight $W$ |
| $V = \overline{V} + \widetilde{V} \in \mathbb{R}^{d \times d}$ | Model parameter of virtual network $g$, signal-noise decomposition of weight $V$ |
| $\widehat{L}(U)$ | The empirical loss over weight $U$, see in Equation 3 |
| $K(\overline{U})$ | The training loss over signal weight $\overline{U}$, see in Equation 5 |
| $K^1(\overline{W})$ | The training loss over signal weight $\overline{W}$, see in Equation 6 |
| $K^2(\overline{V})$ | The training loss over signal weight $\overline{V}$, see in Equation 6 |

# B  ADDITIONAL COROLLARY FOR SPECTRAL CHARACTERISTICS FROM THEOREMS

**Corollary 1.** *With choices of* $\tau_0 = \mathcal{O}\left(\frac{1}{\sqrt{\log d}}\right), \frac{1}{\lambda} = \mathcal{O}\left(\sqrt{\log d}\right)$ *and* $L = \Theta\left(Poly(d)\right)$. *Denote* $p_1, p_2$ *as the proportions of the negative derivative of logistic loss (i.e.,* $l'(f(W; X_1, Y)), l'(f(V; X_2, Y)) < 0$*). Let* $k_1 \triangleq \max(\|x\|_2^2)\widehat{\mathbb{E}}\left[|l'_-|\boldsymbol{I}^\top \mathbb{1}(X_1^\top x_{L,1})\right], k_2 \triangleq \max(\|x\|_2^2)\widehat{\mathbb{E}}\left[|l'_-|\boldsymbol{I}^\top \mathbb{1}(X_2^\top x_{L,2})\right].$

*(a) In the elementary stage within $t_1 \leq \frac{1}{\eta_1 \lambda}$ iterations, the spectral dynamics satisfy*

$$Tr(W_{t_1}) = (1 - \eta_1\lambda + 2p_1k_1\eta_1/L)^{t_1} Tr(W_0), \quad Tr(V_{t_1}) = (1 - \eta_1\lambda + 2p_1k_2\eta_1/L)^{t_1} Tr(V_0).$$

*Further at iteration $t_1$, we have*

$$Tr(W_{t_1}) > Tr(V_{t_1}).$$

*(b) In the specialized stage within $t_2 \leq \frac{\log^2(1/\epsilon_{V,1})}{\eta_2\lambda\epsilon_{V,1}^2}$ iterations, the spectral dynamics satisfy*

$$Tr(W_{t_1+t_2}) = (1 - \eta_2\lambda + 2p_1k_1\eta_2/L)^{t_2} Tr(W_{t_1}), \quad Tr(V_{t_1+t_2}) = (1 - \eta_2\lambda + 2p_2k_2\eta_2/L)^{t_2} Tr(V_{t_1}).$$

*Further at iteration $t_1 + t_2$, we have*

$$Tr(W_{t_1+t_2}) < Tr(V_{t_1+t_2}).$$

**Remark 5.** *By applying spectral analysis techniques, such as SVD and gradient descent on eigenvalues, we conclude that whether in the elementary stage or specialize stage, $Tr(W_t)$ and $Tr(V_t)$ follow similar update rules. The rate of exponential growth over time primarily depends on three factors: (1) the learning rate $\eta_1$ or $\eta_2$; (2) the proportion $p_1$ or $p_2$ of the negative derivative of logistic loss; and (3) $k_1$ or $k_2$ represents the mean absolute value of the selected negative derivative. By the way, the negative derivative of the logistic loss is selected based on the similarity between query $x_L$ and sequence $X$, i.e. $\mathbb{1}(X_1^\top x_{L,1})$. When comparing the updating rules for the traces of weights in the two stages, we find that the three factors differ and vary with training. However, the overall exponential growth trend remains consistent. Additionally, from Theorems 2 $\sim$ 4, it's straightforward to compare the relationship of $Tr(W)$ and $Tr(V)$ at iteration $t_1$ and $t_1 + t_2$, which will be further verified through experiments on real-world language datasets.*

## C ADDITIONAL EXPERIMENTAL RESULTS

### C.1 EXPERIMENTS ON COUNTERFACT DATASET.

Counterfact (Meng et al., 2022) is a question-answering dataset consisting of knowledge tuples in the form of (subject, relation, answer). These tuples are constructed using entities in Wikidata. Also, there are three paraphrased prompts for each question, resulting in a total of 65,757 examples for the entire dataset. In the following, we provide more discussions about the experimental results in Figure 5 and 6.

**Verify Two-stage Learning of Syntax and Semantics.** In Figure 5, we present the training loss over 200 epochs, highlighting three key moments with representative samples, including questions, gold answers and the model's predictions. At the initial time $T = 1$, many predictions are both syntactically and semantically incorrect. By $T = 5$, we observe a significant decrease in training loss; all predictions meet syntactic requirements, but most are remain semantically incorrect and inconsistent with the true answers. Thus, the period from $T = 1$ to $T = 5$ corresponds to our theoretical Elementary Stage. By $T = 100$, all predictions are syntactically correct, with most being semantically correct and achieving a very low loss value. Therefore, the period from $T = 6$ to $T = 100$ represents our theoretical Specialized Stage. Overall, this experiment supports our theory of two-stage learning for syntax and semantics.

**Verify Spectral Characteristics.** From Theorems 2 $\sim$ 4, based on the relationship of F-norm and trace, it's straightforward to get $Tr(W_{t_1+t_2}) < Tr(V_{t_1+t_2})$ at convergence time $t_1 + t_2$ (Detailed Corollay 1 is shown in Appendix B). We know that weight $W$ of network $h$ operates on the elementary syntax and weight $V$ of network $g$ operates on the specialized semantics. Then the corollary of $Tr(W_{t_1+t_2}) < Tr(V_{t_1+t_2})$ hints that, ***relatively small eigenvalues of attention weights store syntax information and large ones store semantics***.

Thus in Figure 6, we perform model editing on the attention layer weights of the model to analyze the impact of large or small eigenvalues. Concretely, we edit attention weights at time $T = 5$ (fully syntactically correct) and $T = 100$ (fully syntactically correct, nearly fully semantically correct). Using SVD, we sort the eigenvalues of attention weights and set rank preservation coefficient $\rho$, ranging from 0.1 to 1.0. As shown in Figure 6, the numbers in matrices represent the rank preservation coefficient $\rho$ of the current matrix.

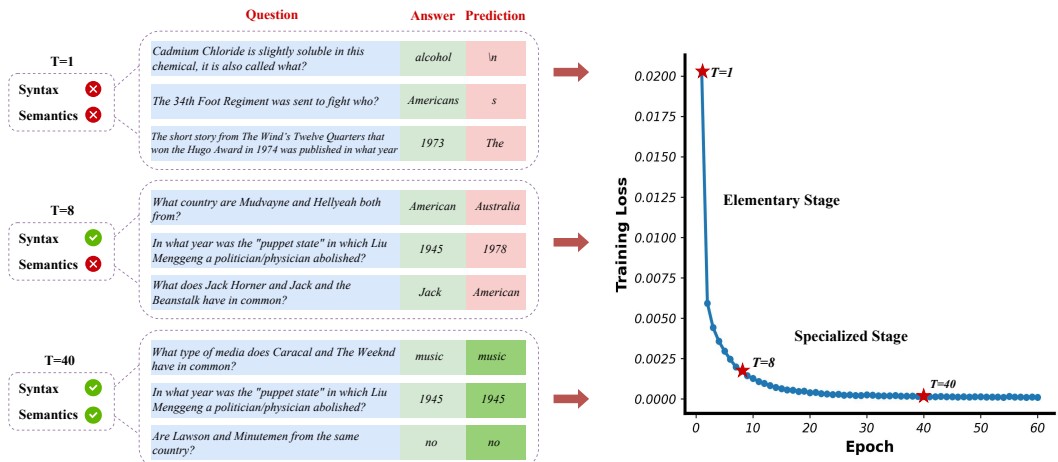

Figure 7: Hotpot dataset: two-stage learning of syntax and semantics.

- **For the left figure**, we first edit attention weights at $T = 100$. Eigenvalues are sorted from largest to smallest and matrices preserve the top $\rho$ proportion of the largest eigenvalues. When $\rho = 0.1$, it means maintaining $10\%$ of the largest eigenvalues and corresponding eigenvectors. The figure displays 10 weight matrices, with $\rho$ ranging from $0.1$ to $1.0$ from left to right. As $\rho$ increases, more large eigenvalues are preserved, and the model's predictions become more semantically similar and accurate.

- **For the right figure**, we further edit attention weights at $T = 5$. Eigenvalues are sorted from smallest to largest and matrices preserve the top $\rho$ proportion of the smallest eigenvalues. From right to left, more small eigenvalues are included. As more eigenvalues of the full matrix are used, the model gradually grasps correct syntax and semantic information.

- **For the middle figure**, it shows that the number of correct predictions increases with larger rank preservation, which is intuitive. In summary, the spectral characteristics insights drawn from our theory are also empirically reasonable.

## C.2 EXPERIMENTS ON HOTPOTQA DATASET.

**HotpotQA.** We choose the HotPotQA dataset available on HuggingFace, with a small size 13,530 (Meng et al., 2022). Taking experiments under the same setting as Section 5, **in Figure 7**, we first verify the two-stage learning of syntax and semantics under this question-answering dataset. The period from $T = 1$ to $T = 8$ corresponds to our theoretical Elementary Stage and the period from $T = 9$ to $T = 40$ represents our theoretical Specialized Stage. **In Figure 8**, we verify the spectral characteristics that relatively small eigenvalues of attention weights store syntax information and large ones store semantics. Specifically, similar to Section 5, we perform model editing on the attention weights at time $T = 8$ (fully syntactically correct) and $T = 40$ (fully syntactically correct, nearly fully semantically correct) and set rank preservation $\rho$ from $0.1$ to $1.0$.

## D USEFUL PROBABILITY CONCENTRATION INEQUALITIES

**Lemma 1** (Hoeffding's Inequality for General Bounded Random Variables, cite HDP p16). *Let* $X_1, \cdots, X_N$ *be independent random variables. Assume that* $X_i \in [m_i, M_i]$ *for every* $i$. *Then, for any* $t > 0$, *we have*

$$\Pr\left(\sum_{i=1}^{N}(X_i - \mathbb{E}[X_i]) \geq t\right) \leq \exp\left(-\frac{2t^2}{\sum_{i=1}^{N}(M_i - m_i)^2}\right)$$

**Lemma 2** (Bernstein's Inequality for Bounded Random Variables, cite ¡concentration.pdf¿, lemma 7.37). *Let* $X_1, \cdots, X_N$ *be i.i.d. and suppose that* $|X_i| \leq c, \mathbb{E}(X_i) = \mu, \sigma^2 = \frac{1}{N}\sum_{i=1}^{N} Var(X_i)$.

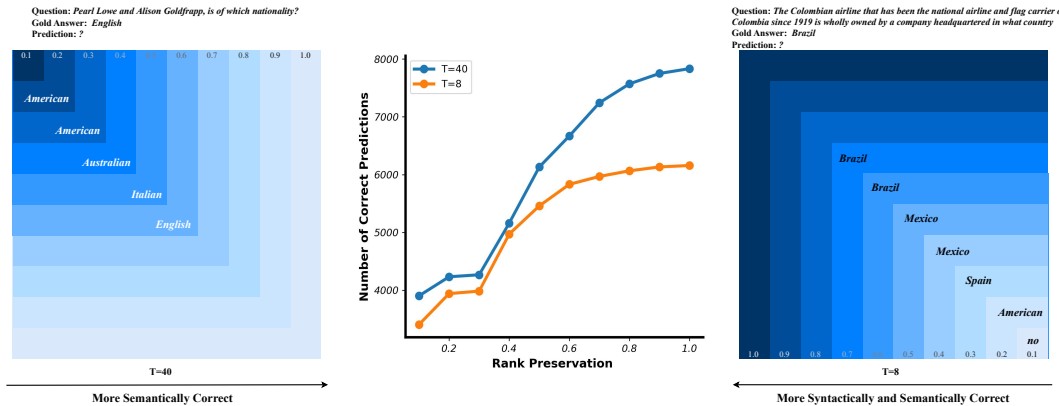

Figure 8: Hotpot dataset: verify spectral characteristics.

*With probability at least $1 - \delta$,*

$$\left| \sum_{i=1}^{N} X_i - \mu \right| \leq \sqrt{\frac{2\sigma^2 \log(1/\delta)}{n}} + \frac{2c \log(1/\delta)}{3n}$$

**Lemma 3** (Norm of Matrix with Gaussian Entries, cite HDP p85). *Let $A$ be an $n \times n$ random matrix whose entries $A_{ij}$ are independent gaussian random variables with $N(0, \sigma^2)$. Then for any $t > 0$, we have*

$$\|A\| \lesssim \sigma \sqrt{n}$$

**Lemma 4** (Standard Gaussian Concentration Inequality). *Suppose that $X = X_1, \cdots, X_N$ are i.i.d. standard complex Gaussian variables, and suppose $F : \mathbb{C}^n \to \mathbb{R}$ is a 1-Lipschitz function with respect to the Euclidean metric. Then $\mathbb{E}[X] < \infty$ and for all $t \geq 0$,*

$$\Pr\left( X - \mathbb{E}[X] > t \right) \leq e^{-t^2}$$

**Lemma 5** (Chernoff Bound for Guassian Variables). *Let $X \sim \mathcal{N}(\mu, \sigma^2)$, then $\mathbb{E}[e^{\lambda X} = \exp\left( \mu\lambda + \sigma^2\lambda^2/2 \right)$ and for all $t \geq 0$,*

$$\Pr\left( |X - \mu| > t \right) \leq 2 \exp\left( -\frac{t^2}{2\sigma^2} \right)$$

$$\Pr\left( \left| \frac{X - \mu}{\sigma} \right| > t \right) \leq 2 \exp\left( -\frac{t^2}{2} \right)$$

# E   PROPOSITIONS, LEMMAS AND COROLLARIES

**Assumption 2.** *For $X_1, X_2 \in \mathbb{R}^{d \times L}$ that satisfies the data structure, let $i$ be $i$-th row, we have*

$$\|[X_1^\top]_i\|_2 \leq u + \gamma_0, \|X_1^\top\|_F \leq \sqrt{L}(u + \gamma_0)$$
$$\|[X_2^\top]_i\|_2 \leq u + r, \|X_2^\top\|_F \leq \sqrt{L}(u + r)$$
$$\|[X^\top]_i\|_2 \leq \max\{u + \gamma_0, u + r\}, \|X^\top\|_F \leq \sqrt{L(u + \gamma_0)^2 + L(u + r)^2}$$

*Proof.* For $X_1$, we have

$$\|w^\star\|_2 = 1, \|[X^\top]_i\|_2 \leq u + \gamma_0, \|X^\top\|_F \leq \sqrt{L}(u + \gamma_0)$$

For $X_2$, we have

$$\langle z, \zeta \rangle = 0, \|z\|_2 = u, \|\zeta\|_2 = r$$
$$\|[X^\top]_i\|_2 \leq u + r, \|X^\top\|_F \leq \sqrt{L}(u + r)$$

$\square$

**Proposition 1.** *By signal-noise decomposition, we have the updating rules for signal weight and noise weight:*

$$\overline{U}_t = -\sum_{s=1}^{t} \eta \left(1 - \eta\lambda\right)^{t-s} \nabla_{U_{s-1}} \widehat{L}(U_{s-1}),$$

$$\widetilde{U}_t = \left(1 - \eta\lambda\right)^{t} U_0 - \sum_{s=1}^{t} \eta \left(1 - \eta\lambda\right)^{t-s} \xi_{s-1}.$$

*Proof.* Decoupling the signal and noise, signal weight $\overline{U}$ is affected by the gradient updates, and noise weight $\widetilde{U}$ is affected by noise $\xi$. With $U_{t+1} = (1 - \gamma_t\lambda)U_t - \gamma_t(\nabla_U \widehat{L}(U_t) + \xi_t)$,

$$\overline{U}_t = -\sum_{s=1}^{t} \gamma_{s-1} \left(\prod_{i=s}^{t-1}(1 - \gamma_i\lambda)\right) \nabla_{U_{s-1}} \widehat{L}(U_{s-1})$$

$$\widetilde{U}_t = \left(\prod_{i=0}^{t-1}(1 - \gamma_i\lambda)\right) U_0 - \sum_{s=1}^{t} \gamma_{s-1} \left(\prod_{i=s}^{t-1}(1 - \gamma_i\lambda)\right) \xi_{s-1}$$

When constant learning rate $\gamma_t = \eta$,

$$\overline{U}_t = -\sum_{s=1}^{t} \eta \left(1 - \eta\lambda\right)^{t-s} \nabla_{U_{s-1}} \widehat{L}(U_{s-1})$$

$$\widetilde{U}_t = \left(1 - \eta\lambda\right)^{t} U_0 - \sum_{s=1}^{t} \eta \left(1 - \eta\lambda\right)^{t-s} \xi_{s-1}. \tag{7}$$

Since $U = \begin{bmatrix} W & 0 \\ 0 & V \end{bmatrix}$, then

$$\begin{bmatrix} W_{t+1} & 0 \\ 0 & V_{t+1} \end{bmatrix} = (1 - \gamma_t\lambda) \begin{bmatrix} W_t & 0 \\ 0 & V_t \end{bmatrix} - \gamma_t(\nabla_U \widehat{L}(U_t) + \xi_t)$$

$$W_{t+1} = (1 - \gamma_t\lambda)W_t - \gamma_t(\nabla_{W_t} \widehat{L}(U_t) + \xi_t)$$

$$V_{t+1} = (1 - \gamma_t\lambda)V_t - \gamma_t(\nabla_{V_t} \widehat{L}(U_t) + \xi_t)$$

Similar to the signal-noise decomposition of $U$ with learning rate $\gamma_t = \eta$, we naturally have

$$\overline{W}_t = -\sum_{s=1}^{t} \eta \left(1 - \eta\lambda\right)^{t-s} \nabla_{W_{s-1}} \widehat{L}(U_{s-1})$$

$$\widetilde{W}_t = \left(1 - \eta\lambda\right)^{t} W_0 - \sum_{s=1}^{t} \eta \left(1 - \eta\lambda\right)^{t-s} \xi_{s-1} \tag{8}$$

$$\overline{V}_t = -\sum_{s=1}^{t} \eta \left(1 - \eta\lambda\right)^{t-s} \nabla_{V_{s-1}} \widehat{L}(U_{s-1})$$

$$\widetilde{V}_t = \left(1 - \eta\lambda\right)^{t} V_0 - \sum_{s=1}^{t} \eta \left(1 - \eta\lambda\right)^{t-s} \xi_{s-1} \tag{9}$$

$\square$

**Proposition 2.** *For any $U \in \mathbb{R}^{2d\times 2d}, W, V \in \mathbb{R}^{d\times d}, X \in \mathbb{R}^{2d\times 2L}, X_1, X_2 \in \mathbb{R}^{d\times L}, \widetilde{Y} \in \mathbb{R}^{1\times 2L}, Y \in \mathbb{R}^{1\times L}$, then we have the derivative over weight $U$ of empirical loss, i.e. $\nabla\widehat{L}(U)$*

*and its component $[\nabla\widehat{L}(U)]_i$ is the $i$-th row of $\nabla\widehat{L}(U)$,*

$$\nabla\widehat{L}(U) = \widehat{\mathbb{E}}\left[1/2L \cdot l'(f(U;X,\widetilde{Y}))X \cdot diag\left(\mathbb{1}(X^\top U x_L)\right) x_L^\top\right]$$

$$[\nabla\widehat{L}(U)]_i = \widehat{\mathbb{E}}\left[1/2L \cdot l'(f(U;X,\widetilde{Y}))\mathbb{1}([X^\top]_i U x_L)[X^\top]_i x_L^\top\right]$$

*Additionally, for the derivative over weight $W$,*

$$\nabla_W\widehat{L}(U) = \widehat{\mathbb{E}}\left[1/2L \cdot l'(f(U;X,\widetilde{Y}))X_1 \cdot diag\left(\mathbb{1}(X_1^\top W x_{L,1})\right) x_{L,1}^\top\right]$$

$$[\nabla_W\widehat{L}(U)]_i = \widehat{\mathbb{E}}\left[1/2L \cdot l'(f(U;X,\widetilde{Y}))\mathbb{1}([X_1^\top]_i W x_{L,1})[X_1]_i x_{L,1}^\top\right]$$

*for the derivative over weight $V$,*

$$\nabla_V\widehat{L}(U) = \widehat{\mathbb{E}}\left[1/2L \cdot l'(f(U;X,\widetilde{Y}))X_2 \cdot diag\left(\mathbb{1}(X_2^\top V x_{L,2})\right) x_{L,2}^\top\right]$$

$$[\nabla_V\widehat{L}(U)]_i = \widehat{\mathbb{E}}\left[1/2L \cdot l'(f(U;X,\widetilde{Y}))\mathbb{1}([X_2^\top]_i V x_{L,2})[X_2]_i x_{L,2}^\top\right]$$

*Proof.* According to the definition of training objective, define

$$l(f(U;X,\widetilde{Y})) = -\log\sigma\left(y_L f\left(U;X,\widetilde{Y}\right)\right)$$

then we have the derivative of empirical loss with weight $U$,

$$\nabla\widehat{L}(U) = \widehat{\mathbb{E}}\left[l'(f(U;X,\widetilde{Y}))\nabla(y_L f(U;X,\widetilde{Y}))\right]$$

$$= \widehat{\mathbb{E}}\left[l'(f(U;X,\widetilde{Y}))y_L \nabla\left(\widetilde{Y}/2L \cdot \text{ReLU}\left(X^\top U x_L\right)\right)\right]$$

$$= \widehat{\mathbb{E}}\left[1/2L \cdot l'(f(U;X,\widetilde{Y}))y_L \sum_{i=1}^{2L} y_i \nabla\text{ReLU}\left([X^\top]_i U x_L\right)\right]$$

$$= \widehat{\mathbb{E}}\left[1/2L \cdot l'(f(U;X,\widetilde{Y}))y_L \sum_{i=1}^{2L} y_i \mathbb{1}([X^\top]_i U x_L)[X^\top]_i x_L^\top\right]$$

$$= \widehat{\mathbb{E}}\left[1/2L \cdot l'(f(U;X,\widetilde{Y}))X \cdot \text{diag}\left(\mathbb{1}(X^\top U x_L)\right) x_L^\top\right]$$

and $[\nabla\widehat{L}(U)]_i = \widehat{\mathbb{E}}\left[1/2L \cdot l'(f(U;X,\widetilde{Y}))\mathbb{1}([X^\top]_i U x_L)[X^\top]_i x_L^\top\right]$.

Furthermore, when taking derivative over $W$,

$$\nabla_W\widehat{L}(U) = \widehat{\mathbb{E}}\left[l'(f(U;X,\widetilde{Y}))\nabla_W\left(y_L f(U;(X,\widetilde{Y}))\right)\right]$$

$$= \widehat{\mathbb{E}}\left[l'(f(U;X,\widetilde{Y}))y_L \nabla_W\left(\widetilde{Y}/2L \cdot \text{ReLU}\left(X^\top U x_L\right)\right)\right]$$

$$= \widehat{\mathbb{E}}\left[1/2L \cdot l'(f(U;X,\widetilde{Y}))y_L \sum_{i=1}^{L} \begin{bmatrix} y_i & y_i \end{bmatrix} \nabla_W\text{ReLU}\left(\begin{bmatrix} [X_1^\top]_i W x_{L,1} \\ [X_2^\top]_i V x_{L,2} \end{bmatrix}\right)\right]$$

$$= \widehat{\mathbb{E}}\left[1/2L \cdot l'(f(U;X,\widetilde{Y}))y_L \sum_{i=1}^{L} y_i \mathbb{1}([X_1^\top]_i W x_{L,1})[X_1]_i x_{L,1}^\top\right]$$

$$= \widehat{\mathbb{E}}\left[1/2L \cdot l'(f(U;X,\widetilde{Y}))X_1 \cdot \text{diag}\left(\mathbb{1}(X_1^\top W x_{L,1})\right) x_{L,1}^\top\right]$$

and $[\nabla_W\widehat{L}(U)]_i = \widehat{\mathbb{E}}\left[1/2L \cdot l'(f(U;X,\widetilde{Y}))\mathbb{1}([X_1^\top]_i W x_{L,1})[X_1]_i x_{L,1}^\top\right]$. Similarly, when taking derivative over $V$, we have

$$\nabla_V\widehat{L}(U) = \widehat{\mathbb{E}}\left[1/2L \cdot l'(f(U;X,\widetilde{Y}))X_2 \cdot \text{diag}\left(\mathbb{1}(X_2^\top V x_{L,2})\right) x_{L,2}^\top\right]$$

$$[\nabla_V\widehat{L}(U)]_i = \widehat{\mathbb{E}}\left[1/2L \cdot l'(f(U;X,\widetilde{Y}))\mathbb{1}([X_2^\top]_i V x_{L,2})[X_2]_i x_{L,2}^\top\right]$$

$\square$

**Proposition 3.** *Assume that $\widehat{L}$ is $K$-Lipschitz continuous, then we have*

$$\left\|\nabla\widehat{L}(U)\right\|_F \lesssim K, \left\|[\nabla\widehat{L}(U)]_i\right\|_2 \lesssim \frac{K}{\sqrt{2d}}$$

$$\left\|\nabla_W\widehat{L}(U)\right\|_F \lesssim K, \left\|[\nabla_W\widehat{L}(U)]_i\right\|_2 \lesssim \frac{K}{\sqrt{d}}$$

$$\left\|\nabla_V\widehat{L}(U)\right\|_F \lesssim K, \left\|[\nabla_V\widehat{L}(U)]_i\right\|_2 \lesssim \frac{K}{\sqrt{d}}$$

**Proposition 4.** *With Assumption 2 and Proposition 3 , we have that signal weight norm satisfies, for $X_1$*

$$\|\overline{U}_t\|_F \lesssim \frac{K}{\lambda}, \|[\overline{U}_t]_i\|_2 \lesssim \frac{K}{\lambda\sqrt{2d}}$$

$$\|\overline{W}_t\|_F \lesssim \frac{K}{\lambda}, \|[\overline{W}_t]_i\|_2 \lesssim \frac{K}{\lambda\sqrt{d}}$$

$$\|\overline{V}_t\|_F \lesssim \frac{K}{\lambda}, \|[\overline{V}_t]_i\|_2 \lesssim \frac{K}{\lambda\sqrt{d}}$$

*Proof.* By Equation 7, 8 and 9, when $0 < 1 - \eta\lambda < 1$, i.e., $0 < \eta\lambda < 1$,

$$\|\overline{U}_t\|_F = \sum_{\tau=1}^{t} \eta(1-\eta\lambda)^{t-\tau}\left\|\nabla\widehat{L}(U_{\tau-1})\right\|_F \lesssim \frac{K}{\lambda}$$

$$\|[\overline{U}_t]_i\|_2 = \sum_{\tau=1}^{t} \eta(1-\eta\lambda)^{t-\tau}\left\|[\nabla\widehat{L}(U_{\tau-1})]_i\right\|_2 \lesssim \frac{K}{\lambda\sqrt{2d}}$$

$$\|\overline{W}_t\|_F = \sum_{\tau=1}^{t} \eta(1-\eta\lambda)^{t-\tau}\left\|\nabla_W\widehat{L}(U_{\tau-1})\right\|_F \lesssim \frac{K}{\lambda}$$

$$\|[\overline{W}_t]_i\|_2 = \sum_{\tau=1}^{t} \eta(1-\eta\lambda)^{t-\tau}\left\|[\nabla_W\widehat{L}(U_{\tau-1})]_i\right\|_2 \lesssim \frac{K}{\lambda\sqrt{d}}$$

$$\|\overline{V}_t\|_F = \sum_{\tau=1}^{t} \eta(1-\eta\lambda)^{t-\tau}\left\|\nabla_V\widehat{L}(U_{\tau-1})\right\|_F \lesssim \frac{K}{\lambda}$$

$$\|[\overline{V}_t]_i\|_2 = \sum_{\tau=1}^{t} \eta(1-\eta\lambda)^{t-\tau}\left\|[\nabla_V\widehat{L}(U_{\tau-1})]_i\right\|_2 \lesssim \frac{K}{\lambda\sqrt{d}}$$

Furthermore,

$$\left\|[X^\top\overline{U}]_i x_L\right\|_2 \leq \|[X]_i\|_2\|\overline{U}\|_F\|x_L\|_2 \lesssim \frac{K(u+m)^2}{\lambda}$$

$$\left\|[X_1^\top\overline{W}]_i x_{L,1}\right\|_2 \leq \|[X_1]_i\|_2\|\overline{W}\|_F\|x_{L,1}\|_2 \lesssim \frac{K(u+\gamma_0)^2}{\lambda}$$

$$\left\|[X_2^\top\overline{V}]_i x_{L,2}\right\|_2 \leq \|[X_2]_i\|_2\|\overline{V}\|_F\|x_{L,2}\|_2 \lesssim \frac{K(u+r)^2}{\lambda}$$

$\square$

**Proposition 5.** *For time $\tau \leq t$, we have*

*Proof.* For $\tau \leq t$,

$$\widetilde{U}_t = (1-\eta\lambda)^{t-\tau}\widetilde{U}_\tau - \sum_{t'=1}^{t-\tau} \eta(1-\eta\lambda)^{t-\tau-t'}\zeta_{\tau+t'-1}$$

$$= (1-\eta\lambda)^{t-\tau}\widetilde{U}_\tau + \Xi_{t,\tau}$$

where $\Xi_{t,\tau} = -\sum_{t'=1}^{t-\tau} \eta(1-\eta\lambda)^{t-\tau-t'}\zeta_{\tau+t'-1}$.

$\square$

**Lemma 6** (Refer to Lemma A.8 in Li et al. (2019), Lemma 8.2 of Allen-Zhu et al. (2019)). *Let $X \in \mathbb{R}^{2d \times 2L}, x_L \in \mathbb{R}^{2d}$ be a fixed example, with $\|x_L\|_2 \leq B$ and $\|X\|_F \leq \sqrt{2L}B$. With Assumption 2 and Proposition 4, for every $\tau > 0$, let $U = \overline{U} + \widetilde{U}$ where $\widetilde{U} \in \mathbb{R}^{2d \times 2d}$ is a random variable whose columns have i.i.d distribution $\mathcal{N}(0, \tau_0^2 I_{2d \times 2d})$ and $\widetilde{Y} \in \mathbb{R}^{2L}$ such that each entry of $\widetilde{Y}$ is i.i.d. uniform in $\{-1, 1\}$. We have that, w.h.p over the randomness of $\widetilde{U}$ and $\widetilde{Y}$, $\forall \overline{U} \in \mathbb{R}^{2d \times 2d}$, we have that*

$$\|\mathbb{1}(X^\top U x_L) - \mathbb{1}(X^\top \widetilde{U} x_L)\|_1 \lesssim K^{4/3} \lambda^{-4/3} \tau_0^{-4/3} L^{2/3} \triangleq \epsilon_U$$

*Furthermore,*

$$\left| N_U(\overline{U}; X, \widetilde{Y}) - N_{\widetilde{U}}(\overline{U}; X, \widetilde{Y}) \right| \lesssim (u+m)^2 K^{7/3} \lambda^{-7/3} \tau_0^{-4/3} L^{-1/3}$$

*Proof.* With Lemma A.8 of Li, we can compute the difference of activation patterns.

$$\begin{aligned}
\|\mathbb{1}(X^\top U x_L) - \mathbb{1}(X^\top \widetilde{U} x_L)\|_1 &\lesssim \|X^\top \overline{U}\|_F^{4/3} \tau_0^{-4/3} L^{2/3} \\
&\lesssim ((2L)^{1/2} B)^{4/3} \|\overline{U}\|_F^{4/3} (\tau_0 (2L)^{1/2} B)^{-4/3} L^{2/3} \\
&\lesssim \|\overline{U}\|_F^{4/3} \tau_0^{-4/3} L^{2/3}
\end{aligned}$$

With Assumption 2, $B = u + m$, and Proposition 4, then

$$\begin{aligned}
\|\mathbb{1}(X^\top U x_L) - \mathbb{1}(X^\top \widetilde{U} x_L)\|_1 &\lesssim \|\overline{U}\|_F^{4/3} \tau_0^{-4/3} L^{2/3} \\
&\lesssim \|\overline{U}\|_F^{4/3} \tau_0^{-4/3} L^{2/3} \\
&\lesssim K^{4/3} \lambda^{-4/3} \tau_0^{-4/3} L^{2/3} \\
&= \left( \frac{L K^2}{\lambda^2 \tau_0^2} \right)^{2/3}
\end{aligned}$$

Furthermore,

$$\begin{aligned}
\left| N_U(\overline{U}; X, \widetilde{Y}) - N_{\widetilde{U}}(\overline{U}; X, \widetilde{Y}) \right| &= \left\| \widetilde{Y}/2L \cdot \left( \mathbb{1}\left(X^\top U x_L\right) - \mathbb{1}\left(X^\top \widetilde{U} x_L\right) \right) \odot \left(X^\top \overline{U} x_L\right) \right\| \\
&\leq \frac{1}{2L} \sum_{i \in [2L]} \left| [\widetilde{Y}]_i \right| \left| \mathbb{1}\left([X^\top]_i U x_L\right) - \mathbb{1}\left([X^\top]_i \widetilde{U} x_L\right) \right| \left| [X^\top]_i \overline{U} x_L \right| \\
&\leq \frac{1}{2L} \left\| \mathbb{1}\left(X^\top U x_L\right) - \mathbb{1}\left(X^\top \widetilde{U} x_L\right) \right\|_1 \max_i \left| [X^\top \overline{U}]_i x_L \right| \\
&\lesssim K^{4/3} \lambda^{-4/3} \tau_0^{-4/3} L^{-1/3} \frac{K(u+m)^2}{\lambda} \\
&\lesssim (u+m)^2 K^{7/3} \lambda^{-7/3} \tau_0^{-4/3} L^{-1/3}
\end{aligned}$$

$\square$

**Corollary 2.** *Let $X_1 \in \mathbb{R}^{d \times L}, x_{L,1} \in \mathbb{R}^d$ be a fixed example, with Assumption 2 and Proposition 4, $\|x_{L,1}\|_2 \leq u + \gamma_0$ and $\|X_1\|_F \leq \sqrt{L}(u + \gamma_0)$. Then, w.h.p over the randomness of $\widetilde{W}$ and $Y$, $\forall \overline{W} \in \mathbb{R}^{d \times d}$, we have that*

$$\|\mathbb{1}(X_1^\top W x_{L,1}) - \mathbb{1}(X_1^\top \widetilde{W} x_{L,1})\|_1 \lesssim K^{4/3} \lambda^{-4/3} \tau_0^{-4/3} L^{2/3} \triangleq \epsilon_W \tag{10}$$

*Furthermore,*

$$\left| N_W(\overline{W}; X_1, Y) - N_{\widetilde{W}}(\overline{W}; X_1, Y) \right| \lesssim (u + \gamma_0)^2 K^{7/3} \lambda^{-7/3} \tau_0^{-4/3} L^{-1/3}$$

***Note.*** In $\epsilon_W$, $K$ is the Lipschitz constant, $\lambda$ denotes the $L_2$ regularization coefficient, $\tau_0$ denotes the variance of initialization parameter and $L$ is prompt length. When with choices in Assumption 1, we have $\epsilon_W = (\text{Poly}(d))^{2/3}$.

**Corollary 3.** *Let $X_2 \in \mathbb{R}^{d \times L}, x_{L,2} \in \mathbb{R}^d$ be a fixed example, with Assumption 2 and Proposition 4, $\|x_{L,2}\|_2 \leq u + r$ and $\|X_2\|_F \leq \sqrt{L}(u+r)$. Then, w.h.p over the randomness of $\widetilde{V}$ and $Y$, $\forall \overline{V} \in \mathbb{R}^{d \times d}$, we have that*

$$\|\mathbb{1}(X_2^\top V x_{L,2}) - \mathbb{1}(X_2^\top \widetilde{V} x_{L,2})\|_1 \lesssim K^{4/3} \lambda^{-4/3} \tau_0^{-4/3} L^{2/3} \triangleq \epsilon_V$$

*Furthermore,*

$$\left| N_V(\overline{V}; X_2, Y) - N_{\widetilde{V}}(\overline{V}; X_2, Y) \right| \lesssim (u+r)^2 K^{7/3} \lambda^{-7/3} \tau_0^{-4/3} L^{-1/3}$$

**Lemma 7.** *Under the same setting as Lemma 6, we have*

$$\left\| \mathbb{1}\left(X^\top U_{t_1+t_2} x_L\right) - \mathbb{1}\left(X^\top U_{t_1} x_L\right) \right\|_1 \lesssim \epsilon_U + L\sqrt{\frac{\eta_2}{\eta_1}} + \sqrt{L \log d}$$

*where $\epsilon_U = K^{4/3} \lambda^{-4/3} \tau_0^{-4/3} L^{2/3}$. Furthermore,*

$$\left| N_{U_{t_1+t_2}}(\overline{U}_{t_1+t_2}; X, \widetilde{Y}) - N_{U_{t_1}}(\overline{U}_{t_1+t_2}; X, \widetilde{Y}) \right| \lesssim \left( \epsilon_U + L\sqrt{\frac{\eta_2}{\eta_1}} + \sqrt{L \log d} \right) \frac{K(u+m)^2}{L\lambda}$$

*and*

$$\left| N_{U_{t_1+t_2}}(U_{t_1+t_2}; X, Y) - N_{U_{t_1}}(\overline{U}_{t_1+t_2}; X, Y) \right|$$

$$\lesssim \frac{\epsilon(u+r)^4 \sqrt{d}}{\lambda L} + \frac{(u+r)^4 \sqrt{Ld\eta_2/\eta_1}}{\lambda} + (u+r)^4 \sqrt{d \log d}$$

$$\lesssim \left( \epsilon + \sqrt{\frac{\eta_2}{\eta_1}} L + \sqrt{L \log d} \right) \frac{(u+r)^4 \sqrt{d}}{\lambda L}$$

*Proof.* To analysis that how the sign of $U_{t_1+t_2}$ correlates to $U_{t_1}$,

$$\left\| \mathbb{1}\left(X^\top U_{t_1+t_2} x_L\right) - \mathbb{1}\left(X^\top U_{t_1} x_L\right) \right\|_1$$

$$= \left\| \mathbb{1}\left(X^\top U_{t_1+t_2} x_L\right) - \mathbb{1}\left(X^\top \widetilde{U}_{t_1+t_2} x_L\right) + \mathbb{1}\left(X^\top \widetilde{U}_{t_1+t_2} x_L\right) - \mathbb{1}\left(X^\top \widetilde{U}_{t_1} x_L\right) \right.$$

$$\left. + \mathbb{1}\left(X^\top \widetilde{U}_{t_1} x_L\right) - \mathbb{1}\left(X^\top U_{t_1} x_L\right) \right\|_1$$

$$\leq \underbrace{\left\| \mathbb{1}\left(X^\top U_{t_1+t_2} x_L\right) - \mathbb{1}\left(X^\top \widetilde{U}_{t_1+t_2} x_L\right) \right\|_1}_{A} + \underbrace{\left\| \mathbb{1}\left(X^\top \widetilde{U}_{t_1+t_2} x_L\right) - \mathbb{1}\left(X^\top \widetilde{U}_{t_1} x_L\right) \right\|_1}_{B}$$

$$+ \underbrace{\left\| \mathbb{1}\left(X^\top \widetilde{U}_{t_1} x_L\right) - \mathbb{1}\left(X^\top U_{t_1} x_L\right) \right\|_1}_{C}$$

For term $A$ and term $C$, With Lemma 6, we have

$$\|\mathbb{1}(X^\top U_{t_1+t_2} x_L) - \mathbb{1}(X^\top \widetilde{U}_{t_1+t_2} x_L)\|_1 \lesssim K^{4/3} \lambda^{-4/3} \tau_0^{-4/3} L^{2/3} \triangleq \epsilon_U \tag{11}$$

$$\|\mathbb{1}(X^\top U_{t_1} x_L) - \mathbb{1}(X^\top \widetilde{U}_{t_1} x_L)\|_1 \lesssim K^{4/3} \lambda^{-4/3} \tau_0^{-4/3} L^{2/3} \triangleq \epsilon_U \tag{12}$$

For term $B$, we first analysis the relationship between $\widetilde{U}_{t_1+t_2}$ and $\widetilde{U}_{t_1}$. With Proposition 5, for $\tau \leq t$, we have

$$\widetilde{V}_t = (1 - \eta\lambda)^{t-\tau} \widetilde{V}_\tau - \sum_{t'=1}^{t-\tau} \eta(1-\eta\lambda)^{t-\tau-t'} \zeta_{\tau+t'-1}$$

$$= (1 - \eta\lambda)^{t-\tau} \widetilde{V}_\tau + \Xi_{t,\tau}$$

where $\Xi_{t,\tau} = -\sum_{t'=1}^{t-\tau} \eta(1-\eta\lambda)^{t-\tau-t'} \zeta_{\tau+t'-1}$. Assume that there are $t_1$ iterations in the first stage, let $\tau = t_1, t = t_1 + t_2$, and $t - \tau = t_2$, then

$$\widetilde{U}_{t_1+t_2} = (1 - \eta_2\lambda)^{t_2} \widetilde{U}_{t_1} - \sum_{t'=1}^{t_2} \eta_2(1-\eta_2\lambda)^{t_2-t'} \zeta_{t_1+t'-1}$$

$$= (1 - \eta_2\lambda)^{t_2} \widetilde{U}_{t_1} + \Xi_{t_1+t_2,t_1} \tag{13}$$

where $\Xi_{t_1+t_2,t_1} = -\sum_{t'=1}^{t_2} \eta_2(1-\eta_2\lambda)^{t_2-t'}\zeta_{t_1+t'-1}$.

Consider $[\Xi_{t_1+t_2,t_1}]_{ij} \sim \mathcal{N}(0, \sigma^2_{t_1+t_2,t_1})$, for $0 < 1-\eta_2\lambda < 1$, with a technical assumption that $\tau_\zeta^2 = \frac{\tau_0^2-(1-\eta_1\lambda)^2\tau_0^2}{\eta_1^2}$,

$$\sigma^2_{t_1+t_2,t_1} = \sum_{t'=1}^{t_2} \eta_2^2(1-\eta_2\lambda)^{2(t_2-t')}\tau_\zeta^2 = \eta_2^2\tau_\zeta^2\frac{1-(1-\eta_2\lambda)^{2t_2-1}}{\eta_2\lambda}$$

$$\leq \eta_2^2\tau_\zeta^2\frac{1}{\eta_2\lambda} = \eta_2^2\frac{\tau_0^2-(1-\eta_1\lambda)^2\tau_0^2}{\eta_1^2}\frac{1}{\eta_2\lambda} \leq \eta_2^2\frac{2\eta_1\lambda\tau_0^2}{\eta_1^2}\frac{1}{\eta_2\lambda}$$

$$= \frac{2\eta_2\tau_0^2}{\eta_1}$$

Since $\eta_2 \ll \eta_1$, then $\sigma_{t_1+t_2,t_1} \ll \tau_0$. This implies that additional noise in the second stage is small.

With Equation 13, we have

$$X^\top \widetilde{U}_{t_1+t_2}x_L = (1-\eta_2\lambda)^{t_2}X^\top\widetilde{U}_{t_1}x_L + X^\top\Xi_{t_1+t_2,t_1}x_L$$

since $[\widetilde{U}_{t_1}]_{ij} \sim \mathcal{N}(0,\tau_0^2)$ and $[\Xi_{t_1+t_2,t_1}]_i \sim \mathcal{N}(0,\sigma^2_{t_1+t_2,t_1})$,

$$\mathrm{Var}\left(X^\top\widetilde{U}_{t_1+t_2}x_L\right) \gtrsim \tau_0^2\|X\|_F^2\|x_L\|_2^2$$

$$\mathrm{Var}\left(X^\top\Xi_{t_1+t_2,t_1}x_L\right) \lesssim \frac{\eta_2\tau_0^2}{\eta_1}\|X\|_F^2\|x_L\|_2^2$$

then naturally we have

$$\Pr\left[\mathbb{1}\left(X^\top\widetilde{U}_{t_1+t_2}x_L\right) \neq \mathbb{1}\left(X^\top\Xi_{t_1,t_1+t_2}x_L\right)\right] \lesssim \sqrt{\frac{\eta_2\tau_0^2\|X\|_F^2\|x\|^2/\eta_1}{\tau_0^2\|X\|_F^2\|x\|^2}} = \sqrt{\frac{\eta_2}{\eta_1}} \qquad (14)$$

and

$$\mathbb{E}\left[\left|\mathbb{1}\left([X^\top]_i\widetilde{U}_{t_1+t_2}x_L\right) - \mathbb{1}\left([X^\top]_i\widetilde{U}_{t_1}x_L\right)\right|\right]$$

$$= \Pr\left[\mathbb{1}\left([X^\top]_i\widetilde{U}_{t_1+t_2}x_L\right) \neq \mathbb{1}\left([X^\top]_i\widetilde{U}_{t_1}x_L\right)\right]$$

$$\lesssim \sqrt{\frac{\eta_2}{\eta_1}}$$

Using Hoeffding's inequality in Lemma 1, with probability at least $1-\frac{1}{d}$,

$$\left\|\mathbb{1}\left(X^\top\widetilde{U}_{t_1+t_2}x_L\right) - \mathbb{1}\left(X^\top\widetilde{U}_{t_1}x_L\right)\right\|_1 \lesssim 2L\sqrt{\frac{\eta_2}{\eta_1}} + \sqrt{4L\log d}$$

$$\lesssim L\sqrt{\frac{\eta_2}{\eta_1}} + \sqrt{L\log d} \qquad (15)$$

Combine term $A, B, C$, Finally, with Equation 11, 12 and 15, we have

$$\left\|\mathbb{1}\left(X^\top U_{t_1+t_2}x_L\right) - \mathbb{1}\left(X^\top U_{t_1}x_L\right)\right\|_1 \lesssim \epsilon_U + L\sqrt{\frac{\eta_2}{\eta_1}} + \sqrt{L\log d}$$

where $\epsilon_U = K^{4/3}\lambda^{-4/3}\tau_0^{-4/3}L^{2/3}$.

Furthermore, with Proposition 4,

$$\left|N_{U_{t_1+t_2}}(\overline{U}_{t_1+t_2}; X, \widetilde{Y}) - N_{U_{t_1}}(\overline{U}_{t_1+t_2}; X, \widetilde{Y})\right|$$

$$= \frac{1}{L}\sum_{i\in[L]}|[Y]_i|\left|\mathbb{1}\left([X^\top]_iU_{t_1+t_2}x_L\right) - \mathbb{1}\left([X^\top]_iU_{t_1}x_L\right)\right|\left|[X^\top]_i\overline{U}_{t_1+t_2}x_L\right|$$

$$\leq \frac{1}{L}\left\|\mathbb{1}\left(X^\top U_{t_1+t_2}x_L\right) - \mathbb{1}\left(X^\top U_{t_1}x_L\right)\right\|_1 \max_i\left|[X^\top\overline{U}_{t_1+t_2}]_ix_L\right|$$

$$\lesssim \left(\epsilon_U + L\sqrt{\frac{\eta_2}{\eta_1}} + \sqrt{L\log d}\right)\frac{K(u+m)^2}{L\lambda}$$

$\square$

**Corollary 4.** *Let $X_1 \in \mathbb{R}^{d \times L}, x_{L,1} \in \mathbb{R}^d$ be a fixed example, with Assumption 2 and Proposition 4, $\|x_{L,1}\|_2 \le u + \gamma_0$ and $\|X_1\|_F \le \sqrt{L}(u + \gamma_0)$. Then, w.h.p over the randomness of $\widetilde{W}$ and $Y$, $\forall \overline{W} \in \mathbb{R}^{d \times d}$, we have that*

$$\left\| \mathbb{1}\left(X_1^\top W_{t_1 + t_2} x_{L,1}\right) - \mathbb{1}\left(X_1^\top W_{t_1} x_{L,1}\right) \right\|_1 \lesssim \epsilon_W + L\sqrt{\frac{\eta_2}{\eta_1}} + \sqrt{L \log d}$$

*where $\epsilon_W = K^{4/3} \lambda^{-4/3} \tau_0^{-4/3} L^{2/3}$. Furthermore,*

$$\left| N_{W_{t_1+t_2}}(\overline{W}_{t_1+t_2}; X_1, Y) - N_{W_{t_1}}(\overline{W}_{t_1+t_2}; X_1, Y) \right| \lesssim \left( \epsilon_W + L\sqrt{\frac{\eta_2}{\eta_1}} + \sqrt{L \log d} \right) \frac{K(u + \gamma_0)^2}{L\lambda}$$

**Corollary 5.** *Let $X_2 \in \mathbb{R}^{d \times L}, x_{L,2} \in \mathbb{R}^d$ be a fixed example, with Assumption 2 and Proposition 4, $\|x_{L,2}\|_2 \le u + r$ and $\|X_2\|_F \le \sqrt{L}(u + r)$. Then, w.h.p over the randomness of $\widetilde{V}$ and $Y$, $\forall \overline{V} \in \mathbb{R}^{d \times d}$, we have that*

$$\left\| \mathbb{1}\left(X_2^\top V_{t_1 + t_2} x_{L,2}\right) - \mathbb{1}\left(X_2^\top V_{t_1} x_{L,2}\right) \right\|_1 \lesssim \epsilon_V + L\sqrt{\frac{\eta_2}{\eta_1}} + \sqrt{L \log d}$$

*where $\epsilon_V = K^{4/3} \lambda^{-4/3} \tau_0^{-4/3} L^{2/3}$. Furthermore,*

$$\left| N_{V_{t_1+t_2}}(\overline{V}_{t_1+t_2}; X_2, Y) - N_{V_{t_1}}(\overline{V}_{t_1+t_2}; X_2, Y) \right| \lesssim \left( \epsilon_V + L\sqrt{\frac{\eta_2}{\eta_1}} + \sqrt{L \log d} \right) \frac{K(u + r)^2}{L\lambda}$$

**Proposition 6.** *Under the same setting as Lemma 6, we have w.h.p over the randomness of $\widetilde{U}$,*

$$\left| N_{\widetilde{U}}(\widetilde{U}; X, Y) \right| \lesssim \tau_0 (u + m)^2 \sqrt{\frac{d \log d}{L}}$$

*Proof.* We have

$$N_{\widetilde{U}}(\widetilde{U}; X, \widetilde{Y}) = \frac{1}{2L} \sum_{i \in [2L]} [\widetilde{Y}]_i \left[ [X^\top]_i \widetilde{U} x_L \right]_+$$

With Lemma 3, we have $\|\widetilde{U}\| \lesssim \tau_0 \sqrt{d}$. Then

$$\left\| \left[ [X^\top]_i \widetilde{U} x_L \right]_+ \right\|_2 \le \left\| [X^\top]_i \widetilde{U} x_L \right\|_2 \lesssim \tau_0 \sqrt{d} \|x\|_2^2$$

Using Hoeffding's inequality in Lemma 1, since $[Y]_i \in \{-1, 1\}$, $m_i = -\frac{1}{2L} \left\| \left[ [X^\top]_i \widetilde{U} x_L \right]_+ \right\|_2$, $M_i = \frac{1}{2L} \left\| \left[ [X^\top]_i \widetilde{U} x_L \right]_+ \right\|_2$, then we have

$$\Pr\left( \left| \frac{1}{2L} \sum_{i \in [2L]} [Y]_i \left[ [X^\top]_i \widetilde{U} x_L \right]_+ \right| \ge t \right) \le 2 \exp\left( -\frac{2t^2}{\sum_{i \in [2L]} \left( 2 \cdot \frac{1}{2L} \left\| \left[ [X^\top]_i \widetilde{U} x_L \right]_+ \right\|_2 \right)^2} \right)$$

$$\le 2 \exp\left( -\frac{2t^2}{\frac{1}{L^2} \sum_{i \in [2L]} \left\| \left[ [X^\top]_i \widetilde{U} x_L \right]_+ \right\|_2^2} \right)$$

$$\lesssim 2 \exp\left( -\frac{t^2}{\frac{1}{L} (\tau_0 \sqrt{d} \|x\|^2)^2} \right)$$

Let $\delta = 2\exp\left(-\frac{t^2}{\frac{1}{L}(\tau_0\sqrt{d}\|x\|^2)^2}\right)$, then with $\delta = \frac{1}{d}$

$$t = \sqrt{\frac{1}{L}(\tau_0\sqrt{d}\|x\|^2)^2 \log\frac{2}{\delta}}$$

$$\lesssim \tau_0\sqrt{d}\|x\|^2\sqrt{\frac{1}{L}\log\frac{2}{\delta}}$$

$$= \tau_0\|x\|^2\sqrt{\frac{d\log d}{L}}$$

Thus, with $1-\delta$ prob, we get

$$\left|N_{\widetilde{U}}(\widetilde{U};X,Y)\right| = \left|\frac{1}{2L}\sum_{i\in[2L]}[Y]_i\left[[X^\top]_i\widetilde{U}x_L\right]_+\right| \lesssim \tau_0\|x\|^2\sqrt{\frac{d\log d}{L}}$$

Since $\|x\|_2 \le u+m$, then

$$\left|N_{\widetilde{U}}(\widetilde{U};X,Y)\right| \lesssim \tau_0(u+m)^2\sqrt{\frac{d\log d}{L}}$$

$\square$

**Proposition 7.** *Under the same setting as Lemma 6, with Proposition 6, we have w.h.p over the randomness of $\widetilde{U}$, $\forall \overline{U} \in \mathbb{R}^{2d\times 2d}$,*

$$\left|N_U(\widetilde{U};X,\widetilde{Y}) - N_{\widetilde{U}}(\widetilde{U};X,\widetilde{Y})\right| \lesssim (u+m)^2 K^{7/3}\lambda^{-7/3}\tau_0^{-4/3}L^{-1/3}$$

*and*

$$\left|N_U(\widetilde{U};X,\widetilde{Y})\right| \lesssim (u+m)^2 K^{7/3}\lambda^{-7/3}\tau_0^{-4/3}L^{-1/3} + \tau_0(u+m)^2\sqrt{\frac{d\log d}{L}}$$

*Proof.* For every $i$, $\mathbb{1}([X^\top U]_i x_L) \ne \mathbb{1}([X^\top\widetilde{U}]_i x_L)$, it holds that $|[X^\top\widetilde{U}]_i x_L| \le |[X^\top\overline{U}]_i x_L|$. Then

$$\left|N_U(\widetilde{U};X,\widetilde{Y}) - N_{\widetilde{U}}(\widetilde{U};X,\widetilde{Y})\right| = \left\|\widetilde{Y}/2L\cdot\left(\mathbb{1}\left(X^\top U x_L\right) - \mathbb{1}\left(X^\top\widetilde{U}x_L\right)\right)\odot\left(X^\top\widetilde{U}x_L\right)\right\|$$

$$\le \frac{1}{2L}\sum_{i\in[2L]}\left|[\widetilde{Y}]_i\right|\left|\mathbb{1}\left([X^\top]_i U x_L\right) - \mathbb{1}\left([X^\top]_i\widetilde{U}x_L\right)\right|\left|[X^\top]_i\widetilde{U}x_L\right|$$

$$\le \frac{1}{2L}\left\|\mathbb{1}\left(X^\top U x_L\right) - \mathbb{1}\left(X^\top\widetilde{U}x_L\right)\right\|_1\max_i\left|[X^\top\overline{U}]_i x_L\right|$$

$$\lesssim K^{4/3}\lambda^{-4/3}\tau_0^{-4/3}L^{-1/3}\frac{K(u+m)^2}{\lambda}$$

$$\lesssim (u+m)^2 K^{7/3}\lambda^{-7/3}\tau_0^{-4/3}L^{-1/3}$$

With Proposition 6, using triangle inequality, we have

$$\left|N_U(\widetilde{U};X,\widetilde{Y})\right| \lesssim (u+m)^2 K^{7/3}\lambda^{-7/3}\tau_0^{-4/3}L^{-1/3} + \tau_0(u+m)^2\sqrt{\frac{d\log d}{L}}$$

$$= K(u+m)^2\lambda^{-1}\epsilon_U + \tau_0(u+m)^2\sqrt{\frac{d\log d}{L}}$$

$\square$

**Corollary 6.** *Let $X_1 \in \mathbb{R}^{d\times L}, x_{L,1} \in \mathbb{R}^d$ be a fixed example, with Assumption 2 and Proposition 7, $\|x_{L,1}\|_2 \le u+\gamma_0$ and $\|X_1\|_F \le \sqrt{L}(u+\gamma_0)$. Then, w.h.p over the randomness of $\widetilde{W}$ and $Y$, $\forall\overline{W}\in\mathbb{R}^{d\times d}$,*

$$\left|N_W(\widetilde{W};X_1,Y)\right| \lesssim (u+\gamma_0)^2 K^{7/3}\lambda^{-7/3}\tau_0^{-4/3}L^{-1/3} + \tau_0(u+\gamma_0)^2\sqrt{\frac{d\log d}{L}}$$

*With choice of small $u, r$, $\tau_0 = \mathcal{O}\left(\frac{1}{\sqrt{\log d}}\right)$, $\frac{1}{\lambda} = \mathcal{O}\left(\sqrt{\log d}\right)$ and $L = \Theta\left(Poly(d)\right)$, then*

$$\left|N_W(\widetilde{W};X_1,Y)\right| \lesssim \tau_0(u+\gamma_0)^2\sqrt{\frac{d\log d}{L}} \triangleq \epsilon_{W,1} \tag{16}$$

**Note.** In $\epsilon_{W,1}$, $\tau_0$ denotes the variance of initialization parameter, $L$ is prompt length and $d$ represents the input dimension. When with choices in Assumption 1, we have $\epsilon_{W,1} = \Theta\left(\frac{1}{\text{Poly}(d)}\right)$.

**Corollary 7.** *Let $X_2 \in \mathbb{R}^{d \times L}, x_{L,2} \in \mathbb{R}^d$ be a fixed example, with Assumption 2 and Proposition 7, $\|x_{L,2}\|_2 \leq u + r$ and $\|X_2\|_F \leq \sqrt{L}(u + r)$. Then, w.h.p over the randomness of $\widetilde{V}$ and $Y$, $\forall \overline{V} \in \mathbb{R}^{d \times d}$,*

$$\left| N_V(\widetilde{V}; X_2, Y) \right| \lesssim (u + r)^2 K^{7/3} \lambda^{-7/3} \tau_0^{-4/3} L^{-1/3} + \tau_0 (u + r)^2 \sqrt{\frac{d \log d}{L}}$$

*With choice of small $u, r$, $\tau_0 = \mathcal{O}\left(\frac{1}{\sqrt{\log d}}\right)$, $\frac{1}{\lambda} = \mathcal{O}\left(\sqrt{\log d}\right)$ and $L = \Theta\left(Poly(d)\right)$, then*

$$\left| N_V(\widetilde{V}; X_1, Y) \right| \lesssim \tau_0 (u + r)^2 \sqrt{\frac{d \log d}{L}} \triangleq \epsilon_{V,1} \tag{17}$$

**Note.** In $\epsilon_{V,1}$, $\tau_0$ denotes the variance of initialization parameter, $L$ is prompt length and $d$ represents the input dimension. When with choices in Assumption 1, we have $\epsilon_{V,1} = \Theta\left(\frac{1}{\text{Poly}(d)}\right)$.

## F  PROOF FOR THE ELEMENTARY STAGE

### F.1  PROOF OF THEOREM 1

**Theorem.** *In the elementary stage with $\eta_1 = \Theta(1)$ and $t \leq t_1 \triangleq \frac{1}{\eta_1 \lambda}$ where $\lambda$ denotes the $L_2$ regularization coefficient. With Assumption 1, initial weights $V_0 \to \mathbf{0}_{d \times d}$ and $N$ training prompts, it holds that*

***(a.1)** For the model parameter $V$ of network $g$, through gradient descent optimization from iteration $0$ to $t_1$, $\|\overline{V}_{t_1}\|_F$ satisfies*

$$\|\overline{V}_{t_1}\|_F = \Theta\left(\frac{1}{Poly(d)}\right).$$

***(a.2)** With random and small noise weight, the training loss of hard-to-fit component $\mathcal{Q}$ over signal weight (Definition in Equation 6) at iteration $t_1$ satisfies*

$$K_{t_1}^2\left(\overline{V}_{t_1}\right) \gtrsim \log 2 - \frac{1}{\sqrt{\log d}} - \sqrt{\frac{\log d}{N}}.$$

*Namely, the hard-to-fit component $\mathcal{Q}$ is not efficiently learned by $g$ within $t_1$ iterations.*

**Remark 6** (**Proof Sketch**). *We summarize the proof sketch and main techniques in Proof of Theorem 1. **At the starting point**, using signal-noise decomposition technique, we assume that the approximate output $\widetilde{g}$ uses noise part to compute activation and signal part as the weight to compute attention score. We show that $\widetilde{g}$ is very close to $g$ primarily through Corollary 3 and 7. Relevant corollaries are crucial for describing the differences in activation and network output under various activation and weight schemes. In the following analysis, we turn to focus on the approximation $\widetilde{g}$. **As a key step**, we focus on the network $g$'s ability to distinguish between positive and negative class samples by examining the differences in their respective outputs, i.e. $|\widetilde{g}_t(X_2, z - \zeta) + \widetilde{g}_t(X_2, z + \zeta) - 2\widetilde{g}_t(X_2, z)|$. Decompose it into two parts $\Phi$ and $\Psi$, where each part separately contains $z$ and $\zeta$. Then, give the upper bound of $\Phi$ and $\Psi$ by applying concentration inequalities like Chernoff, Bernstein and complex probability analysis like Gaussian integrals. Combining the above, we show that the prediction difference of the network for positive and negative samples is upper bounded by a small value, $1/\sqrt{\log d}$. Consequently, we derive a straightforward lower bound $2 - 1/\sqrt{\log d}$, demonstrating that the network $g$ cannot simultaneously make accurate predictions for both positive and negative samples.*

*From the network output, we further derive the changes in weight and loss. **For (a.1) and (a.2)**: At **an initial step**, to compute the high-probability proportions for query $x_{L,2} = z' = \{z - \zeta, z + \zeta\}$ and $x_{L,2} = z$, we express the training loss in terms of the network outputs for positive and negative class samples based on the proportion, dividing it into two parts with terms $g_{t_1}(X_2, z')$ and $g_{t_1}(X_2, z)$ respectively. **As an essential step**, by leveraging the convexity and Lipschitz properties of the logistic loss, we derive a lower bound for the training loss in (a.2). Using Taylor expansion techniques in combination with this lower bound, we further deduce a corollary of Theorem 1, which states: $|g_{t_1}(X_2, z)|, |g_{t_1}(X_2, z - \zeta)|, |g_{t_1}(X_2, z + \zeta)| \lesssim \frac{1}{(\log d)^{1/4}}$. By utilizing the expression of normalized ReLU self-attention, this corollary can be further extended to give (a.1).*

**Proof.** Using noise part to compute activation and signal part as weight.

$$\begin{aligned}
\widetilde{g}_t(X_2) &= N_{\widetilde{V}_t}(\overline{V}_t; X_2, Y) \\
&= Y\left(\mathbb{1}\left(X_2^\top \widetilde{V}_t x_{L,2}\right) \odot \left(X_2^\top \overline{V}_t x_{L,2}\right)\right)
\end{aligned}$$

Using triangle inequality, with Corollary 3 and 7,

$$|g_t(X_2) - \widetilde{g}_t(X_2)|$$

$$= \left| N_{V_t}(V_t; X_2, Y) - N_{\widetilde{V}_t}(\overline{V}_t; X_2, Y) \right|$$

$$= \left| N_{V_t}(\overline{V}_t; X_2, Y) + N_{V_t}(\widetilde{V}_t; X_2, Y) - N_{\widetilde{V}_t}(\overline{V}_t; X_2, Y) \right|$$

$$\leq \left| N_{V_t}(\overline{V}_t; X_2, Y) - N_{\widetilde{V}_t}(\overline{V}_t; X_2, Y) \right| + \left| N_{V_t}(\widetilde{V}_t; X_2, Y) \right|$$

$$\lesssim (u+r)^2 K^{7/3} \lambda^{-7/3} \tau_0^{-4/3} L^{-1/3} + (u+r)^2 K^{7/3} \lambda^{-7/3} \tau_0^{-4/3} L^{-1/3} + \tau_0 (u+r)^2 \sqrt{\frac{d \log d}{L}}$$

With choice of small $u, r, \tau_0 = \mathcal{O}\left(\frac{1}{\sqrt{\log d}}\right), \frac{1}{\lambda} = \mathcal{O}\left(\sqrt{\log d}\right)$ and $L = \Theta\left(\text{Poly}(d)\right)$,

$$|g_t(X_2) - \widetilde{g}_t(X_2)| \lesssim \frac{(\sqrt{\log d})^{11/3}}{(\text{Poly}(d))^{1/3}} + \frac{1}{\sqrt{\log d}} \sqrt{\frac{d \log d}{\text{Poly}(d)}}$$

$$\lesssim \frac{1}{\text{Poly}(d)}$$

In the following, we focus on $\widetilde{g}_t(X_2)$.

**Definition 1.** *For any time $t$, input $X \in \mathbb{R}^{d \times L}$ with query $x_L \in \mathbb{R}^d$, define $\epsilon_t^{X, x_L} \triangleq \{i \in [L] : [X^\top]_i \widetilde{V}_t x_L \geq 0\}$ and $\overline{\epsilon}_t^{X, x_L} \triangleq \{i \in [L] : [X^\top]_i \widetilde{V}_t x_L < 0\}$. Note that $X$ aligns with $X_2$ and $x_L$ aligns with $x_{L,2}$. Then $\mathbb{1}(\epsilon) \subset \{0, 1\}^L$. Naturally, we have*

$$\mathbb{1}(\epsilon_t^{X, x_L}) = \mathbb{1}(X^\top \widetilde{V}_t x_L).$$

Let $Q_t = \text{diag}(Y^\top) X_2^\top \overline{V}_t$, then

$$\widetilde{g}_t(X_2) = N_{\widetilde{V}_t}(\overline{V}_t; X_2, Y)$$

$$= Y/L \left( \mathbb{1}\left(X_2^\top \widetilde{V}_t x_{L,2}\right) \odot \left(X_2^\top \overline{V}_t x_{L,2}\right) \right)$$

$$= 1/L \cdot \mathbb{1}\left(X_2^\top \widetilde{V}_t x_{L,2}\right)^\top \left(\text{diag}(Y^\top) X_2^\top \overline{V}_t\right) x_{L,2}$$

$$= 1/L \cdot \mathbb{1}\left(X_2^\top \widetilde{V}_t x_{L,2}\right)^\top Q_t x_{L,2}$$

**To simplify, we use $X$ that represents $X_2$ and $x_L$ represents $x_{L,2}$, in this Lemma, if there are no confusion.**

Define $\widetilde{g}_t(X, z - \zeta)$ as sequence $X$ with $x_L = z - \zeta$, similarly for $\widetilde{g}_t(X, z + \zeta)$ and $\widetilde{g}_t(X, z)$. Then with Definition 1,

$$|\widetilde{g}_t(X, z - \zeta) + \widetilde{g}_t(X, z + \zeta) - 2\widetilde{g}_t(X, z)|$$

$$= 1/L \cdot \left| \mathbb{1}\left(\epsilon_t^{X, z-\zeta}\right)^\top Q_t(z - \zeta) + \mathbb{1}\left(\epsilon_t^{X, z+\zeta}\right)^\top Q_t(z + \zeta) - 2\mathbb{1}\left(\epsilon_t^{X, z}\right)^\top Q_t z \right|$$

$$\leq 1/L \cdot \underbrace{\left| \left(\mathbb{1}\left(\epsilon_t^{X, z-\zeta}\right) + \mathbb{1}\left(\epsilon_t^{X, z+\zeta}\right) - 2\mathbb{1}\left(\epsilon_t^{X, z}\right)\right)^\top Q_t z \right|}_{\Phi} + 1/L \cdot \underbrace{\left| \left(\mathbb{1}\left(\epsilon_t^{X, z+\zeta}\right) - \mathbb{1}\left(\epsilon_t^{X, z-\zeta}\right)\right)^\top Q_t \zeta \right|}_{\Psi}$$

**Deal with term $\Psi$.** First, consider the second term $\left| \left(\mathbb{1}\left(\epsilon_t^{X, z+\zeta}\right) - \mathbb{1}\left(\epsilon_t^{X, z-\zeta}\right)\right)^\top Q_t \zeta \right|$. With Assumption 2 that $\|\zeta\|_2 = r$,

$$\left| \left(\mathbb{1}\left(\epsilon_t^{X, z+\zeta}\right) - \mathbb{1}\left(\epsilon_t^{X, z-\zeta}\right)\right)^\top Q_t \zeta \right| \leq \left\| \left(\mathbb{1}\left(\epsilon_t^{X, z+\zeta}\right) - \mathbb{1}\left(\epsilon_t^{X, z-\zeta}\right)\right)^\top Q_t \right\|_2 \|\zeta\|_2$$

$$\leq r \left| \epsilon_t^{X, z+\zeta} \oplus \epsilon_t^{X, z-\zeta} \right| \cdot \max \|[Q_t]_i\|_2$$

**For $\epsilon_t^{X,z+\zeta} \oplus \epsilon_t^{X,z-\zeta}$ in term $\Psi$.** For $i \in \epsilon_t^{X,z+\zeta} \oplus \epsilon_t^{X,z-\zeta}$, with $[X^\top]_i \widetilde{V}_t(z + \zeta) \geq 0$ and $[X^\top]_i \widetilde{V}_t(z - \zeta) \leq 0$, then

$$-[X^\top]_i \widetilde{V}_t \zeta \leq [X^\top]_i \widetilde{V}_t z \leq [X^\top]_i \widetilde{V}_t \zeta$$

$$\left| [X^\top]_i \widetilde{V}_t z \right| \leq \left| [X^\top]_i \widetilde{V}_t \zeta \right|$$

Using chernoff bound for Gaussian variable in Lemma 5, let $\delta = 2 \exp\left(\frac{-t^2}{2\sigma^2}\right) = \frac{1}{d}$, then $t = \sigma\sqrt{2\log\frac{2}{\delta}} = \sigma\sqrt{2\log 2d}$. Substitute $\widetilde{V}_t$, given that it is a Gaussian vector with each component $[\widetilde{V}_t]_{ij} \sim \mathcal{N}(0, \tau_0^2)$, we have w.h.p $1 - \delta$

$$\left| [X^\top]_i \widetilde{V}_t \zeta \right| \leq r(u+r)|\widetilde{V}_t| \leq \tau_0 r(u+r)\sqrt{\log d}$$

$$\left| [X^\top]_i \widetilde{V}_t z \right| \leq \left| [X^\top]_i \widetilde{V}_t \zeta \right| \leq \tau_0 r(u+r)\sqrt{\log d}$$

i.e., $\Pr\left(\left| [X^\top]_i \widetilde{V}_t z \right| \leq \tau_0 r(u+r)\sqrt{\log d}\right) \gtrsim 1 - \frac{1}{d}$.

In the following, we try to give the upper bound of $\Pr\left(\left| [X^\top]_i \widetilde{V}_t z \right| \leq \tau_0 r(u+r)\sqrt{\log d}\right)$. Define the standardized variable $\frac{[X^\top \widetilde{V}_t]_i z}{\tau_0 u(u+r)} \sim \mathcal{N}(0, 1)$. We have $\Pr(|X| \leq a) = 2\Phi(a) - 1$ where $\Phi$ is CDF. of standard Gaussian random variable. Substituting $\frac{[X^\top \widetilde{V}_t]_i z}{\tau_0 u(u+r)}$ and $a = \frac{r\sqrt{\log d}}{u}$, then with large $d$ (i.e. large a),

$$\Pr\left(|[X^\top \widetilde{V}_t]_i z| \leq \tau_0 r(u+r)\sqrt{\log d}\right) = \Pr\left(\left| \frac{[X^\top \widetilde{V}_t]_i z}{\tau_0 u(u+r)} \right| \leq \frac{\tau_0 r(u+r)\sqrt{\log d}}{\tau_0 u(u+r)}\right)$$

$$= 2\Phi\left(\frac{r\sqrt{\log d}}{u}\right) - 1$$

$$\approx \frac{2 \cdot \frac{r\sqrt{\log d}}{u}}{\sqrt{2\pi}} \lesssim \frac{r\sqrt{\log d}}{u}$$

i.e., $\Pr\left(\left| [X^\top]_i \widetilde{V}_t z \right| \leq \tau_0 r(u+r)\sqrt{\log d}\right) \lesssim \frac{r\sqrt{\log d}}{u}$.

With Bernstein inequality in Lemma 2, define new random variable $R_i = \mathbb{I}(|[X^\top \widetilde{V}_t]_i z| \leq \tau_0 r(u+r)\sqrt{\log d})$ where $\mathbb{I}(\cdot)$ is the indicator function, $\mathbb{E}[R_i] = \Pr(|[X^\top \widetilde{V}_t]_i z| \leq \tau_0 r(u+r)\sqrt{\log d}) \lesssim \frac{r\sqrt{\log d}}{u}$. Then w.h.p. $1 - \delta = 1 - \frac{1}{d}$ we have

$$\frac{1}{L}\sum_{i=1}^{L} R_i - \mathbb{E}[R_i] \leq \sqrt{\frac{2\sigma^2 \log(1/\delta)}{L}} + \frac{2c\log(1/\delta)}{3L}$$

$$\sum_{i=1}^{L} R_i \leq L\sqrt{\frac{2\sigma^2 \log(1/\delta)}{L}} + L\frac{2c\log(1/\delta)}{3L} + \frac{rL\sqrt{\log d}}{u}$$

$$\lesssim \sqrt{L\log d} + \log d + \frac{rL\sqrt{\log d}}{u}$$

i.e. $|\epsilon_t^{X,z-\zeta} \oplus \epsilon_t^{X,z+\zeta}| \lesssim \sqrt{L\log d} + \log d + \frac{rL\sqrt{\log d}}{u}$. For sufficiently large $L$,

$$|\epsilon_t^{X,z-\zeta} \oplus \epsilon_t^{X,z+\zeta}| \lesssim \frac{rL\sqrt{\log d}}{u} \tag{18}$$

**For $[Q_t]_i$ in term $\Psi$.** For $Q_t = \mathrm{diag}(Y^\top)X^\top \overline{V}_t$, using Cauchy-Schwarz inequality, Assumption 2 and Proposition 4,

$$
\begin{aligned}
\|[Q_t]_i\|_2 = \left\|[Y^\top]_i[X^\top \overline{V}_t]_i\right\|_2 &= \left\|y_i \sum_{j=1}^d [X^\top]_{ij}[\overline{V}_t]_j\right\|_2 \\
&\leq \|[X]_i\|_2 \|\overline{V}_t\|_F \\
&\lesssim \frac{K(u+r)}{\lambda}
\end{aligned}
\tag{19}
$$

**Combine Equation 18 and 19.** For term B, we have

$$
\begin{aligned}
\left|\left(\mathbb{1}\left(\epsilon_t^{X,z+\zeta}\right) - \mathbb{1}\left(\epsilon_t^{X,z-\zeta}\right)\right)^\top Q_t \zeta\right| &\leq \left\|\left(\mathbb{1}\left(\epsilon_t^{X,z+\zeta}\right) - \mathbb{1}\left(\epsilon_t^{X,z-\zeta}\right)\right)^\top Q_t\right\|_2 \|\zeta\|_2 \\
&\leq r\left|\epsilon_t^{X,z+\zeta} \oplus \epsilon_t^{X,z-\zeta}\right| \cdot \max \|[Q_t]_i\|_2 \\
&\lesssim \frac{rL\sqrt{\log d}}{u} \cdot \frac{K(u+r)}{\lambda} \\
&\lesssim \frac{r(u+r)KL\sqrt{\log d}}{u\lambda}
\end{aligned}
$$

**Since then, we have completed term $\Psi$ in Equation.**

**Deal with term $\Phi$.** Consider term $\Phi = \left|\left(\mathbb{1}\left(\epsilon_t^{X,z-\zeta}\right) + \mathbb{1}\left(\epsilon_t^{X,z+\zeta}\right) - 2\mathbb{1}\left(\epsilon_t^{X,z}\right)\right)^\top Q_t z\right|$ in this part. Let $a = \left(\mathbb{1}\left(\epsilon_t^{X,z-\zeta}\right) + \mathbb{1}\left(\epsilon_t^{X,z+\zeta}\right) - 2\mathbb{1}\left(\epsilon_t^{X,z}\right)\right)^\top$, then

$$
\left(\mathbb{1}\left(\epsilon_t^{X,z-\zeta}\right) + \mathbb{1}\left(\epsilon_t^{X,z+\zeta}\right) - 2\mathbb{1}\left(\epsilon_t^{X,z}\right)\right)^\top Q_t z = a^\top Q_t z
$$

According to the definition of $Q_t$ and $\overline{V}_t$, we have

$$
\begin{aligned}
a^\top Q_t &= a^\top \mathrm{diag}(Y^\top)X^\top \overline{V}_t \\
&= a^\top \mathrm{diag}(Y^\top)X^\top \sum_{\tau=1}^t \eta_1 (1-\eta_1\lambda)^{t-\tau} \nabla_{V_{\tau-1}} \widehat{L}(U_{\tau-1}) \\
&= a^\top \sum_{\tau=1}^t \eta_1 (1-\eta_1\lambda)^{t-\tau} \Delta Q_{\tau-1}
\end{aligned}
$$

where $\Delta Q_\tau = \mathrm{diag}(Y^\top)X^\top \nabla_{V_\tau} \widehat{L}(U_\tau)$. Then

$$
\begin{aligned}
&\left|\left(\mathbb{1}\left(\epsilon_t^{X,z-\zeta}\right) + \mathbb{1}\left(\epsilon_t^{X,z+\zeta}\right) - 2\mathbb{1}\left(\epsilon_t^{X,z}\right)\right)^\top Q_t z\right| \\
&\leq \eta_1 u \sum_{\tau=1}^t \left\|\left(\mathbb{1}\left(\epsilon_t^{X,z-\zeta}\right) + \mathbb{1}\left(\epsilon_t^{X,z+\zeta}\right) - 2\mathbb{1}\left(\epsilon_t^{X,z}\right)\right)^\top \Delta Q_{\tau-1}\right\|_2
\end{aligned}
$$

**For $\Delta Q_\tau$ in term $\Phi$.**

**Definition 2.** *For any time $t$, input $X \in \mathbb{R}^{d \times L}$ with query $x_L \in \mathbb{R}^d$, define $\mathcal{G}_\tau^{X,x_L} \triangleq \{i \in [L] : [X^\top]_i V_\tau x_L \geq 0\}$ and $\overline{\mathcal{G}}_\tau^{X,x_L} \triangleq \{i \in [L] : [X^\top]_i V_\tau x_L < 0\}$. Similar to Definition 1, note that $X$ aligns with $X_2$ and $x_L$ aligns with $x_{L,2}$.*

Suppose $i, j$ satisfy that, for input $x_L = z - \zeta$ and $x_L = z + \zeta$ have the same activation pattern, then with Definition 2 we have

$$
i, j \in \mathcal{G}_\tau^{X,z-\zeta} \cap \mathcal{G}_\tau^{X,z+\zeta} \text{ or } i, j \in \overline{\mathcal{G}}_\tau^{X,z-\zeta} \cap \overline{\mathcal{G}}_\tau^{X,z+\zeta}
$$

Consider the relationship between $[\Delta Q_\tau]_i$ and $[\Delta Q_\tau]_j$ for the above $i, j$. We have $\Delta Q_\tau = \mathrm{diag}(Y^\top) X^\top \nabla_{V_\tau} \widehat{L}(U_\tau)$, then

$$[\Delta Q_\tau]_i = \left[\mathrm{diag}(Y^\top)\right]_i \left[X^\top \nabla_{V_\tau} \widehat{L}(U_\tau)\right]_i = y_i \left[X^\top \nabla_{V_\tau} \widehat{L}(U_\tau)\right]_i$$

$$[\Delta Q_\tau]_j = \left[\mathrm{diag}(Y^\top)\right]_j \left[X^\top \nabla_{V_\tau} \widehat{L}(U_\tau)\right]_j = y_j \left[X^\top \nabla_{V_\tau} \widehat{L}(U_\tau)\right]_j$$

With Proposition 2, then

$$[\Delta Q_\tau]_i = y_i[X^\top \nabla_{V_\tau}\widehat{L}(U_\tau)]_i = y_i[X^\top]_i \widehat{\mathbb{E}}\left[1/2L \cdot l'(f(U_\tau; X, Y))\mathbb{1}([X^\top]_i U_\tau x_L)[X]_i x_L^\top\right]$$

$$[\Delta Q_\tau]_j = y_j[X^\top \nabla_{V_\tau}\widehat{L}(U_\tau)]_j = y_j[X^\top]_j \widehat{\mathbb{E}}\left[1/2L \cdot l'(f(U_\tau; X, Y))\mathbb{1}([X^\top]_j U_\tau x_L)[X]_j x_L^\top\right]$$

Thus for $x_L \in \{0, z, z - \varsigma, z + \varsigma\}$. If $x_L = 0$, $[\Delta Q_\tau]_i = [\Delta Q_\tau]_j$. For all $x_L \in \{z, z - \varsigma, z + \varsigma\}$, $i, j \in \mathcal{G}_\tau^{X, z-\varsigma} \cap \mathcal{G}_\tau^{X, z+\varsigma}$, and then $i, j \in \mathcal{G}_\tau^{X, z}$. Thus,

$$\mathbb{1}([X^\top]_i V_\tau x_L) = \mathbb{1}([X^\top]_j V_\tau x_L) = 1$$

For fixed $X$, $[\nabla_{V_\tau}\widehat{L}(U_\tau)]_i = [\nabla_{V_\tau}\widehat{L}(U_\tau)]_j$. If $[X]_i = [X]_j$, then $y_i = y_j$,

$$[\Delta Q_\tau]_i = [\Delta Q_\tau]_j$$

If $[X]_i, [X]_j = z - \varsigma, z + \varsigma$, then $y_i = y_j$,

$$[\Delta Q_\tau]_i = (z - \varsigma)C, [\Delta Q_\tau]_j = (z + \varsigma)C$$
$$[\Delta Q_\tau]_i = (z + \varsigma)C, [\Delta Q_\tau]_j = (z - \varsigma)C$$
$$[\Delta Q_\tau]_i - [\Delta Q_\tau]_j = \pm 2\varsigma C$$

where $C = \widehat{\mathbb{E}}\left[l'(f(U_\tau; X, Y))\mathbb{1}([X^\top]_i U_\tau x_L)(z \pm \varsigma)x_L^\top\right]$. If $[X_2]_i, [X_2]_j = z \pm \varsigma, z$, then $y_i = -y_j$,

$$[\Delta Q_\tau]_i = (z \pm \varsigma)C, [\Delta Q_\tau]_j = zC$$
$$[\Delta Q_\tau]_i = zC, [\Delta Q_\tau]_j = (z \pm \varsigma)C$$
$$[\Delta Q_\tau]_i - [\Delta Q_\tau]_j = (-2z \pm \varsigma)C, \pm \varsigma C$$

where $C = \widehat{\mathbb{E}}\left[l'(f(U_\tau; X, Y))\mathbb{1}([X^\top]_i U_\tau x_L)(z(\pm\varsigma))x_L^\top\right]$.

**For $\left(\mathbb{1}\left(\epsilon_t^{X, z-\varsigma}\right) + \mathbb{1}\left(\epsilon_t^{X, z+\varsigma}\right) - 2\mathbb{1}\left(\epsilon_t^{X, z}\right)\right)^\top \Delta Q_\tau$ in term $\Phi$.** With Definition 1, we have

$$\mathbb{1}\left(\epsilon_t^{X, z-\varsigma}\right) + \mathbb{1}\left(\epsilon_t^{X, z+\varsigma}\right) - 2\mathbb{1}\left(\epsilon_t^{X, z}\right)$$

$$=\mathbb{1}\left(\epsilon_t^{X, z-\varsigma} \cap \epsilon_t^{X, z}\right) + \mathbb{1}\left(\epsilon_t^{X, z-\varsigma} \setminus \epsilon_t^{X, z}\right) + \mathbb{1}\left(\epsilon_t^{X, z+\varsigma} \cap \epsilon_t^{X, z}\right) + \mathbb{1}\left(\epsilon_t^{X, z+\varsigma} \setminus \epsilon_t^{X, z}\right)$$

$$- \mathbb{1}\left(\epsilon_t^{X, z} \cap \epsilon_t^{X, z-\varsigma}\right) - \mathbb{1}\left(\epsilon_t^{X, z} \setminus \epsilon_t^{X, z-\varsigma}\right) - \mathbb{1}\left(\epsilon_t^{X, z} \cap \epsilon_t^{X, z+\varsigma}\right) - \mathbb{1}\left(\epsilon_t^{X, z} \setminus \epsilon_t^{X, z+\varsigma}\right)$$

$$=\mathbb{1}\left(\epsilon_t^{X, z-\varsigma} \setminus \epsilon_t^{X, z}\right) + \mathbb{1}\left(\epsilon_t^{X, z+\varsigma} \setminus \epsilon_t^{X, z}\right) - \mathbb{1}\left(\epsilon_t^{X, z} \setminus \epsilon_t^{X, z-\varsigma}\right) - \mathbb{1}\left(\epsilon_t^{X, z} \setminus \epsilon_t^{X, z+\varsigma}\right)$$

$$=\underbrace{\mathbb{1}\left(\epsilon_t^{X, z+\varsigma} \setminus \epsilon_t^{X, z}\right) - \mathbb{1}\left(\epsilon_t^{X, z} \setminus \epsilon_t^{X, z-\varsigma}\right)}_{\text{Part I}} + \underbrace{\mathbb{1}\left(\epsilon_t^{X, z-\varsigma} \setminus \epsilon_t^{X, z}\right) - \mathbb{1}\left(\epsilon_t^{X, z} \setminus \epsilon_t^{X, z+\varsigma}\right)}_{\text{Part II}}$$

Observe that Part I and Part II are similar, and we deal with Part I first. Let $A = \epsilon_t^{X, z+\varsigma} \setminus \epsilon_t^{X, z}$ and $B = \epsilon_t^{X, z} \setminus \epsilon_t^{X, z-\varsigma}$. Similar to Definition 1, we give the following definition to divide sets $A$ and $B$, based on the above high probability results that is $\left|[X^\top]_i \widetilde{V}_t z\right| \lesssim \tau_0 r(u + r)\sqrt{\log d}$.

**Definition 3.** *For any time $\tau$, input $X \in \mathbb{R}^{d \times L}$ with query $x_L = z \in \mathbb{R}^d$, define $\mathcal{F}_\tau^+ \triangleq \{i \in [L] : [X^\top]_i \widetilde{V}_\tau z \gtrsim \tau_0 r(u + r)\sqrt{\log d}\}$, $\mathcal{F}_\tau^- \triangleq \{i \in [L] : [X^\top]_i \widetilde{V}_\tau z \lesssim -\tau_0 r(u + r)\sqrt{\log d}\}$ and $\mathcal{F}_\tau^c \triangleq \{i \in [L] : \left|[X^\top]_i \widetilde{V}_\tau z\right| \lesssim \tau_0 r(u + r)\sqrt{\log d}\}$. Similar to Definition 1, note that $X$ aligns with $X_2$.*

With Definition 3,

$$\left\| \left( \mathbb{1}\left( \epsilon_t^{X,z+\zeta} \setminus \epsilon_t^{X,z} \right) - \mathbb{1}\left( \epsilon_t^{X,z} \setminus \epsilon_t^{X,z-\zeta} \right) \right)^\top \Delta Q_\tau \right\|_2$$

$$= \left\| \sum_{i \in A} [\Delta Q_\tau]_i - \sum_{i \in B} [\Delta Q_\tau]_i \right\|_2$$

$$\leq \left\| \sum_{i \in A \cap \mathcal{F}_\tau^+} [\Delta Q_\tau]_i - \sum_{i \in B \cap \mathcal{F}_\tau^+} [\Delta Q_\tau]_i \right\|_2 + \left\| \sum_{i \in A \cap \mathcal{F}_\tau^-} [\Delta Q_\tau]_i - \sum_{i \in B \cap \mathcal{F}_\tau^-} [\Delta Q_\tau]_i \right\|_2$$

$$+ \left\| \sum_{i \in A \cap \mathcal{F}_\tau^c} [\Delta Q_\tau]_i - \sum_{i \in B \cap \mathcal{F}_\tau^c} [\Delta Q_\tau]_i \right\|_2$$

We have introduced the relationship between $[\Delta Q_\tau]_i$ and $[\Delta Q_\tau]_j$ for $i, j \in \mathcal{G}_\tau^{X,z-\zeta} \cap \mathcal{G}_\tau^{X,z+\zeta}$. In the following, we show that if $k, l \in \mathcal{F}_\tau^+$ (similar for $\mathcal{F}_\tau^-$ and $\mathcal{F}_\tau^c$) then $k, l \in \mathcal{G}_\tau^{X,z-\zeta} \cap \mathcal{G}_\tau^{X,z+\zeta}$, thus we have the same conclusion for $[\Delta Q_\tau]_k$ and $[\Delta Q_\tau]_l$.

Suppose $k, l$ satisfy that, when $x \in \{z - \zeta, z + \zeta\}$

$$[X^\top]_k \widetilde{V}_\tau x \gtrsim \tau_0 r(u + r)\sqrt{\log d}$$

$$[X^\top]_l \widetilde{V}_\tau x \gtrsim \tau_0 r(u + r)\sqrt{\log d}$$

Naturally, we have $[X^\top]_k \widetilde{V}_\tau z \gtrsim \tau_0 r(u + r)\sqrt{\log d}$ and $[X^\top]_l \widetilde{V}_\tau z \gtrsim \tau_0 r(u + r)\sqrt{\log d}$, i.e., $k, l \in \mathcal{F}_t^+$. Then

$$- \left| [X^\top]_k \overline{V}_\tau z \right| \leq [X^\top]_k \overline{V}_\tau z = [X^\top]_k (V_\tau - \widetilde{V}_\tau) z \leq \left| [X^\top]_k \overline{V}_\tau z \right|$$

and with Assumption 2 and Proposition 4,

$$[X^\top]_k V_\tau z \geq [X^\top]_k \widetilde{V}_\tau z - \left| [X^\top]_k \overline{V}_\tau z \right|$$

$$\geq \tau_0 r(u + r)\sqrt{\log d} - \frac{u(u + r)K}{\lambda}$$

$$\gtrsim \tau_0 r(u + r)\sqrt{\log d}$$

where the last inequality comes from $\frac{1}{\lambda} = \mathcal{O}(\sqrt{\log d})$. Since $\{[X^\top]_k V_\tau z \gtrsim \tau_0 r(u + r)\sqrt{\log d}\} \subset \{[X^\top]_k V_\tau z \geq 0\} \subset \mathcal{G}_\tau^{X,z-\zeta} \cap \mathcal{G}_\tau^{X,z+\zeta}$, then we have $k, l \in \mathcal{G}_\tau^{X,z-\zeta} \cap \mathcal{G}_t^{X,z+\zeta}$. Thus, if $k, l \in \mathcal{F}_\tau^+, \mathcal{F}_\tau^-, \mathcal{F}_\tau^c$, $[\Delta Q_\tau]_k$ and $[\Delta Q_\tau]_l$ hold the same conclusion as $[\Delta Q_\tau]_i$ and $[\Delta Q_\tau]_j$.

Therefore, with the definition of data structure, assume that the probability of $[X]_i = [X]_j$, i.e. $[\Delta Q_\tau]_i = [\Delta Q_\tau]_j$, is $P$, then

$$\left\| \left( \mathbb{1}\left( \epsilon_t^{X,z+\zeta} \setminus \epsilon_t^{X,z} \right) - \mathbb{1}\left( \epsilon_t^{X,z} \setminus \epsilon_t^{X,z-\zeta} \right) \right)^\top \Delta Q_\tau \right\|_2$$

$$= \left\| \sum_{i \in A} [\Delta Q_\tau]_i - \sum_{i \in B} [\Delta Q_\tau]_i \right\|_2$$

$$\leq \left\| \sum_{i \in A \cap \mathcal{F}_\tau^+} [\Delta Q_\tau]_i - \sum_{i \in B \cap \mathcal{F}_\tau^+} [\Delta Q_\tau]_i \right\|_2 + \left\| \sum_{i \in A \cap \mathcal{F}_\tau^-} [\Delta Q_\tau]_i - \sum_{i \in B \cap \mathcal{F}_\tau^-} [\Delta Q_\tau]_i \right\|_2$$

$$+ \left\| \sum_{i \in A \cap \mathcal{F}_\tau^c} [\Delta Q_\tau]_i - \sum_{i \in B \cap \mathcal{F}_\tau^c} [\Delta Q_\tau]_i \right\|_2$$

$$\leq \max \|[\Delta Q_\tau]_i\|_2 \left( |A \cap \mathcal{F}_\tau^+| + |B \cap \mathcal{F}_\tau^+| + |A \cap \mathcal{F}_\tau^-| + |B \cap \mathcal{F}_\tau^-| + |A \cap \mathcal{F}_\tau^c| + |B \cap \mathcal{F}_\tau^c| \right)$$

$$\leq (u + r)K \Big( P \big| |A \cap \mathcal{F}_\tau^+| - |B \cap \mathcal{F}_\tau^+| \big| + P \big| |A \cap \mathcal{F}_\tau^-| - |B \cap \mathcal{F}_\tau^-| \big| + (1 - P)\left( |A \cap \mathcal{F}_\tau^+| + |B \cap \mathcal{F}_\tau^+| \right)$$

$$+ (1 - P)\left( |A \cap \mathcal{F}_\tau^-| + |B \cap \mathcal{F}_\tau^-| \right) + |A \cap \mathcal{F}_\tau^c| + |B \cap \mathcal{F}_\tau^c| \Big)$$

**For** $|A \cap \mathcal{F}_\tau^+|$, $|B \cap \mathcal{F}_\tau^+|$ **and** $||A \cap \mathcal{F}_\tau^+| - |B \cap \mathcal{F}_\tau^+||$. It is related to $[X^\top]_i \widetilde{V}_t z, [X^\top]_i \widetilde{V}_t \zeta, [X^\top]_i \widetilde{V}_t z$. At time $\tau \leq t$, we can establish the relationship of $[X^\top]_i \widetilde{V}_\tau z, [X^\top]_i \widetilde{V}_t z$. With Proposition 5 and $\eta = \eta_1$, we have

$$[X^\top]_i \widetilde{V}_t z = (1 - \eta_1 \lambda)^{t-\tau} [X^\top]_i \widetilde{V}_\tau z - \sum_{t'=1}^{t-\tau} (1 - \eta_1 \lambda)^{t-\tau-t'} [X^\top]_i \zeta_{\tau+t'-1} z$$

$$= (1 - \eta_1 \lambda)^{t-\tau} [X^\top]_i \widetilde{V}_\tau z + [X^\top]_i \Xi_{t,\tau} z$$

where $\Xi_{t,\tau} = -\sum_{t'=1}^{t-\tau} \eta_1 (1 - \eta_1 \lambda)^{t-\tau-t'} \zeta_{\tau+t'-1}$. Let $Y_1 = [X^\top]_i \widetilde{V}_t z$, $Y_2 = [X^\top]_i \widetilde{V}_\tau z$, $Y_3 = [X^\top]_i \widetilde{V}_\tau \zeta$, $Y_4 = [X^\top]_i \Xi_{t,\tau} z$, $\beta = (1 - \eta_1 \lambda)^{t-\tau} \lesssim 1$, we have $Y_1 = Y_4 + \beta Y_2$.

Consider $Y_1$, given that $[\widetilde{V}_\tau]_{ij} \sim \mathcal{N}(0, \tau_0^2)$, then

$$\mathrm{Var}([X^\top]_i \widetilde{V}_\tau z) = \tau_0^2 \|z\|_2^2 \sum_j X_{ji}^2 = \tau_0^2 \|z\|_2^2 \|[X]_i\|_2^2$$

With Assumption 2, we have $Y_2 \sim \mathcal{N}(0, \tau_0^2 u^2 (u+r)^2)$. Similarly, $Y_1 \sim \mathcal{N}(0, \tau_0^2 u^2 (u+r)^2)$, $Y_3 \sim \mathcal{N}(0, \tau_0^2 r^2 (u+r)^2)$

Consider $Y_4$, denote its variance as $\sigma_{t,\tau}$.

$$\mathrm{Var}([X^\top]_i \widetilde{V}_\tau z) = (1 - \eta_1 \lambda)^{2(t-\tau)} \mathrm{Var}([X^\top]_i \widetilde{V}_\tau z) + \mathrm{Var}([X^\top]_i \Xi_{t,\tau} z)$$

$$\tau_0^2 u^2 (u+r)^2 = (1 - \eta_1 \lambda)^{2(t-\tau)} \tau_0^2 u^2 (u+r)^2 + \sigma_{t,\tau}^2$$

$$\sigma_{t,\tau} = \sqrt{\tau_0^2 u^2 (u+r)^2 \left(1 - (1 - \eta_1 \lambda)^{2(t-\tau)}\right)} \gtrsim \tau_0 u (u+r) \sqrt{\eta_1 \lambda (t-\tau)}$$

Let $\kappa = \tau_0 r (u+r) \sqrt{\log d}$, with Chernoff bound for Gaussian Variable in Lemma 5, and we have Gaussian Integral that $\int_{-\infty}^{\infty} e^{-ax^2} = \sqrt{\frac{\pi}{a}}$, then

$$
\begin{aligned}
\Pr(A \cap \mathcal{F}_\tau^+) &= \Pr[i \in \epsilon_t^{X, z+\zeta}, i \notin \epsilon_t^{x,z}, i \in \mathcal{F}_\tau^+] \\
&= \Pr[Y_2 + Y_3 \geq 0, Y_2 \leq 0, Y_1 \geq \kappa] \\
&= \Pr[Y_2 + Y_3 \geq 0, Y_2 \leq 0, Y_4 \geq \kappa - \beta Y_2] \\
&= \mathbb{E}_{Y_2} [\Pr[Y_3 \geq -Y_2 \mid Y_2, Y_2 \leq 0, Y_4 \geq \kappa - \beta Y_2 \mid Y_2]] \\
&= \mathbb{E}_{Y_2} [\Pr[Y_3 \geq -Y_2 \mid Y_2] \mathbf{1}(Y_2 \leq 0) \Pr[Y_4 \geq \kappa - \beta Y_2 \mid Y_2]] \\
&\lesssim \int_{-\infty}^0 e^{-\frac{z^2}{2\tau_0^2 r^2 (u+r)^2}} e^{-\frac{(\kappa - \beta z)^2}{2\sigma_{t,\tau}^2}} \, dz \lesssim \int_{-\infty}^0 e^{\left(-\frac{1}{2\tau_0^2 r^2 (u+r)^2} - \frac{\beta^2}{2\sigma_{t,\tau}^2}\right) z^2} \, dz \\
&\lesssim \frac{\sqrt{\pi}}{2\sqrt{\frac{1}{2\tau_0^2 r^2 (u+r)^2} + \frac{\beta^2}{2\sigma_{t,\tau}^2}}} \lesssim \tau_0 r (u+r)
\end{aligned}
$$

$$
\begin{aligned}
\Pr(B \cap \mathcal{F}_\tau^+) &= \Pr[i \in \epsilon_t^z, i \notin \epsilon_t^{z-\zeta}, i \in \mathcal{F}_\tau^+] \\
&= \Pr[Y_2 \geq 0, Y_2 - Y_3 \leq 0, Y_1 \geq \kappa] \\
&= \Pr[-Y_2 \geq 0, -Y_2 - Y_3 \leq 0, -Y_1 \geq \kappa] \\
&= \mathbb{E}_{Y_2} [\mathbf{1}(Y_2 \leq 0) \Pr[Y_3 \geq -Y_2 \mid Y_2] \Pr[Y_4 \leq -\kappa - \beta Y_2 \mid Y_2]] \\
&= \mathbb{E}_{Y_2} [\mathbf{1}(Y_2 \leq 0) \Pr[Y_3 \geq -Y_2 \mid Y_2] \Pr[Y_4 \geq \kappa + \beta Y_2 \mid Y_2]] \\
&\lesssim \int_{-\infty}^0 e^{-\frac{z^2}{2\tau_0^2 r^2 (u+r)^2}} e^{-\frac{(\kappa + \beta z)^2}{2\sigma_{t,\tau}^2}} \, dz \lesssim \int_{-\infty}^0 e^{\left(-\frac{1}{2\tau_0^2 r^2 (u+r)^2} - \frac{\beta^2}{2\sigma_{t,\tau}^2}\right) z^2} \, dz \\
&\lesssim \frac{\sqrt{\pi}}{2\sqrt{\frac{1}{2\tau_0^2 r^2 (u+r)^2} + \frac{\beta^2}{2\sigma_{t,\tau}^2}}} \lesssim \tau_0 r (u+r)
\end{aligned}
$$

Using Bernstein inequality in Lemma 2, to bound $|A \cap \mathcal{F}_\tau^+|$ and $|B \cap \mathcal{F}_\tau^+|$. Suppose $M_i = \mathbf{1}(i \in \epsilon_t^{z+\zeta}, i \notin \epsilon_t^z, i \in \mathcal{F}_\tau^+)$ and $N_i = \mathbf{1}(i \in \epsilon_t^z, i \notin \epsilon_t^{z-\zeta}, i \in \mathcal{F}_\tau^+)$.

$$|A \cap \mathcal{F}_\tau^+| = \sum_{i=1}^{L} M_i, |B \cap \mathcal{F}_\tau^+| = \sum_{i=1}^{L} N_i$$

$$\mathbb{E}[|A \cap \mathcal{F}_\tau^+|] = \mathbb{E}[M_i] = \Pr(M_i) \lesssim \tau_0 r(u+r),$$

$$\mathbb{E}[|B \cap \mathcal{F}_\tau^+|] = \mathbb{E}[N_i] = \Pr(N_i) \lesssim \tau_0 r(u+r)$$

Then with high probability at least $1 - \delta$, and let $\delta = \frac{1}{d}$,

$$\sum_{i=1}^{L} M_i \lesssim \sqrt{L \log d} + \log d + \tau_0 r(u+r)L$$

$$\sum_{i=1}^{L} N_i \lesssim \sqrt{L \log d} + \log d + \tau_0 r(u+r)L$$

Finally, for $L = \Theta\left(\text{Poly}(d)\right)$, we conclude that

$$|A \cap \mathcal{F}_\tau^+| \lesssim \tau_0 r(u+r)L$$

$$|B \cap \mathcal{F}_\tau^+| \lesssim \tau_0 r(u+r)L$$

Furthermore, we derive that

$$
\begin{aligned}
\left|\Pr(A \cap \mathcal{F}_\tau^+) - \Pr(B \cap \mathcal{F}_\tau^+)\right| &= \left|\Pr[i \in \epsilon_t^{z+\zeta}, i \notin \epsilon_t^z, i \in \mathcal{F}_\tau^+] - \Pr[i \in \epsilon_t^z, i \notin \epsilon_t^{z-\zeta}, i \in \mathcal{F}_\tau^+]\right| \\
&= \mathbb{E}_{Y_2}\left[\mathbf{1}(Y_2 \leq 0)\Pr[Y_3 \geq -Y_2 \mid Y_2]\Pr[\kappa - \beta Y_2 \leq Y_4 \leq \kappa + \beta Y_2 \mid Y_2]\right] \\
&\lesssim \mathbb{E}_{Y_2}\left[\mathbf{1}(Y_2 \leq 0)e^{-\frac{|Y_2|^2}{2\tau_0^2 r^2(u+r)^2}}\frac{|Y_2|}{\sigma_{t,\tau}}\right] \\
&\lesssim \int_{-\infty}^{0} e^{-\frac{z^2}{2\tau_0^2 r^2(u+r)^2}}\frac{|z|}{\sigma_{t,\tau}}dz \lesssim \frac{1}{\sigma_{t,\tau}}\int_{0}^{\infty} z e^{-\frac{z^2}{2\tau_0^2 r^2(u+r)^2}}dz \\
&\lesssim \frac{\tau_0^2 r^2(u+r)^2}{\sigma_{t,\tau}}\int_{0}^{\infty} e^{-v}dv \\
&\lesssim \frac{\tau_0^2 r^2(u+r)^2}{\sigma_{t,\tau}} \lesssim \frac{\tau_0^2 r^2(u+r)^2}{\tau_0 u(u+r)\sqrt{\eta_1 \lambda(t-\tau)}} \lesssim \frac{\tau_0 r^2(u+r)}{u\sqrt{\eta_1 \lambda(t-\tau)}}
\end{aligned}
$$

Using Bernstein inequality in Lemma 2, to bound $||A \cap \mathcal{F}_\tau^+| - |B \cap \mathcal{F}_\tau^+||$. Suppose $M_i = \mathbf{1}(i \in \epsilon_t^{z+\zeta}, i \notin \epsilon_t^z, i \in \mathcal{F}_\tau^+)$ and $N_i = \mathbf{1}(i \in \epsilon_t^z, i \notin \epsilon_t^{z-\zeta}, i \in \mathcal{F}_\tau^+)$.

$$\left||A \cap \mathcal{F}_\tau^+| - |B \cap \mathcal{F}_\tau^+|\right| = \left|\sum_{i=1}^{L}(M_i - N_i)\right|$$

$$
\begin{aligned}
\left|\mathbb{E}[|A \cap \mathcal{F}_\tau^+| - |B \cap \mathcal{F}_\tau^+|]\right| &= \mathbb{E}\left[M_i - N_i\right] \\
&= |\Pr(M_i) - \Pr(N_i)| \\
&= \left|\Pr[i \in \epsilon_t^{z+\zeta}, i \notin \epsilon_t^z, i \in \mathcal{F}_\tau^+] - \Pr[i \in \epsilon_t^z, i \notin \epsilon_t^{z-\zeta}, i \in \mathcal{F}_\tau^+]\right| \\
&\lesssim \frac{\tau_0 r^2(u+r)}{u\sqrt{\eta_1 \lambda(t-\tau)}}
\end{aligned}
$$

Then with high probability at least $1 - \delta$, and let $\delta = \frac{1}{d}$,

$$\frac{1}{L}\sum_{i=1}^{L}(M_i - N_i) - \mathbb{E}\left[M_i - N_i\right] \leq \sqrt{\frac{2\sigma^2 \log(1/\delta)}{L}} + \frac{2c\log(1/\delta)}{3L}$$

$$\sum_{i=1}^{L}(M_i - N_i) \leq L\sqrt{\frac{2\sigma^2 \log(1/\delta)}{L}} + L\frac{2c\log(1/\delta)}{3L} + \frac{\tau_0 r^2(u+r)L}{u\sqrt{\eta_1\lambda(t-\tau)}}$$

$$\sum_{i=1}^{L}(M_i - N_i) \lesssim \sqrt{L\log d} + \log d + \frac{\tau_0 r^2(u+r)L}{u\sqrt{\eta_1\lambda(t-\tau)}}$$

Finally, for $L = \Theta\left(\mathrm{Poly}(d)\right)$, we get that

$$\left||A \cap \mathcal{F}_\tau^+| - |B \cap \mathcal{F}_\tau^+|\right| \lesssim \frac{\tau_0 r^2(u+r)L}{u\sqrt{\eta_1\lambda(t-\tau)}}$$

**For $|A \cap \mathcal{F}_\tau^-|$, $|B \cap \mathcal{F}_\tau^-|$ and $||A \cap \mathcal{F}_\tau^-| - |B \cap \mathcal{F}_\tau^-||$.** Similar to the above part, we have
$$|A \cap \mathcal{F}_\tau^-| \lesssim \tau_0 r(u+r)L$$
$$|B \cap \mathcal{F}_\tau^-| \lesssim \tau_0 r(u+r)L$$
$$\left||A \cap \mathcal{F}_\tau^-| - |B \cap \mathcal{F}_\tau^-|\right| \lesssim \frac{\tau_0 r^2(u+r)L}{u\sqrt{\eta_1\lambda(t-\tau)}}$$

**For $|A \cap \mathcal{F}_s^c|$ and $|B \cap \mathcal{F}_s^c|$.**
$$\begin{aligned}
\Pr[i \in \epsilon_t^{z+\zeta}, i \notin \epsilon_t^z, i \in \mathcal{F}_s^c] &= \Pr\left[Y_2 + Y_3 \geq 0, Y_2 \leq 0, |Y_1| \leq \kappa\right] \\
&= \mathbb{E}\left[\Pr\left[Y_2 + Y_3 \geq 0, Y_2 \leq 0, |Y_4 - \beta Y_2| \leq \kappa\right]\right] \\
&= \mathbb{E}_{Y_2}\left[\mathbf{1}(Y_2 \leq 0)\Pr\left[Y_3 \geq -Y_2 \mid Y_2\right] \cdot \frac{\kappa}{\sigma_{s,t}}\right] \\
&\lesssim \mathbb{E}_{Y_2}\left[\mathbf{1}(Y_2 \leq 0)e^{-\frac{|Y_2|^2}{2\tau_0^2 r^2(u+r)^2}}\frac{\kappa}{\sigma_{t,\tau}}\right] \\
&\lesssim \frac{\tau_0 r(u+r)\kappa}{\sigma_{t,\tau}}\frac{\sqrt{2\pi}}{2} \lesssim \frac{\tau_0 r(u+r)\tau_0 r(u+r)\sqrt{\log d}}{\tau_0 u(u+r)\sqrt{\eta_1\lambda(t-\tau)}} \\
&\lesssim \frac{\tau_0 r^2(u+r)\sqrt{\log d}}{u\sqrt{\eta_1\lambda(t-\tau)}}
\end{aligned}$$

Similarly, using Bernstein inequality in Lemma 2, $|A \cap \mathcal{F}_s^c| \lesssim \frac{\tau_0 r^2(u+r)L\sqrt{\log d}}{u\sqrt{\eta_1\lambda(t-\tau)}}$, and $|B \cap \mathcal{F}_s^c| \lesssim \frac{\tau_0 r^2(u+r)L\sqrt{\log d}}{u\sqrt{\eta_1\lambda(t-\tau)}}$.

Finally,

$$\left\|\left(\mathbf{1}\left(\epsilon_t^{X,z-\zeta}\right) + \mathbf{1}\left(\epsilon_t^{X,z+\zeta}\right) - 2\mathbf{1}\left(\epsilon_t^{X,z}\right)\right)^\top \Delta Q_\tau\right\|_2$$

$$\leq (u+r)K\left(P\left||A \cap \mathcal{F}_\tau^+| - |B \cap \mathcal{F}_\tau^+|\right| + P\left||A \cap \mathcal{F}_\tau^-| - |B \cap \mathcal{F}_\tau^-|\right| + (1-P)\left(|A \cap \mathcal{F}_\tau^+| + |B \cap \mathcal{F}_\tau^+|\right)\right.$$

$$\left. + (1-P)\left(|A \cap \mathcal{F}_\tau^-| + |B \cap \mathcal{F}_\tau^-|\right) + |A \cap \mathcal{F}_\tau^c| + |B \cap \mathcal{F}_\tau^c|\right)$$

$$\lesssim (u+r)K\left(2P\frac{\tau_0 r^2(u+r)L}{u\sqrt{\eta_1\lambda(t-\tau)}} + (1-2P)\tau_0 r(u+r)L + \frac{\tau_0 r^2(u+r)L\sqrt{\log d}}{u\sqrt{\eta_1\lambda(t-\tau)}}\right)$$

$$\lesssim (u+r)K\frac{\tau_0 r^2(u+r)L\sqrt{\log d}}{u\sqrt{\eta_1\lambda(t-\tau)}}$$

$$\lesssim \frac{\tau_0 r^2(u+r)^2 KL\sqrt{\log d}}{u\sqrt{\eta_1\lambda(t-\tau)}}$$

When $t \le \frac{1}{\eta_1 \lambda}$, we conclude that term $\Phi$ is

$$\left| \left( \mathbb{1}\left(\epsilon_t^{X,z-\zeta}\right) + \mathbb{1}\left(\epsilon_t^{X,z+\zeta}\right) - 2\mathbb{1}\left(\epsilon_t^{X,z}\right)\right)^\top Q_t z \right|$$

$$\le \eta_1 u \sum_{\tau=1}^{t} \left\| \left( \mathbb{1}\left(\epsilon_t^{X,z-\zeta}\right) + \mathbb{1}\left(\epsilon_t^{X,z+\zeta}\right) - 2\mathbb{1}\left(\epsilon_t^{X,z}\right)\right)^\top \Delta Q_{\tau-1} \right\|_2$$

$$\lesssim \eta_1 u \sum_{\tau=1}^{t} \frac{\tau_0 r^2 (u+r)^2 KL\sqrt{\log d}}{u\sqrt{\eta_1 \lambda (t-\tau)}}$$

$$\lesssim \tau_0 r^2 (u+r)^2 KL\sqrt{\log d}\sqrt{\frac{t\eta_1}{\lambda}}$$

$$\lesssim \tau_0 \lambda^{-1} r^2 (u+r)^2 KL\sqrt{\log d}$$

**Combine term $\Psi$ and term $\Phi$.**

$$|\widetilde{g}_t(X, z-\zeta) + \widetilde{g}_t(X, z+\zeta) - 2\widetilde{g}_t(X,z)|$$

$$= 1/L \cdot \left| \mathbb{1}\left(\epsilon_t^{X,z-\zeta}\right)^\top Q_t(z-\zeta) + \mathbb{1}\left(\epsilon_t^{X,z+\zeta}\right)^\top Q_t(z+\zeta) - 2\mathbb{1}\left(\epsilon_t^{X,z}\right)^\top Q_t z \right|$$

$$\le 1/L \cdot \underbrace{\left| \left(\mathbb{1}\left(\epsilon_t^{X,z-\zeta}\right) + \mathbb{1}\left(\epsilon_t^{X,z+\zeta}\right) - 2\mathbb{1}\left(\epsilon_t^{X,z}\right)\right)^\top Q_t z \right|}_{\Phi} + 1/L \cdot \underbrace{\left| \left(\mathbb{1}\left(\epsilon_t^{X,z+\zeta}\right) - \mathbb{1}\left(\epsilon_t^{X,z-\zeta}\right)\right)^\top Q_t \zeta \right|}_{\Psi}$$

$$\lesssim \tau_0 \lambda^{-1} r^2 (u+r)^2 K\sqrt{\log d} + \lambda^{-1} r u^{-1}(u+r)K\sqrt{\log d}$$

with choice of small $u, r, \tau_0 = \mathcal{O}\left(\frac{1}{\sqrt{\log d}}\right), \frac{1}{\lambda} = \mathcal{O}\left(\sqrt{\log d}\right)$ and $L = \Theta\left(\text{Poly}(d)\right)$, therefore, we conclude that

$$|\widetilde{g}_t(X, z-\zeta) + \widetilde{g}_t(X, z+\zeta) - 2\widetilde{g}_t(X,z)| \lesssim \frac{1}{\sqrt{\log d}}\frac{1}{\sqrt{\log d}}\sqrt{\log d} \lesssim \frac{1}{\sqrt{\log d}}$$

**Deal with $|g_{t_1}(X_2)|$.** Assume that $|g_{t_1}(X_2, z-\zeta) + g_{t_1}(X_2, z+\zeta) - 2g_{t_1}(X_2, z)| \lesssim \xi$ and from Theorem 1 we have $\xi = \frac{1}{\sqrt{\log d}}$. We would first like to analysis $|g_{t_1}(X_2, z)|, |g_{t_1}(X_2, z-\zeta)|, |g_{t_1}(X_2, z+\zeta)|$. Naturally, we have

$$g_{t_1}(X_2, z) = \frac{1}{2}\left(g_{t_1}(X_2, z+\zeta) + g_{t_1}(X_2, z-\zeta)\right) + \gamma$$

where $|\gamma| \le \xi$.

Then consider the proportion of $x_{L,2} = \{z-\zeta, z+\zeta, z\}$ in $N$ training sequences with high probability. For $x_{L,2} = \{z-\zeta, z+\zeta\}$, its expected proportion is $\frac{1}{4}$ and for $x_{L,2} = z$, its expected proportion $\frac{1}{2}$. Using Hoeffding's inequality in Lemma 1, for example $x_{L,2} = z - \zeta$, define random variables,

$$X_n = \begin{cases} 1 & if \ X_{L,2}^n = z - \zeta, \\ 0 & else. \end{cases}$$

Since $X_n$ are i.i.d. and $E[X_n] = \frac{1}{4}$,

$$\Pr\left( \left| \frac{1}{N}\sum_{n=1}^{N} X_n - \frac{1}{4} \right| \ge t \right) \le 2\exp\left(-2Nt^2\right)$$

Let $\delta = 2\exp\left(-2Nt^2\right)$, then $t = \sqrt{\frac{\log \frac{2}{\delta}}{2N}}$. If $1 - \delta = 1 - \frac{1}{d}, t = \sqrt{\frac{\log d}{N}}$, then with probability at least $1 - \delta$, the proportion of $x_{L,2} = z - \zeta$ is $\frac{1}{4} + \sqrt{\frac{\log d}{N}}$, Naturally, the proportion of $x_{L,2} = z + \zeta$ is $\frac{1}{4} + \sqrt{\frac{\log d}{N}}$, and the proportion of $x_{L,2} = z$ is $\frac{1}{2} + \sqrt{\frac{\log d}{N}}$.

With the definition of empirical loss, $l$ is the logistic loss, and $l(f(V; \cdot); X_2, Y) = \log\left(1 + e^{-y_L f(V; X_2, Y)}\right)$. Then w.h.p. at least $1 - \delta$,

$$
\begin{aligned}
\widehat{L}(V_{t_1}) =& \frac{1}{N} \sum_{n \in [N]} l(f(V_{t_1}; \cdot); X_2, Y) \\
=& \left( \frac{1}{4} \pm \mathcal{O}\left( \sqrt{\frac{\log d}{N}} \right) \right) l(g_{t_1}(X_2, z + \zeta)) + \left( \frac{1}{4} \pm \mathcal{O}\left( \sqrt{\frac{\log d}{N}} \right) \right) l(g_{t_1}(X_2, z - \zeta)) \\
& + \left( \frac{1}{2} \pm \mathcal{O}\left( \sqrt{\frac{\log d}{N}} \right) \right) l(g_{t_1}(X_2, z)) \\
=& \left( \frac{1}{4} \pm \mathcal{O}\left( \sqrt{\frac{\log d}{N}} \right) \right) \left( l(g_{t_1}(X_2, z + \zeta)) + l(g_{t_1}(X_2, z - \zeta)) + 2l(g_{t_1}(X_2, z)) \right) \\
=& \left( \frac{1}{4} \pm \mathcal{O}\left( \sqrt{\frac{\log d}{N}} \right) \right) \Big( \underbrace{l(g_{t_1}(X_2, z + \zeta)) + l(g_{t_1}(X_2, z - \zeta)) - 2l(g_{t_1}(X_2, z) - \gamma)}_{A} \\
& + \underbrace{2l(g_{t_1}(X_2, z) - \gamma) + 2l(g_{t_1}(X_2, z))}_{B} \Big)
\end{aligned}
$$

For term $A$, since $l$ is convex, then

$$
\begin{aligned}
A =& l(g_{t_1}(X_2, z + \zeta)) + l(g_{t_1}(X_2, z - \zeta)) - 2l(g_{t_1}(X_2, z) - \gamma) \\
=& l(g_{t_1}(X, z + \zeta)) + l(g_{t_1}(X_2, z - \zeta)) - 2l\left( \frac{g_{t_1}(X_2, z + \zeta) + g_{t_1}(X_2, z - \zeta)}{2} \right) \\
\geq& 0
\end{aligned}
$$

Further since $l$ is a 2-Lipschitz function, we have

$$
\begin{aligned}
|l(g_t(X, z)) - l(g_t(X, z) - \gamma)| \leq& 2\gamma \\
B =& 2l(g_{t_1}(X_2, z) - \gamma) + 2l(g_{t_1}(X_2, z)) \\
\geq& 2l(g_{t_1}(X_2, z) - \gamma) + 2l(g_{t_1}(X_2, z) - \gamma) - 4\gamma
\end{aligned}
$$

Finally, from Theorem 1 we have $\xi = \frac{1}{\sqrt{\log d}}$, we have the lower bound of $\widehat{L}(V_{t_1})$,

$$
\begin{aligned}
\widehat{L}(V_{t_1}) =& \left( \frac{1}{4} \pm \mathcal{O}\left( \sqrt{\frac{\log d}{N}} \right) \right) (A + B) \\
\geq& \left( \frac{1}{4} - \mathcal{O}\left( \sqrt{\frac{\log d}{N}} \right) \right) (4 \log 2 - 4\gamma) \\
\geq& \log 2 - \mathcal{O}(\xi) - \mathcal{O}\left( \sqrt{\frac{\log d}{N}} \right) \\
\geq& \log 2 - \mathcal{O}\left( \frac{1}{\sqrt{\log d}} \right) - \mathcal{O}\left( \sqrt{\frac{\log d}{N}} \right)
\end{aligned}
$$

According to the definition of training loss of component $\mathcal{Q}$ on signal weight, i.e. $K^1(\overline{V})$, we have

$$
K^1_{t_1}(\overline{V}_{t_1}) \gtrsim \log 2 - \mathcal{O}\left( \frac{1}{\sqrt{\log d}} \right) - \mathcal{O}\left( \sqrt{\frac{\log d}{N}} \right)
$$

Naturally, assume that $\widehat{L}(V_{t_1}) \leq \log 2 + \mathcal{O}(\xi')$,

$$\widehat{L}(V_{t_1}) \geq \left(\frac{1}{4} - \mathcal{O}\left(\sqrt{\frac{\log d}{N}}\right)\right)(A + 4\log 2 - 4\gamma)$$

$$= \left(\frac{1}{4} - \mathcal{O}\left(\sqrt{\frac{\log d}{N}}\right)\right)(A + 4\log 2 - \mathcal{O}(\xi))$$

$$\widehat{L}(V_{t_1}) \leq \log 2 + \mathcal{O}(\xi')$$

Then,

$$\left(\frac{1}{4} - \mathcal{O}\left(\sqrt{\frac{\log d}{N}}\right)\right)A \leq \log 2 + \mathcal{O}(\xi') - \left(\frac{1}{4} - \mathcal{O}\left(\sqrt{\frac{\log d}{N}}\right)\right)(4\log 2 - \mathcal{O}(\xi))$$

$$\left(\frac{1}{4} - \mathcal{O}\left(\sqrt{\frac{\log d}{N}}\right)\right)A \leq \mathcal{O}(\xi) + \mathcal{O}(\xi')$$

$$A \leq \frac{\mathcal{O}(\xi') + \mathcal{O}(\xi)}{1 - \mathcal{O}\left(\sqrt{\frac{\log d}{N}}\right)}$$

Consider the Taylor expression of $A$, including the 2nd order, and $u = g_{t_1}(X_2, z + \zeta), v = g_{t_1}(X_2, z - \zeta)$

$$\log 2 + \frac{u}{2} + \frac{u^2}{8} + \log 2 + \frac{v}{2} + \frac{v^2}{8} - 2\left(\log 2 + \frac{u+v}{4} + \frac{(u+v)^2}{32}\right)$$

$$= \frac{u^2}{8} + \frac{v^2}{8} - \frac{(u+v)^2}{16} = \frac{(u+v)^2}{16}$$

$$\leq A \leq \frac{\mathcal{O}(\xi') + \mathcal{O}(\xi)}{1 - \mathcal{O}\left(\sqrt{\frac{\log d}{N}}\right)}$$

Finally, we have

$$|g_{t_1}(X, z)|, |g_{t_1}(X, z - \zeta)|, |g_{t_1}(X, z + \zeta)| \leq \mathcal{O}\left(\sqrt{\frac{\xi' + \xi}{1 - \sqrt{\frac{\log d}{N}}}}\right)$$

then we derive

$$|g_{t_1}(X_2, z - \zeta) + g_{t_1}(X_2, z + \zeta) - 2g_{t_1}(X_2, z)| \leq |g_{t_1}(X_2, z - \zeta)| + |g_{t_1}(X_2, z + \zeta)| + 2|g_{t_1}(X_2, z)|$$

$$\lesssim \sqrt{\frac{\xi' + \xi}{1 - \sqrt{\frac{\log d}{N}}}} \lesssim \xi$$

From Theorem 1 we have $\xi = \frac{1}{\sqrt{\log d}}$, thus $\xi' = \frac{1}{\log d}$.

Finally, we conclude that

$$|g_{t_1}(X_2, z)|, |g_{t_1}(X_2, z - \zeta)|, |g_{t_1}(X_2, z + \zeta)| \lesssim \sqrt{\frac{\xi' + \xi}{1 - \sqrt{\frac{\log d}{N}}}} \lesssim \sqrt{(\xi' + \xi)\left(1 + \sqrt{\frac{\log d}{N}}\right)}$$

$$\lesssim \sqrt{\frac{1}{\log d} + \frac{1}{\sqrt{N}\log d} + \frac{1}{\sqrt{\log d}} + \frac{1}{\sqrt{N}}}$$

$$\lesssim \frac{1}{(\log d)^{1/4}}$$

**Deal with** $\|V_{t_1}\|_F$. Through $|g_{t_1}(X_2)|$, we then analysis $\|V_{t_1}\|_F$. With Corollary 7,

$$
\begin{aligned}
|g_{t_1}(X_2)| =& N_{V_{t_1}}(V_{t_1}; X_2, Y) \\
=& N_{V_{t_1}}(\overline{V}_{t_1}; X_2, Y) + N_{V_{t_1}}(\widetilde{V}_{t_1}; X_2, Y) \\
\lesssim& \frac{1}{L} \sum_{i=1}^{L} y_i \mathbb{1}([X_2^\top]_i V_{t_1} x_{L,2}) \cdot \left([X_2^\top]_i \overline{V}_{t_1} x_{L,2}\right) + \epsilon_{V,1} \\
\lesssim& \frac{1}{L} \left\|\mathbb{1}(X_2^\top V_{t_1} x_{L,2})\right\|_1 \max\left([X_2^\top]_i \overline{V}_{t_1} x_{L,2}\right) + \epsilon_{V,1}
\end{aligned}
\tag{20}
$$

For $\left\|\mathbb{1}(X_2^\top V_{t_1} x_{L,2})\right\|_1$, using Corollary 3,

$$
\|\mathbb{1}(X_2^\top V_{t_1} x_{L,2}) - \mathbb{1}(X_2^\top \widetilde{V}_{t_1} x_{L,2})\|_1 \lesssim K^{4/3} \lambda^{-4/3} \tau_0^{-4/3} L^{2/3} \triangleq \epsilon_V
$$

thus further consider $\left\|\mathbb{1}(X_2^\top \widetilde{V}_{t_1} x_{L,2})\right\|_1$,

$$
\left\|\mathbb{1}(X_2^\top \widetilde{V}_{t_1} x_{L,2})\right\|_1 = \sum_{i \in [L]} \mathbb{1}([X_2^\top]_i \widetilde{V}_{t_1} x_{L,2})
$$

where $\mathbb{1}([X_2^\top]_i \widetilde{V}_{t_1} x_{L,2})$ is Bernoulli r.v., then using Hoeffding's inequality in Lemma 1,

$$
\Pr\left(\sum_{i \in [L]} \mathbb{1}([X_1^\top]_i \widetilde{W}_t x_{L,1}) \geq t\right) \leq e^{-\frac{t^2}{2}}
$$

Let $\delta = e^{-\frac{t^2}{2}}$, with $\delta = \frac{1}{d}$, $t = \sqrt{2\log\frac{1}{\delta}} = \sqrt{2\log d}$, then with probability at least $1 - \delta$ (i.e., $1 - \frac{1}{d}$),

$$
\left\|\mathbb{1}(X_2^\top \widetilde{V}_{t_1} x_{L,2})\right\|_1 \lesssim \sqrt{\log d}
$$

Using triangle inequality, we know that

$$
\left\|\mathbb{1}(X_2^\top V_{t_1} x_{L,2})\right\|_1 \lesssim \left\|\mathbb{1}(X_2^\top \widetilde{V}_{t_1} x_{L,2})\right\|_1 + \epsilon_V \lesssim \sqrt{\log d} + \epsilon_V
$$

Substitute into Equation 20, we have

$$
\begin{aligned}
|g_{t_1}(X_2)| \lesssim& \frac{1}{L} \left\|\mathbb{1}(X_2^\top V_{t_1} x_{L,2})\right\|_1 \max\left([X_2^\top]_i \overline{V}_{t_1} x_{L,2}\right) + \epsilon_{V,1} \\
\lesssim& \frac{1}{L}\left(\sqrt{\log d} + \epsilon_V\right)(u+r)^2 \|V_{t_1}\|_F + \epsilon_{V,1} \\
\lesssim& \|V_{t_1}\|_F \left(\sqrt{\log d} + \epsilon_V\right)\frac{(u+r)^2}{L} + \epsilon_{V,1} \\
\lesssim& \|V_{t_1}\|_F \frac{1}{\text{Poly}(d)} + \frac{1}{\text{Poly}(d)}
\end{aligned}
$$

with $|g_{t_1}(X_2)| \lesssim \frac{1}{(\log d)^{1/4}}$, we have

$$
\|\overline{V}_{t_1}\|_F \leq \|V_{t_1}\|_F \lesssim \frac{1}{\text{Poly}(d)}
$$

## F.2    PROOF OF THEOREM 2

**Theorem.** *In the elementary stage with $\eta_1 = \Theta(1)$ and $t \leq t_1 \triangleq \frac{1}{\eta_1 \lambda}$ where $\lambda$ denotes the $L_2$ regularization coefficient. With Assumption 1, $\epsilon_{W,1} = \Theta(1/Poly(d))$, $\epsilon_W = \Theta\left((Poly(d))^{2/3}\right)$ and initial weights $W_0 \to \mathbf{0}_{d \times d}$, it holds that*

*(b.1) For the model parameter $W$ of network $h$, there exists an optimal signal weight $W^\star$, $\overline{W}_{t_1}$ can reach $W^\star$ through gradient descent optimization over $t_1$ iterations, i.e., $\|\overline{W}_{t_1}\|_F$ satisfies*

$$\|\overline{W}_{t_1}\|_F = \Theta\left(d \log(1/\epsilon_{W,1})\right).$$

*(b.2) With random and small noise weight, the training loss of easy-to-fit component $\mathcal{P}$ over signal weight (Definition in Equation 6) at iteration $t_1$ satisfies*

$$K_{t_1}^1(\overline{W}_{t_1}) \lesssim \epsilon_{W,1} + \frac{\sqrt{d}\log d}{L}\epsilon_W + \frac{1}{\sqrt{\log d}}.$$

*Namely, the network $h$ learns the easy-to-fit component $\mathcal{P}$ within $t_1$ iterations.*

**Remark 7** (**Proof Sketch**). *We summarize the proof sketch and main techniques in Proof of Theorem 2. **For (b.1) and (b.2):** **In the beginning**, we first analyze the network $h$'s output under the optimal weight, with signal-noise decomposition, separating it into the outputs under the optimal signal weight and small random noise weights, respectively. The upper bound of the latter relies on the key Proposition 6, 7 and Corollary 6, where the calculation of activations and attention scores is explicitly written out, leveraging the differences in activation patterns. The upper bound analysis of the former utilizes the properties of $W^\star$ and the data construction attributes of component $\mathcal{P}$. **Moving forward**, we use this network output to represent the upper bound of the optimal loss. Furthermore, through gradient descent analysis, we measure $\|\overline{W}_{t_1} - W^\star\|$ and $\|K_{t_1}^1(\overline{W}_{t_1}) - K_{t_1}^1(W^\star)\|$. We use proof by contradiction to give (b.1) and (b.2), showing that there exists a fixed target signal matrix which will classify $\mathcal{P}$ correctly no matter the small noise weight.*

**Proof.** According to Theorem 1, we conclude that the large learning rate creates too much noise to learn $Q$. Also, from above we conclude that in the first stage, the network weight $V_{t_1}$ on $Q$ changes small.

**Definition 4.** *In the elementary stage, denote the optimal weight as $U_1^\star = \begin{bmatrix} W^\star & 0 \\ 0 & \overline{V}_{t_1} = \Delta V \end{bmatrix}$ with initial $W_0 = V_0 \to \mathbf{0}_{d \times d}$, where $W^\star \triangleq d \log(1/\epsilon_{W,1}) w^\star (w^\star)^\top \in \mathbb{R}^{d \times d}$, and $\|\overline{V}_{t_1}\|_F \lesssim \frac{1}{Poly(d)}$.*

**In this section, we primarily focus on the process of optimizing from $W_0$ to $W^\star$.** With the decomposition of signal and noise weight, consider random and small noise, we will prove that $\overline{W}_0$ can be optimized to $\overline{W}_{t_1}$, which is close to $W^\star$, at the end of this section through gradient descent analysis.

Since $f_t$ is the function of signal weight with random noise weight, then we first consider the decomposition of $f_t(W^\star; X_1, Y)$

$$f_t(W^\star; X_1, Y) = N_{W_t}(W^\star + \widetilde{W}_t; X_1, Y)$$
$$= N_{W_t}(W^\star; X_1, Y) + N_{W_t}(\widetilde{W}_t; X_1, Y)$$

**Deal with term $N_{W_t}(\widetilde{W}_t; X_1, Y)$.** With Corollary 6, and choice of small $u, r$, $\tau_0 = \mathcal{O}\left(\frac{1}{\sqrt{\log d}}\right)$, $\frac{1}{\lambda} = \mathcal{O}\left(\sqrt{\log d}\right)$ and $L = \text{Poly}(d)$, then we have

$$N_{W_t}(\widetilde{W}_t; X_1, Y) \lesssim \tau_0 (u + \gamma_0)^2 \sqrt{\frac{d \log d}{L}} \lesssim \frac{1}{\text{Poly}(d)} \triangleq \epsilon_{W,1} \tag{21}$$

**Deal with term $N_{W_t}(W^\star; X_1, Y)$.** For the term $N_{W_t}(W^\star; X_1, Y)$, we know that

$$N_{W_t}(W^\star; X_1, Y) = Y/L \cdot \left(\mathbb{1}(X_1^\top W_t x_{L,1}) \odot (X_1^\top W^\star x_{L,1})\right)$$
$$= \frac{1}{L} \sum_{i=1}^{L} y_i \mathbb{1}([X_1^\top]_i W_t x_{L,1}) \cdot ([X_1^\top]_i W^\star x_{L,1})$$

According to the data structure of $X_1$, assume that $\gamma_0 = 1/\sqrt{d}$, with Definition 4 and Assumption 2 that $(w^\star)^2 = 1$. We find that

$$\|W^\star\|_F^2 = (d \log(1/\epsilon_{W,1}))^2 \|w^\star (w^\star)^\top\|_F^2$$
$$= d^2 \log^2(1/\epsilon_{W,1}) \tag{22}$$

We can derive that,

$$N_{W_t}(W^\star; X_1, Y) = Y/L \cdot \left(\mathbb{1}(X_1^\top W_t x_{L,1}) \odot (X_1^\top W^\star x_{L,1})\right)$$

$$= \frac{1}{L} \sum_{i=1}^{L} y_i \mathbb{1}([X_1^\top]_i W_t x_{L,1}) \cdot ([X_1^\top]_i W^\star x_{L,1})$$

$$\leq \frac{1}{L} \sum_{i=1}^{L} \mathbb{1}([X_1^\top]_i W_t x_{L,1}) \cdot \left(d \log(1/\epsilon_{W,1}) [X_1^\top]_i w^\star (w^\star)^\top x_{L,1}\right)$$

$$= d \log(1/\epsilon_{W,1}) [X_1^\top]_i w^\star (w^\star)^\top x_{L,1} \left\|\mathbb{1}(X_1^\top W_t x_{L,1})\right\|_1 / L$$

For $d \log(1/\epsilon_{W,1}) [X_1^\top]_i w^\star (w^\star)^\top x_{L,1}$, with $\epsilon_{W,1} = \frac{1}{\text{Poly}(d)}$,

$$d \log(1/\epsilon_{W,1}) [X_1^\top]_i w^\star (w^\star)^\top x_{L,1} = d \log(1/\epsilon_{W,1}) \left(\text{sign}(\langle w^\star, e\rangle) \frac{1}{\sqrt{d}} + \langle w^\star, e\rangle\right)^2$$

$$\lesssim d \log(\text{Poly}(d))$$

For $\left\|\mathbb{1}(X_1^\top W_t x_{L,1})\right\|_1$, using Corollary 2,

$$\|\mathbb{1}(X_1^\top W x_{L,1}) - \mathbb{1}(X_1^\top \widetilde{W} x_{L,1})\|_1 \lesssim K^{4/3} \lambda^{-4/3} \tau_0^{-4/3} L^{2/3} \triangleq \epsilon_W$$

thus further consider $\left\|\mathbb{1}(X_1^\top \widetilde{W}_t x_{L,1})\right\|_1$,

$$\left\|\mathbb{1}(X_1^\top \widetilde{W}_t x_{L,1})\right\|_1 = \sum_{i \in [L]} \mathbb{1}([X_1^\top]_i \widetilde{W}_t x_{L,1})$$

where $\mathbb{1}([X_1^\top]_i \widetilde{W}_t x_{L,1})$ is Bernoulli r.v., then using Hoeffding's inequality in Lemma 1,

$$\Pr\left(\sum_{i \in [L]} \mathbb{1}([X_1^\top]_i \widetilde{W}_t x_{L,1}) \geq t\right) \leq e^{-\frac{t^2}{2}}$$

Let $\delta = e^{-\frac{t^2}{2}}$, with $\delta = \frac{1}{d}$, $t = \sqrt{2 \log \frac{1}{\delta}} = \sqrt{2 \log d}$, then with probability at least $1 - \delta$ (i.e., $1 - \frac{1}{d}$),

$$\left\|\mathbb{1}(X_1^\top \widetilde{W}_t x_{L,1})\right\|_1 \lesssim \sqrt{\log d}$$

Using triangle inequality, we know that

$$\left\|\mathbb{1}(X_1^\top W_t x_{L,1})\right\|_1 \lesssim \|\mathbb{1}(X^\top \widetilde{W}_t x_L)\|_1 + \epsilon_W \lesssim \sqrt{\log d} + \epsilon_W$$

Finally,

$$N_{W_t}(W^\star; X_1, Y) = d \log(1/\epsilon_{W,1}) [X_1^\top]_i w^\star (w^\star)^\top x_{L,1} \left\|\mathbb{1}(X_1^\top W_t x_{L,1})\right\|_1 / L$$

$$\leq d \log(\text{Poly}(d)) \left(\sqrt{\log d} + \epsilon_W\right) \cdot 1/L$$

$$\leq d \log d \left(\sqrt{\log d} + \epsilon_W\right) \cdot 1/L$$

$$\lesssim \frac{1}{L} \left(d \log d \sqrt{\log d} + \epsilon_W \sqrt{d} \log d\right)$$

where $\epsilon_W = (\text{Poly}(d))^{2/3} \gg \sqrt{\log d}$, then

$$N_{W_t}(W^\star; X_1, Y) \lesssim \frac{\sqrt{d} \log d}{L} \epsilon_W \tag{23}$$

**Combine Equation 21 and Equation 23.** Combine Equation 21 and Equation 23, we have

$$
\begin{aligned}
f_t^1(W^\star; X_1, Y) &= N_{W_t}(W^\star + \widetilde{W}_t; X_1, Y) \\
&= N_{W_t}(W^\star; X_1, Y) + N_{W_t}(\widetilde{W}_t; X_1, Y) \\
&\lesssim \tau_0(u + \gamma_0)^2 \sqrt{\frac{d \log d}{L}} + K^{4/3} \lambda^{-4/3} \tau_0^{-4/3} L^{2/3} L^{-1} \sqrt{d} \log(\mathrm{Poly}(d)) \\
&\lesssim \epsilon_{W,1} + \frac{\sqrt{d} \log d}{L} \epsilon_W
\end{aligned}
$$

with choice of small $u, r, \tau_0 = \mathcal{O}\left(\frac{1}{\sqrt{\log d}}\right), \frac{1}{\lambda} = \mathcal{O}\left(\sqrt{\log d}\right)$ and $L = \mathrm{Poly}(d)$, we have

$$
\begin{aligned}
f_t^1(W^\star; X_1, Y) &\lesssim \epsilon_{W,1} + \frac{\sqrt{d} \log d}{L} \epsilon_W \\
&\lesssim \frac{1}{\mathrm{Poly}(d)} + \frac{\sqrt{d} \log d}{\mathrm{Poly}(d)}
\end{aligned}
$$

Then consider the loss with signal weight $\overline{W}_t = W^\star$ and random noise weight $\widetilde{W}_t$ at time $t$,

$$
\begin{aligned}
&K_t^1(W^\star) \\
=& \frac{1}{N} \sum_{n=1}^N l\left(f_t^1(W^\star; X_1^n, Y^n)\right) \\
\lesssim& \max\left\{\log\left(1 + \exp(-(\epsilon_{W,1} + \sqrt{d}/L \log d \epsilon_W))\right), \log\left(1 + \exp(\epsilon_{W,1} + \sqrt{d}/L \log d \epsilon_W)\right)\right\} \\
\approx& \max\left\{\log 2 - \frac{1}{2}\left(\epsilon_{W,1} + \frac{\sqrt{d} \log d}{L} \epsilon_W\right), \log 2 + \frac{1}{2}\left(\epsilon_{W,1} + \frac{\sqrt{d} \log d}{L} \epsilon_W\right)\right\} \\
\lesssim& \epsilon_{W,1} + \frac{\sqrt{d} \log d}{L} \epsilon_W
\end{aligned}
$$

**Deal with gradient descent to find $W^\star$.** Consider the graident descent of signal $\overline{W}$,

$$
\begin{aligned}
\overline{W}_{t+1} &= \overline{W}_t - \eta_1 \nabla K_t(\overline{W}_t) - \eta_1 \lambda W_t \\
&= (1 - \eta_1 \lambda)\overline{W}_t - \eta_1 \nabla K_t(\overline{W}_t)
\end{aligned}
$$

With $\|W^*\|_F = d \log(1/\epsilon_{W,1}) \triangleq B$ from Equation 22, loss $K_t$ is K-Lipschitz, i.e. $\|\nabla K_t(\overline{W}_t)\|_F \leq K$, assume that $\|\overline{W}_t - W^\star\|_F \leq R \ll B$, then we can measure the distance of $W_t$ and $W^\star$.

$$
\begin{aligned}
\left\|\overline{W}_{t+1} - W^\star\right\|_2^2 &= \left\|(1 - \eta_1 \lambda)\overline{W}_t - \eta_1 \nabla K_t - W^\star\right\|_2^2 \\
&= \left\|(1 - \eta_1 \lambda)(\overline{W}_t - W^\star) - \eta_1(\lambda W^\star + \nabla K_t)\right\|_2^2 \\
&= \left\|(1 - \eta_1 \lambda)(\overline{W}_t - W^\star)\right\|_2^2 + \eta_1^2 \|\lambda W^\star + \nabla K_t\|_2^2 - 2\eta_1(1 - \eta_1 \lambda)\langle \overline{W}_t - W^\star, \lambda W^\star\rangle \\
&\quad - 2\eta_1(1 - \eta_1 \lambda)\langle \overline{W}_t - W^\star, \nabla K_t\rangle \\
&= \left\|(1 - \eta_1 \lambda)(\overline{W}_t - W^\star)\right\|_2^2 + \eta_1^2 \|(\lambda W^\star + \nabla K_t)\|_2^2 - 2\eta\lambda(1 - \eta_1 \lambda)\langle \overline{W}_t, W^\star\rangle \\
&\quad + 2\eta_1 \lambda(1 - \eta_1 \lambda)\langle W^\star, W^\star\rangle - 2\eta_1(1 - \eta_1 \lambda)(K_t(\overline{W}_t) - K_t(W^\star)) \\
&\leq \left\|(1 - \eta_1 \lambda)(\overline{W}_t - W^\star)\right\|_2^2 + 2\eta_1^2(\lambda^2 B^2 + K^2) - 2\eta_1 \lambda(1 - \eta_1 \lambda)(R + B)B \\
&\quad + 2\eta_1 \lambda(1 - \eta_1 \lambda)B^2 - 2\eta_1(1 - \eta_1 \lambda)(K_t(\overline{W}_t) - K_t(W^\star)) \\
&\leq \left\|(1 - \eta\lambda)(\overline{W}_t - W^\star)\right\|_2^2 + 2\eta_1^2(\lambda^2 B^2 + K^2) - 2\eta_1 \lambda(1 - \eta_1 \lambda)RB \\
&\quad - 2\eta_1(1 - \eta_1 \lambda)(K_t(\overline{W}_t) - K_t(W^\star))
\end{aligned}
$$

For the sake of contradiction, assume that $K_t^1(\overline{W}_t) - K_t^1(W^\star) \geq C$, let $0 < 1 - \eta_1\lambda < 1$, and $\eta_1 \ll \frac{\lambda BR + C}{\lambda^2 B^2 + \lambda^2 BR + K^2 + \lambda C}$, and $\lambda R^2 \ll C$,

$$\left\|\overline{W}_{t+1} - W^\star\right\|_2^2 \leq \left\|(\overline{W}_t - W^\star)\right\|_2^2 + 2\eta_1^2(\lambda^2 B^2 + \lambda^2 BR + K^2 + \lambda C) - 2\eta_1(\lambda BR + C)$$
$$\leq \left\|(\overline{W}_t - W^\star)\right\|_2^2 - 2\eta_1(\lambda BR + C)$$
$$\leq \left\|(\overline{W}_t - W^\star)\right\|_2^2 - 2\eta_1 C$$

Thus, in the elementary stage with $t_1$ iterations, $t \leq t_1 \triangleq \frac{1}{\eta_1\lambda}$,

$$\left\|\overline{W}_{t_1} - W^\star\right\|_2^2 \leq \left\|(\overline{W}_0 - W^\star)\right\|_2^2 - 2t_1\eta_1 C \leq R^2 - 2t_1\eta_1 C < 0$$

which is a contradiction, i.e., $K_{t_1}^1(\overline{W}_{t_1}) - K_{t_1}^1(W^\star) \leq C$.

Therefore, in the elementary stage within $t_1$ iterations, $t_1 \leq \frac{1}{\eta_1\lambda}$, through gradient descent optimization, $\|\overline{W}_{t_1}\|_F$ satisfies $\|\overline{W}_{t_1}\|_F \leq B + R$, then

$$\|\overline{W}_{t_1}\|_F = \Theta\left(d\log(1/\epsilon_{W,1})\right)$$

and the training loss satisfies

$$K_{t_1}^1(\overline{W}_{t_1}) \leq K_{t_1}^1(W^\star) + C \lesssim \epsilon_{W,1} + \frac{\sqrt{d}\log d}{L}\epsilon_W + \frac{1}{\sqrt{\log d}}$$

## G  Proof for the Specialized Stage

### G.1  Proof of Theorem 3

**Theorem.** *In the specialized stage with annealing learning rate $\eta_2 = \eta_1 \lambda^2 \epsilon_{V,1}^2 r$ and $t_1 \leq t \leq t_1 + t_2$, where $\epsilon_{V,1} = \Theta(1/Poly(d))$, $t_1 \triangleq \frac{1}{\eta_1 \lambda}$, $t_2 \triangleq \frac{\log^2(1/\epsilon_{V,1})}{\eta_2 \lambda \epsilon_{V,1}^2}$, $\lambda$ denotes the $L_2$ regularization coefficient and data noise $\|\zeta\|_2 = r$ (See Paragraph 3.1). With Assumption 1, it holds that*

*(c.1) For the model parameter $V$ of network $g$, there exists an optimal signal weight $\overline{V}_{t_1} + V^\star$, $\overline{V}_{t_1+t_2}$ can reach $\overline{V}_{t_1} + V^\star$ through gradient descent optimization over $t_2$ iterations, i.e., $\|\overline{V}_{t_1+t_2}\|_F$ satisfies*

$$\|\overline{V}_{t_1+t_2}\|_F = \Theta\left(\frac{\log(1/\epsilon_{V,1})}{\epsilon_{V,1}} + \frac{1}{Poly(d)}\right).$$

*(c.2) With random and small noise weight, the training loss of hard-to-fit component $\mathcal{Q}$ over signal weight (Definition in Equation 6) satisfies*

$$K_{t_1+t_2}^2(\overline{V}_{t_1+t_2}) \lesssim \epsilon_{V,1} + \frac{1}{(\log d)^{1/4}} + \frac{1}{\sqrt{\log d}}.$$

*Namely, the network $g$ learns hard-to-fit component $\mathcal{Q}$ within $t_2$ iterations.*

**Remark 8** (**Proof Sketch**). *We summarize the proof sketch and main techniques in Proof of Theorem 3. **To begin with**, we explore the properties of optimal weight $\overline{V}_{t_1} + V^\star$ and analyze the network $g$'s output under the optimal weight at timepoint $t_1 + t_2$. Using triangle inequality, we need to handle three parts $A, B, C$ separately. Part $A$ exploits the characteristics of $V^\star$ in detail. Part $B$ uses the key Lemma 7 and Corollary 5 to analyze the relationship between the network output at time $t_1 + t_2$ and at time $t_1$, taking into account the signal weight update formula. Part $C$ utilizes the properties of the network output at time $t_1$ to facilitate the analysis. **Thereafter**, we use this network output to represent the upper bound of the optimal loss. Furthermore, through gradient descent analysis, we measure $\|\overline{V}_{t_1+t_2} - (\overline{V}_{t_1} + V^\star)\|$ and $\|K_{t_1+t_2}^2(\overline{V}_{t_1+t_2}) - K_{t_1+t_2}^2(\overline{V}_{t_1} + V^\star)\|$. We use proof by contradiction to give (a) and (b), showing that there exists a fixed target signal matrix which will classify $\mathcal{Q}$ correctly no matter the small noise weight.*

**Proof.**

**Definition 5.** *For time $t_1$, input $X \in \mathbb{R}^{d \times L}$ with query $x_L = z - \zeta, z, z + \zeta \in \mathbb{R}^d$, define*

$$\mathcal{H}_1 \triangleq \{i \in [L] \mid [X^\top]_i V_{t_1}(z - \zeta) \geq 0, [X^\top]_i V_{t_1} z \geq 0, [X^\top]_i V_{t_1}(z + \zeta) < 0\}$$

$$\mathcal{H}_2 \triangleq \{i \in [L] \mid [X^\top]_i V_{t_1}(z - \zeta) \geq 0, [X^\top]_i V_{t_1} z < 0, [X^\top]_i V_{t_1}(z + \zeta) < 0\}$$

$$\mathcal{H}_3 \triangleq \{i \in [L] \mid [X^\top]_i V_{t_1}(z - \zeta) < 0, [X^\top]_i V_{t_1} z < 0, [X^\top]_i V_{t_1}(z + \zeta) \geq 0\}$$

$$\mathcal{H}_4 \triangleq \{i \in [L] \mid [X^\top]_i V_{t_1}(z - \zeta) < 0, [X^\top]_i V_{t_1} z \geq 0, [X^\top]_i V_{t_1}(z + \zeta) \geq 0\}$$

*Similar to Definition 1, note that $X$ aligns with $X_2$ and $x_L$ aligns with $x_{L,2}$.*

We first try to analyze the probability of $i \in \mathcal{H}_i$. With Assumption 2, we can compute the cosine of $z - \zeta$ and $z$,

$$\cos\theta = \frac{\langle z - \zeta, z\rangle}{\|z\|_2 \|z - \zeta\|_2} = \frac{u^2 - \langle \zeta, z\rangle}{u\sqrt{u^2 - 2\langle \zeta, z\rangle + r^2}} = \frac{u^2 - ur\cos\theta_0}{u\sqrt{u^2 + r^2 - 2ur\cos\theta_0}}$$

$$\sin\theta = \sqrt{1 - \cos^2\theta} = \frac{r\sin\theta_0}{\sqrt{u^2 + r^2 - 2ur\cos\theta_0}}$$

For small $r$, with Taylor expansion of $\arcsin\theta$, we have that the angle of $z - \zeta$ and $z$ is $\theta = \frac{r}{u} + \mathcal{O}(r^2)$.

For $\mathcal{H}_1$, when $[X^\top]_i \widetilde{V}_{t_1}$ fall into the middle of $z - \zeta$ and $z$, as well as not in the positive half space of $z + \zeta$, its probability is approximately the proportion of the spherical surface area corresponding

to the angle $\frac{r}{u} + \mathcal{O}(r^2)$. Using Hoeffding's inequality in Lemma 1 and further consider Corollary 3, let $X_i = \mathbf{1}\{i \in \mathcal{H}_1\}$, then $|\mathcal{H}_1| = \sum_{i=1}^{L} X_i$

$$\mathbb{E}\left[|\mathcal{H}_1|\right] = L \cdot \Pr(i \in \mathcal{H}_1) \approx L \cdot \frac{r}{2\pi u} + \epsilon_V$$

Then, let $\delta = 2\exp\left(-\frac{2t^2}{L}\right)$, $t = \sqrt{\frac{1}{2}L\log\frac{2}{\delta}}$, and $1 - \delta = 1 - \frac{1}{d}$, then with probability at least $1 - \delta$,

$$\left||\mathcal{H}_1| - \mathbb{E}[|\mathcal{H}_1|]\right| \leq \sqrt{\frac{1}{2}L\log\frac{2}{\delta}} \lesssim \sqrt{L\log d}$$

$$|\mathcal{H}_1| \lesssim \frac{rL}{2\pi u} + \epsilon_V + \sqrt{L\log d}$$

Similarly, we have

$$|\mathcal{H}_1|, |\mathcal{H}_2|, |\mathcal{H}_3|, |\mathcal{H}_4| \lesssim \frac{rL}{2\pi u} + \epsilon_V + \sqrt{L\log d}$$

**Definition 6.** *In the second stage, denote the optimal weight as* $U_2^\star = \begin{bmatrix} \overline{W}_{t_1} + \Delta W & 0 \\ 0 & \overline{V}_{t_1} + V^\star \end{bmatrix} = \begin{bmatrix} \overline{W}_{t_1+t} & 0 \\ 0 & \overline{V}_{t_1} + V^\star \end{bmatrix}$, $\|\overline{W}_{t_1+t}\|_F \lesssim d\log(1/\epsilon_{W,1})$, *and* $V^\star \in \mathbb{R}^{d \times d}$ *satisfies*

$$[X_2^\top V^*]_i = \begin{cases} \frac{\log(1/\epsilon_{V,1})}{r\epsilon_{V,1}} z^\top & \text{if } i \in \mathcal{H}_1; \\ -\frac{2\log(1/\epsilon_{V,1})}{r\epsilon_{V,1}} z^\top & \text{if } i \in \mathcal{H}_2; \\ \frac{\log(1/\epsilon_{V,1})}{r\epsilon_{V,1}} z^\top & \text{if } i \in \mathcal{H}_3; \\ -\frac{2\log(1/\epsilon_{V,1})}{r\epsilon_{V,1}} z^\top & \text{if } i \in \mathcal{H}_4; \\ 0 & \text{otherwise.} \end{cases} \tag{24}$$

We have that $\|\overline{W}_{t_1+t} - \overline{W}_{t_1}\|_F \ll \|\overline{V}_{t_1} + V^\star - V_{t_1}\| = \|V^\star\|_F$, and we still have $\|\overline{W}_{t_1+t}\|_F \lesssim d\log(1/\epsilon_{W,1})$ from Theorem 2. In this section, we primarily focus on the process of optimizing from $\overline{V}_{t_1}$ to $\overline{V}_{t_1} + V^\star$. To calculate the Frobenius norm $\|V^*\|_F$,

$$\|X_2^\top V^*\|_2^2$$

$$= \sum_{i \in \mathcal{H}_1} \left(\frac{\log(1/\epsilon_{V,1})}{r\epsilon_{V,1}}\right)^2 \|z^\top\|_2^2 + \sum_{i \in \mathcal{H}_2} \left(-\frac{2\log(1/\epsilon_{V,1})}{r\epsilon_{V,1}}\right)^2 \|z^\top\|_2^2 + \sum_{i \in \mathcal{H}_3} \left(\frac{\log(1/\epsilon_{V,1})}{r\epsilon_{V,1}}\right)^2 \|z^\top\|_2^2$$

$$+ \sum_{i \in \mathcal{H}_4} \left(-\frac{2\log(1/\epsilon_{V,1})}{r\epsilon_{V,1}}\right)^2 \|z^\top\|_2^2$$

$$\lesssim u^2 |\mathcal{H}| \left(\frac{\log(1/\epsilon_{V,1})}{r\epsilon_{V,1}}\right)^2 \lesssim u^2 \left(\frac{rL}{2\pi u} + \epsilon_V + \sqrt{L\log d}\right) \frac{\log^2(1/\epsilon_{V,1})}{r^2\epsilon_{V,1}^2}$$

$$\lesssim \frac{uL\log^2(1/\epsilon_{V,1})}{r\epsilon_{V,1}^2}$$

and then $\|V^\star\|_F = \mathcal{O}\left(\frac{\log(1/\epsilon_{V,1})}{\epsilon_{V,1}}\right)$, where $c$ is a constant.

In the following, we focus on the empirical loss with optimal weight $\overline{V}_{t_1} + V^\star$.

$$K_{t_1+t}^2(\overline{V}_{t_1} + V^\star) = \widehat{L}(N_{V_{t_1+t}}(\overline{V}_{t_1} + V^\star; X_2, Y))$$

$$= \frac{1}{N} \sum_{n \in [N]} \log\left(1 + \exp\left(-y_L^n N_{V_{t_1+t}}(\overline{V}_{t_1} + V^\star; X_2^n, Y^n)\right)\right)$$

and then consider $yN_{V_{t_1+t}}(\overline{V}_{t_1} + V^*; X_2, Y)$,

$$yN_{V_{t_1+t}}(\overline{V}_{t_1} + V^*; X_2, Y)$$
$$\geq yN_{V_{t_1+t}}(V^*; X_2, Y) - yN_{V_{t_1+t}}(\overline{V}_{t_1}; X_2, Y)$$
$$\geq \underbrace{yN_{V_{t_1}}(V^\star; X_2, Y)}_{A} - \underbrace{\left| yN_{V_{t_1+t}}(V^\star; X_2, Y) - yN_{V_{t_1}}(V^\star; X_2, Y) \right|}_{B} - \underbrace{yN_{V_{t_1+t}}(\overline{V}_{t_1}; X_2, Y)}_{C}$$

**Deal with term $A$.** We have

$$yN_{V_{t_1}}(V^*; X_2, Y) = y \cdot Y/L \cdot \left( \mathbb{1}\left( X_2^\top V_{t_1} x_{L,2} \right) \odot \left( X_2^\top V^* x_{L,2} \right) \right)$$

$$= \frac{1}{L} \sum_{i=1}^{L} \left( \mathbb{1}\left( [X_2^\top]_i V_{t_1} x_{L,2} \right) \odot \left( [X_2^\top]_i V^* x_{L,2} \right) \right)$$

For $x_{L,2} = z - \zeta$, we have that

$$N_{V_{t_1}}(V^*; X_2, Y, x_{L,2} = z - \zeta) \leq \frac{|\mathcal{H}_1|}{L} \frac{\log(1/\epsilon_{V,1})}{r\epsilon_{V,1}} z^\top (z - \zeta) - \frac{|\mathcal{H}_2|}{L} \frac{2\log(1/\epsilon_{V,1})}{r\epsilon_{V,1}} z^\top (z - \zeta)$$

$$\lesssim -\frac{\log(1/\epsilon_{V,1})u(u+r)}{r\epsilon_{V,1}} \left( \frac{r}{2\pi u} + \frac{\epsilon_V}{L} + \sqrt{\frac{\log d}{L}} \right)$$

$$\lesssim -\frac{\log(1/\epsilon_{V,1})(u+r)}{\epsilon_{V,1}}$$

and for $x_{L,2} = z + \zeta$, we have that

$$N_{V_{t_1}}(V^*; X_2, Y, x_{L,2} = z + \zeta) \leq \frac{|\mathcal{H}_3|}{L} \frac{\log(1/\epsilon_{V,1})}{r\epsilon_{V,1}} z^\top (z + \zeta) - \frac{|\mathcal{H}_4|}{L} \frac{2\log(1/\epsilon_{V,1})}{r\epsilon_{V,1}} z^\top (z + \zeta)$$

$$\lesssim -\frac{\log(1/\epsilon_{V,1})u(u+r)}{r\epsilon_{V,1}} \left( \frac{r}{2\pi u} + \frac{\epsilon_V}{L} + \sqrt{\frac{\log d}{L}} \right)$$

$$\lesssim -\frac{\log(1/\epsilon_{V,1})(u+r)}{\epsilon_{V,1}}$$

and for $x_{L,2} = z$, we have that

$$N_{V_{t_1}}(V^*; X_2, Y, x_{L,2} = z + \zeta) \leq \frac{|\mathcal{H}_1|}{L} \frac{\log(1/\epsilon_{V,1})}{r\epsilon_{V,1}} z^\top z - \frac{|\mathcal{H}_4|}{L} \frac{2\log(1/\epsilon_{V,1})}{r\epsilon_{V,1}} z^\top z$$

$$\lesssim -\frac{\log(1/\epsilon_{V,1})u^2}{r\epsilon_{V,1}} \left( \frac{r}{2\pi u} + \frac{\epsilon_V}{L} + \sqrt{\frac{\log d}{L}} \right)$$

$$\lesssim -\frac{\log(1/\epsilon_{V,1})u}{\epsilon_{V,1}}$$

Finally, with small $r \ll u$, for $x_{L,2} \in \{z - \zeta, z, z + \zeta\}$, we have

$$yN_{V_{t_1}}(V^*; X_2, Y) \gtrsim \frac{u\log(1/\epsilon_{V,1})}{\epsilon_{V,1}}$$

**Deal with term $B$.** With the definition of $\|X_2^\top V^*\|_F^2$, and $\|X_2^\top V^*\|_2^2 \lesssim \frac{uL\log^2(1/\epsilon_{V,1})}{r\epsilon_{V,1}^2}$, we derive that

$$\left| [X_2^\top V^*]_i \right| \lesssim \frac{\log(1/\epsilon_{V,1})}{\epsilon_{V,1}} \sqrt{\frac{u}{r}}$$

$$\left| [X_2^\top V^*]_i x_{L,2} \right| \lesssim \frac{\log(1/\epsilon_{V,1})}{\epsilon_{V,1}} \sqrt{\frac{u(u+r)^2}{r}}$$

With Corollary 5,

$$\left| yN_{V_{t_1+t}}(V^\star; X_2, Y) - yN_{V_{t_1}}(V^\star; X_2, Y) \right| \lesssim \left( \epsilon_V + L\sqrt{\frac{\eta_2}{\eta_1}} + \sqrt{L \log d} \right) \frac{\log(1/\epsilon_{V,1})}{\epsilon_{V,1} L} \sqrt{\frac{u(u+r)^2}{r}}$$

$$\lesssim \frac{\log(1/\epsilon_{V,1})}{\epsilon_{V,1} L\sqrt{r}} L \cdot \left( \lambda \epsilon_{V,1} \sqrt{r} \right)$$

where the last step satisfies when with choice of small $u, r$, $\tau_0 = \mathcal{O}\left(\frac{1}{\sqrt{\log d}}\right), \frac{1}{\lambda} = \mathcal{O}\left(\sqrt{\log d}\right)$ and $L = \Theta\left(\text{Poly}(d)\right)$, and $\eta_2 = \eta_1 \lambda^2 \epsilon_{V,1}^2 r$. Finally,

$$\left| yN_{V_{t_1+t}}(V^\star; X_2, Y) - yN_{V_{t_1}}(V^\star; X_2, Y) \right| \lesssim \lambda \log(1/\epsilon_{V,1})$$

**Deal with term $C$.**   Before, we have

$$|g_{t_1}(X, z)|, |g_{t_1}(X, z-\zeta)|, |g_{t_1}(X, z+\zeta)| \lesssim \mathcal{O}\left( \sqrt{\frac{\zeta' + \zeta}{1 - \sqrt{\frac{\log d}{N}}}} \right) \lesssim \frac{1}{(\log d)^{1/4}}$$

Then, combine with Corollary 7,

$$|N_{V_{t_1}}(\overline{V}_{t_1}; X_2, Y)| \le |g_{t_1}(X_2)| + |N_{V_{t_1}}(\overline{V}_{t_1}; X_2, Y) - N_{V_{t_1}}(V_{t_1}; X_2, Y)|$$

$$\le |g_{t_1}(X_2)| + |N_{V_{t_1}}(\widetilde{V}_{t_1}; X_2, Y)|$$

$$\lesssim \frac{1}{(\log d)^{1/4}} + \epsilon_{V,1}$$

With Corollary 5 and $\|\overline{V}_{t_1}\| \lesssim \frac{1}{\text{Poly(d)}}$

$$\left| yN_{V_{t_1+t}}(\overline{V}_{t_1}; X_2, Y) - yN_{V_{t_1}}(\overline{V}_{t_1}; X_2, Y) \right| \lesssim \left( \epsilon_V + L\sqrt{\frac{\eta_2}{\eta_1}} + \sqrt{L \log d} \right) \frac{1}{L \cdot \text{Poly(d)}}$$

$$\lesssim \frac{1}{\text{Poly(d)}} \sqrt{\frac{\eta_2}{\eta_1}}$$

Finally, we get

$$\left| yN_{V_{t_1+t}}(\overline{V}_{t_1}; X_2, Y) \right| \lesssim \frac{1}{(\log d)^{1/4}} + \epsilon_{V,1} + \frac{1}{\text{Poly(d)}} \sqrt{\frac{\eta_2}{\eta_1}}$$

$$\lesssim \frac{1}{(\log d)^{1/4}} + \epsilon_{V,1} + \frac{\lambda \epsilon_{V,1}}{\sqrt{\log d}}$$

$$\lesssim \frac{1}{(\log d)^{1/4}} + \epsilon_{V,1}$$

when with choice of $\eta_2 = \eta_1 \lambda^2 \epsilon_{V,1}^2 r$, $\frac{1}{\lambda} = \mathcal{O}\left(\sqrt{\log d}\right)$.

**Combine term $A, B$ and $C$.**

$$yN_{V_{t_1+t}}(\overline{V}_{t_1} + V^*; X_2, Y)$$

$$\ge \underbrace{yN_{V_{t_1}}(V^\star; X_2, Y)}_{A} - \underbrace{\left| yN_{V_{t_1+t}}(V^\star; X_2, Y) - yN_{V_{t_1}}(V^\star; X_2, Y) \right|}_{B} - \underbrace{\left| yN_{V_{t_1+t}}(\overline{V}_{t_1}; X_2, Y) \right|}_{C}$$

$$\gtrsim \frac{u \log(1/\epsilon_{V,1})}{\epsilon_{V,1}} - \lambda \log(1/\epsilon_{V,1}) - \frac{1}{(\log d)^{1/4}} - \epsilon_{V,1}$$

Finally, when $u \approx \epsilon_{V,1} \lambda$,

$$
\begin{aligned}
K_{t_1+t}^2(\overline{V}_{t_1} + V^\star) &= \widehat{L}(N_{V_{t_1+t}}(\overline{V}_{t_1} + V^\star; X_2, Y)) \\
&= \frac{1}{N} \sum_{n \in [N]} \log\left(1 + \exp\left(-y_L^n N_{V_{t_1+t}}(\overline{V}_{t_1} + V^\star; X_2^n, Y^n)\right)\right) \\
&\lesssim \log\left(1 + \exp\left(-\frac{u \log(1/\epsilon_{V,1})}{\epsilon_{V,1}} + \lambda \log(1/\epsilon_{V,1}) + \frac{1}{(\log d)^{1/4}} + \epsilon_{V,1}\right)\right) \\
&\lesssim \frac{1}{(\log d)^{1/4}} + \epsilon_{V,1}
\end{aligned}
$$

where $\epsilon_{V,1} = \tau_0(u+r)^2 \sqrt{\frac{d \log d}{L}}$.

**Deal with gradient descent to find $\overline{V}_{t_1} + V^\star$.** Consider the graident descent of signal $\overline{V}$,

$$
\begin{aligned}
\overline{V}_{t+1} &= \overline{V}_t - \eta_1 \nabla K_t(\overline{V}_t) - \eta_1 \lambda V_t \\
&= (1 - \eta_1 \lambda)\overline{V}_t - \eta_1 \nabla K_t(\overline{V}_t)
\end{aligned}
$$

Similar to gradient descent of $\overline{W}$, let $\overline{V}_{t_1} + V^\star$ be $W^\star$, then $\left\|\overline{V}_{t_1} + V^\star\right\|_F = \Theta\left(\frac{\log(1/\epsilon_{V,1})}{\epsilon_{V,1}} + \frac{1}{\text{Poly}(d)}\right) \triangleq B$. Let $\left\|\overline{V}_t - (\overline{V}_{t_1} + V^\star)\right\|_F \leq R \ll B$.

$$
\begin{aligned}
&\left\|\overline{V}_{t+1} - (\overline{V}_{t_1} + V^\star)\right\|_2^2 \\
&= \left\|(1 - \eta_2 \lambda)\overline{V}_t - \eta_2 \nabla K_t - (\overline{V}_{t_1} + V^\star)\right\|_2^2 \\
&= \left\|(1 - \eta_2 \lambda)(\overline{V}_t - (\overline{V}_{t_1} + V^\star)) - \eta_2(\lambda(\overline{V}_{t_1} + V^\star) + \nabla K_t)\right\|_2^2 \\
&= \left\|(1 - \eta_2 \lambda)(\overline{V}_t - (\overline{V}_{t_1} + V^\star))\right\|_2^2 + \eta_2^2 \left\|\lambda(\overline{V}_{t_1} + V^\star) + \nabla K_t\right\|_2^2 \\
&\quad - 2\eta_2(1 - \eta_2 \lambda)\langle \overline{V}_t - (\overline{V}_{t_1} + V^\star), \lambda(\overline{V}_{t_1} + V^\star)\rangle \\
&\quad - 2\eta_2(1 - \eta_2 \lambda)\langle \overline{V}_t - (\overline{V}_{t_1} + V^\star), \nabla K_t\rangle \\
&\leq \left\|(1 - \eta_2 \lambda)(\overline{V}_t - (\overline{V}_{t_1} + V^\star))\right\|_2^2 + 2\eta_2^2(\lambda^2 B^2 + K^2) - 2\eta_2 \lambda(1 - \eta_2 \lambda)(R + B)B \\
&\quad + 2\eta_2 \lambda(1 - \eta_2 \lambda)B^2 - 2\eta_2(1 - \eta_2 \lambda)(K_t(\overline{V}_t) - K_t(\overline{V}_{t_1} + V^\star)) \\
&\leq \left\|(1 - \eta_2 \lambda)(\overline{V}_t - (\overline{V}_{t_1} + V^\star))\right\|_2^2 + 2\eta_2^2(\lambda^2 B^2 + K^2) - 2\eta_2 \lambda(1 - \eta_2 \lambda)RB \\
&\quad - 2\eta_2(1 - \eta_2 \lambda)(K_t(\overline{V}_t) - K_t(\overline{V}_{t_1} + V^\star))
\end{aligned}
$$

For the sake of contradiction, assume that $(K_t^2(\overline{V}_t) - K_t^2(\overline{V}_{t_1} + V^\star)) \geq C$, let $0 < 1 - \eta_2 \lambda < 1$, and $\eta_2 \ll \frac{\lambda BR + C}{\lambda^2 B^2 + \lambda^2 BR + K^2 + \lambda C}$, and $\lambda R^2 \ll C$,

$$
\begin{aligned}
\left\|\overline{V}_{t+1} - (\overline{V}_{t_1} + V^\star)\right\|_2^2 &\leq \left\|(\overline{V}_t - (\overline{V}_{t_1} + V^\star))\right\|_2^2 + 2\eta_2^2(\lambda^2 B^2 + \lambda^2 BR + K^2 + \lambda C) - 2\eta_2(\lambda BR + C) \\
&\leq \left\|(\overline{V}_t - (\overline{V}_{t_1} + V^\star))\right\|_2^2 - 2\eta_2(\lambda BR + C) \\
&\leq \left\|(\overline{V}_t - (\overline{V}_{t_1} + V^\star))\right\|_2^2 - 2\eta_2 C
\end{aligned}
$$

Thus, in the specialized stage within $t_1 \leq t \leq t_1 + t_2$ iterations, $t_2 \triangleq \frac{\log^2(1/\epsilon_{V,1})}{\eta_2 \lambda \epsilon_{V,1}^2}$, $t_1 \triangleq \frac{1}{\eta_1 \lambda}$,

$$
\left\|\overline{V}_{t_1+t_2} - (\overline{V}_{t_1} + V^\star)\right\|_2^2 \leq \left\|(\overline{V}_{t_1} - (\overline{V}_{t_1} + V^\star))\right\|_2^2 - 2t\eta_2 C \leq \frac{\log^2(1/\epsilon_{V,1})}{\epsilon_{V,1}^2} - 2t\eta_2 C < 0
$$

which is a contradiction.

Finally, we conclude that, in the specialized stage within $t_2$ iterations, $t_2 \leq \frac{\log^2(1/\epsilon_{V,1})}{\eta_2 \lambda \epsilon_{V,1}^2}$, $t_1 \leq \frac{1}{\eta_1 \lambda}$, through gradient descent optimization, $\|\overline{V}_{t_1+t_2}\|_F$ satisfies $\|\overline{V}_{t_1+t_2}\|_F \leq B + R$, then

$$
\|\overline{V}_{t_1+t_2}\|_F = \Theta\left(\frac{\log(1/\epsilon_{V,1})}{\epsilon_{V,1}} + \frac{1}{\text{Poly}(d)}\right)
$$

and the training loss satisfies

$$K_{t_1+t_2}^2(\overline{V}_{t_1+t_2}) \leq K_{t_1+t_2}^2(\overline{V}_{t_1} + V^\star) + C \lesssim \epsilon_{V,1} + \frac{1}{(\log d)^{1/4}} + \frac{1}{\sqrt{\log d}}$$

## G.2 PROOF OF THEOREM 4

**Theorem.** *In the specialized stage with annealing learning rate $\eta_2 = \eta_1 \lambda^2 \epsilon_{V,1}^2 r$ and $t_1 \leq t \leq t_1 + t_2$, where $\epsilon_{V,1} = \Theta(1/Poly(d))$, $t_1 \triangleq \frac{1}{\eta_1 \lambda}$, $t_2 \triangleq \frac{\log^2(1/\epsilon_{V,1})}{\eta_2 \lambda \epsilon_{V,1}^2}$, $\lambda$ denotes the $L_2$ regularization coefficient and data noise $\|\zeta\|_2 = r$ (See Paragraph 3.1). With Assumption 1 and number of training prompts $N = \Theta(Poly(d))$, it holds that*

*(d.1) For the model parameter $W$ of network $h$, through gradient descent optimization from iteration $t_1$ to $t_1 + t_2$, $\|\overline{W}_{t_1+t_2} - \overline{W}_{t_1}\|_F$ satisfies*

$$\left\|\overline{W}_{t_1+t_2} - \overline{W}_{t_1}\right\|_F \lesssim \frac{\epsilon_{V,1}^2}{\log^2(1/\epsilon_{V,1})\sqrt{\log d}} - \frac{1}{Poly(d)}.$$

*(d.2) With random and small noise weight, the training loss of easy-to-fit component $\mathcal{P}$ over signal weight (Definition in Equation 2) satisfies*

$$\left|K_{t_1+t_2}^1(\overline{W}_{t_1+t_2}) - K_{t_1}^1(\overline{W}_{t_1}))\right| \lesssim \frac{\epsilon_{V,1}^2}{\log^2(1/\epsilon_{V,1})\sqrt{\log d}}.$$

*Namely, the network $h$ continues to preserve the easy-to-fit knowledge like $\mathcal{P}$ within $t_2$ iterations.*

**Remark 9** (**Proof Sketch**). *We summarize the proof sketch and main techniques in Proof of Theorem 4. At the first step, based on the expression for the training loss of component $\mathcal{P}$ over signal weight, we use the triangle and Cauchy-Schwarz inequality to transform the difference in training loss at times $t_1 + t_2$ and $t_1$, i.e. $\|K_{t_1+t_2}^1(\overline{W}_{t_1+t_2}) - K_{t_1}^1(\overline{W}_{t_1})\|$, into the difference in model weights at the two times, i.e. $\|\overline{W}_{t_1+t_2} - \overline{W}_{t_1}\|$. Following that, through gradient descent analysis, similar to the analysis of $\|\overline{W}_{t_1} - W^\star\|$ in Theorem 2, we derive $\|\overline{W}_{t_1+t_2} - W^\star\|$ and combine these to conclude $\|\overline{W}_{t_1+t_2} - \overline{W}_{t_1}\|$ in (a). Naturally utilizing the relationship between $\|K_{t_1+t_2}^1(\overline{W}_{t_1+t_2}) - K_{t_1}^1(\overline{W}_{t_1})\|$ and $\|\overline{W}_{t_1+t_2} - \overline{W}_{t_1}\|$ from the first step to derive (b). In total, we demonstrate that the model weight $W$ and training loss of $\mathcal{P}$ are almost stable.*

**Deal with gradient descent from $\overline{W}_{t_1}$ to $\overline{W}_{t_1+t_2}$.**   Similar to the optimization from $\overline{W}_0$ to $W^\star$ in Appendix F.2, we consider the graident descent of signal $\overline{W}_{t_1}$,

$$\overline{W}_{t+1} = \overline{W}_t - \eta_2 \nabla K_t(\overline{W}_t) - \eta_2 \lambda W_t$$
$$= (1 - \eta_2 \lambda)\overline{W}_t - \eta_2 \nabla K_t(\overline{W}_t)$$

With $\|W^*\|_F = d\log(1/\epsilon_{W,1}) \triangleq B$ from Equation 22, loss $K_t$ is K-Lipschitz, i.e. $\|\nabla K_t(\overline{W}_t)\|_F \leq K$. For $t_1 < t \leq t_1 + t_2$, assume that $\|\overline{W}_t - W^\star\|_F \leq R_2 \ll B$. For the sake of contradiction, assume that $K_t^1(\overline{W}_t) - K_t^1(W^\star) \geq C_2$, let $0 < 1 - \eta_2 \lambda < 1$, and $\eta_2 \ll \frac{\lambda B R_2 + C_2}{\lambda^2 B^2 + \lambda^2 B R_2 + K^2 + \lambda C_2}$, and $\lambda R_2^2 \ll C_2$,

$$\left\|\overline{W}_{t+1} - W^\star\right\|_2^2 \leq \left\|(\overline{W}_t - W^\star)\right\|_2^2 + 2\eta_2^2(\lambda^2 B^2 + \lambda^2 B R_2 + K^2 + \lambda C_2) - 2\eta_2(\lambda B R_2 + C_2)$$
$$\leq \left\|(\overline{W}_t - W^\star)\right\|_2^2 - 2\eta_2(\lambda B R_2 + C_2)$$
$$\leq \left\|(\overline{W}_t - W^\star)\right\|_2^2 - 2\eta_2 C_2$$

From Theorem 3, in the specialized stage within $t_2$ iterations, $t_2 \triangleq \frac{\log^2(1/\epsilon_{V,1})}{\eta_2 \lambda \epsilon_{V,1}^2}$, $t_1 \triangleq \frac{1}{\eta_1 \lambda}$. From the gradient descent in Appendix F.2, we have $\left\|\overline{W}_{t_1} - W^\star\right\|_F \leq R \ll B = d\log(1/\epsilon_{W,1})$, then

$$\left\|\overline{W}_{t_1+t_2} - W^\star\right\|_2^2 \leq \left\|(\overline{W}_{t_1} - W^\star)\right\|_2^2 - 2t_2\eta_2 C \leq R^2 - 2t_2\eta_2 C_2 < 0$$

which is a contradiction. We naturally have $R < \frac{R_2\epsilon_{V,1}}{\log(1/\epsilon_{V,1})}$, then we can derive that $\lambda R^2 < \frac{\lambda R_2^2 \epsilon_{V,1}^2}{\log^2(1/\epsilon_{V,1})} \ll C_2$. Thus, at iteration $t_1 + t_2$, the training loss of component $\mathcal{P}$ over signal weight satisfies

$$K_{t_1+t_2}^1\left(\overline{W}_{t_1+t_2}\right) \leq K_{t_1+t_2}^1(W^\star) + C_2 \lesssim \epsilon_{W,1} + \frac{\sqrt{d}\log d}{L}\epsilon_W + \frac{\epsilon_{V,1}^2}{\log^2(1/\epsilon_{V,1})\sqrt{\log d}}$$

Combining the conclusion in Theorem 2, we have that the difference of loss between iteration $t_1$ and $t_1 + t_2$ is

$$\left| K_{t_1+t_2}^1(\overline{W}_{t_1+t_2}) - K_{t_1}^1(\overline{W}_{t_1})) \right| \lesssim \frac{\epsilon_{V,1}^2}{\log^2\left(1/\epsilon_{V,1}\right)\sqrt{\log d}} \tag{25}$$

In the following, we would like to show that the changes in $W$ is also small. With 1-Lipschitzness of logistic loss, we know that

$$\left| K_{t_1+t_2}^1(\overline{W}_{t_1+t_2}) - K_{t_1}^1(\overline{W}_{t_1})) \right| = \left| \frac{1}{N}\sum_{n\in[N]} \left( l(N_{W_{t_1+t_2}}(\overline{W}_{t_1+t_2}; X_1^n, Y^n)) - l(N_{W_{t_1}}(\overline{W}_{t_1}; X_1^n, Y^n))) \right) \right|$$

$$\leq \frac{1}{N}\sum_{n\in[N]} \underbrace{\left| N_{W_{t_1+t_2}}(\overline{W}_{t_1+t_2}; X_1^n, Y^n) - N_{W_{t_1}}(\overline{W}_{t_1}; X_1^n, Y^n) \right|}_{A}$$

$$\tag{26}$$

With Corollary 4 and Corollary 6, we derive that

$$\left| N_{W_{t_1+t_2}}(\overline{W}_{t_1+t_2}; X_1^n, Y^n) - N_{W_{t_1}}(\overline{W}_{t_1}; X_1^n, Y^n) \right|$$

$$\leq \left| N_{W_{t_1+t_2}}(\overline{W}_{t_1+t_2}; X_1^n, Y^n) - N_{W_{t_1}}(\overline{W}_{t_1+t_2}, X_1^n, Y^n) \right| + \left| N_{W_{t_1}}(\overline{W}_{t_1+t_2}; X_1^n, Y^n) - N_{W_{t_1}}(\overline{W}_{t_1}; X_1^n, Y^n) \right|$$

$$\lesssim \left( \epsilon_W + L\sqrt{\frac{\eta_2}{\eta_1}} + \sqrt{L\log d} \right)\frac{K(u+\gamma_0)^2}{L\lambda} + \left| N_{W_{t_1}}(\overline{W}_{t_1+t_2}; X_1^n, Y^n) - N_{W_{t_1}}(\overline{W}_{t_1}; X_1^n, Y^n) \right|$$

**Deal with Term $A$.** $\tag{27}$

Substitute Equation 27 into Equation 26, and use Cauchy-Shwartz inequality,

$$\left| K_{t_1+t_2}^1(\overline{W}_{t_1+t_2}) - K_{t_1}^1(\overline{W}_{t_1})) \right|$$

$$\lesssim \frac{1}{N}\sum_{n\in[N]}\left| N_{W_{t_1}}(\overline{W}_{t_1+t_2}; X_1^n, Y^n) - N_{W_{t_1}}(\overline{W}_{t_1}; X_1^n, Y^n) \right| + \left( \epsilon_W + L\sqrt{\frac{\eta_2}{\eta_1}} + \sqrt{L\log d} \right)\frac{K(u+\gamma_0)^2}{L\lambda}$$

$$\lesssim \frac{1}{N}\sqrt{\sum_{n\in[N]}\left( N_{W_{t_1}}(\overline{W}_{t_1+t_2}; X_1^n, Y^n) - N_{W_{t_1}}(\overline{W}_{t_1}; X_1^n, Y^n) \right)^2} + \left( \epsilon_W + L\sqrt{\frac{\eta_2}{\eta_1}} + \sqrt{L\log d} \right)\frac{K(u+\gamma_0)^2}{L\lambda}$$

$$\lesssim \frac{1}{N}\sqrt{\sum_{n\in[N]}\underbrace{\left( N_{W_{t_1}}(\overline{W}_{t_1+t_2}; X_1^n, Y^n) - N_{W_{t_1}}(\overline{W}_{t_1}; X_1^n, Y^n) \right)^2}_{B}} + \frac{1}{\text{Poly}(d)}$$

where the last step comes with choice of small $u, r$, $\tau_0 = \mathcal{O}\left(\frac{1}{\sqrt{\log d}}\right)$, $\frac{1}{\lambda} = \mathcal{O}\left(\sqrt{\log d}\right)$ and $L = \Theta\left(\text{Poly}(d)\right)$, and $\eta_2 = \eta_1\lambda^2\epsilon_{V,1}^2 r$. With Assumption 2, We have

$$\left( N_{W_{t_1}}(\overline{W}_{t_1+t_2}; X_1^n, Y^n) - N_{W_{t_1}}(\overline{W}_{t_1}; X_1^n, Y^n) \right)^2$$

$$= \left( Y^n/L\left( \mathbb{1}\left([X_1^n]^\top W_{t_1} x_{L,1}\right) \odot \left([X_1^n]^\top \overline{W}_{t_1+t_2} x_{L,1}\right) \right) \right.$$

$$\left. - Y^n/L\left( \mathbb{1}\left([X_1^n]^\top W_{t_1} x_{L,1}\right) \odot \left([X_1^n]^\top \overline{W}_{t_1} x_{L,1}\right) \right) \right)^2$$

$$\leq \frac{1}{L^2}\max|Y_i^n|_2^2\left\| \mathbb{1}\left([X_1^n]^\top W_{t_1} x_{L,1}\right) \right\|_1^2\left\| [X_1^n]^\top \overline{W}_{t_1+t_2} x_{L,1} - [X_1^n]^\top \overline{W}_{t_1} x_{L,1} \right\|_2^2$$

$$\leq \frac{1}{L^2}\left\| \mathbb{1}\left([X_1^n]^\top W_{t_1} x_{L,1}\right) \right\|_1^2\left\| [X_1^n]^\top \right\|_F^2\left\| \overline{W}_{t_1+t_2} - \overline{W}_{t_1} \right\|_F^2\left\| x_{L,1} \right\|_2^2$$

$$\leq \frac{1}{L^2}\left\| \mathbb{1}\left([X_1^n]^\top W_{t_1} x_{L,1}\right) \right\|_1^2 L(u+\gamma_0)^4\left\| \overline{W}_{t_1+t_2} - \overline{W}_{t_1} \right\|_F^2$$

Using Corollary 2,

$$\|\mathbb{1}(X_1^\top W x_{L,1}) - \mathbb{1}(X_1^\top \widetilde{W} x_{L,1})\|_1 \lesssim K^{4/3}\lambda^{-4/3}\tau_0^{-4/3}L^{2/3} \triangleq \epsilon_W$$

thus further consider $\left\|\mathbb{1}(X_1^\top \widetilde{W}_{t_1} x_{L,1})\right\|_1$,

$$\left\|\mathbb{1}(X_1^\top \widetilde{W}_{t_1} x_{L,1})\right\|_1 = \sum_{i\in[L]} \mathbb{1}([X_1^\top]_i \widetilde{W}_{t_1} x_{L,1})$$

where $\mathbb{1}([X_1^\top]_i \widetilde{W}_{t_1} x_{L,1})$ is Bernoulli r.v., then using Hoeffding's inequality in Lemma 1,

$$\Pr\left(\sum_{i\in[L]} \mathbb{1}([X_1^\top]_i \widetilde{W}_{t_1} x_{L,1}) \geq t\right) \leq e^{-\frac{t^2}{2}}$$

Let $\delta = e^{-\frac{t^2}{2}}$, with $\delta = \frac{1}{d}$, $t = \sqrt{2\log\frac{1}{\delta}} = \sqrt{2\log d}$, then with probability at least $1 - \delta$ (i.e., $1 - \frac{1}{d}$),

$$\left\|\mathbb{1}(X_1^\top \widetilde{W}_{t_1} x_{L,1})\right\|_1 \lesssim \sqrt{\log d}$$

Using triangle inequality, we know that

$$\left\|\mathbb{1}(X_1^\top W_{t_1} x_{L,1})\right\|_1^2 \lesssim \left(\|\mathbb{1}(X^\top \widetilde{W}_{t_1} x_L)\|_1 + \epsilon_W\right)^2 \lesssim \left(\sqrt{\log d} + \epsilon_W\right)^2$$

Thus, for term B, we have

$$\left(N_{W_{t_1}}(\overline{W}_{t_1+t_2}; X_1^n, Y^n) - N_{W_{t_1}}(\overline{W}_{t_1}; X_1^n, Y^n)\right)^2$$

$$\leq \frac{1}{L^2}\left\|\mathbb{1}\left([X_1^n]^\top W_{t_1} x_{L,1}\right)\right\|_1^2 L(u + \gamma_0)^4 \left\|\overline{W}_{t_1+t_2} - \overline{W}_{t_1}\right\|_F^2$$

$$\lesssim \frac{(u + \gamma_0)^2}{L}\left(\sqrt{\log d} + \epsilon_W\right)^2 \left\|\overline{W}_{t_1+t_2} - \overline{W}_{t_1}\right\|_F^2$$

and then for $\left|K_{t_1+t_2}^1(\overline{W}_{t_1+t_2}) - K_{t_1}^1(\overline{W}_{t_1}))\right|$,

$$\left|K_{t_1+t_2}^1(\overline{W}_{t_1+t_2}) - K_{t_1}^1(\overline{W}_{t_1}))\right|$$

$$\lesssim \frac{1}{N}\sqrt{\sum_{n\in[N]} \underbrace{\left(N_{W_{t_1}}(\overline{W}_{t_1+t_2}; X_1^n, Y^n) - N_{W_{t_1}}(\overline{W}_{t_1}; X_1^n, Y^n)\right)^2}_{B}} + \frac{1}{\text{Poly(d)}}$$

$$\lesssim \frac{u + \gamma_0}{\sqrt{LN}}\left(\sqrt{\log d} + \epsilon_W\right)\left\|\overline{W}_{t_1+t_2} - \overline{W}_{t_1}\right\|_F + \frac{1}{\text{Poly(d)}}$$

Combining with Equation 25, we can derive that

$$\left\|\overline{W}_{t_1+t_2} - \overline{W}_{t_1}\right\|_F \lesssim \frac{\epsilon_{V,1}^2}{\log^2(1/\epsilon_{V,1})\sqrt{\log d}} - \frac{1}{\text{Poly}(d)}$$

when $\sqrt{LN} = \Theta(\sqrt{\log d} + \epsilon_W)$, i.e. $N = \Theta(\text{Poly}(d))$. Therefore, we conclude that in the specialized stage, the changes in $W$ and the loss in the $h$ network are both small, and the loss remains very low.

$$\left\|\overline{W}_{t_1+t_2} - \overline{W}_{t_1}\right\|_F \lesssim \frac{\epsilon_{V,1}^2}{\log^2(1/\epsilon_{V,1})\sqrt{\log d}} - \frac{1}{\text{Poly}(d)}$$

and

$$\left|K_{t_1+t_2}^1(\overline{W}_{t_1+t_2}) - K_{t_1}^1(\overline{W}_{t_1}))\right| \lesssim \frac{\epsilon_{V,1}^2}{\log^2(1/\epsilon_{V,1})\sqrt{\log d}}$$

$\square$

## H  PROOF FOR SPECTRAL CHARACTERISTICS

### H.1  PROOF OF COROLLARY 1

**Compute the gradient of weight $W_K$ and $W_Q$.**   With one normalized Relu self-attention layer, we have

$$f(U; X, \widetilde{Y}) = \widetilde{Y} \cdot \frac{1}{2L} \text{ReLU} \left( X^\top W_K^\top W_Q x_L \right)$$

$$= \widetilde{Y}/2L \cdot \text{ReLU} \left( X^\top U x_L \right)$$

where $X \in \mathbb{R}^{2d \times 2L}, U = W_K^\top W_Q \in \mathbb{R}^{2d \times 2d}$. Consider the gradient of weight $W_K$ and $W_Q$,

$$\nabla_{W_K} \widehat{L}(U) = \widehat{\mathbb{E}} \left[ l'(f(U; X, \widetilde{Y})) \nabla (y_L f(U; X, \widetilde{Y})) \right]$$

$$= \widehat{\mathbb{E}} \left[ l'(f(U; X, \widetilde{Y})) y_L \nabla \left( \widetilde{Y}/2L \cdot \text{ReLU} \left( X^\top W_K^\top W_Q x_L \right) \right) \right]$$

$$= \widehat{\mathbb{E}} \left[ 1/2L \cdot l'(f(U; X, \widetilde{Y})) y_L \sum_{i=1}^{2L} y_i \nabla \text{ReLU} \left( [X^\top]_i W_K^\top W_Q x_L \right) \right]$$

$$= \widehat{\mathbb{E}} \left[ 1/2L \cdot l'(f(U; X, \widetilde{Y})) y_L \sum_{i=1}^{2L} y_i \mathbb{1}([X^\top]_i W_K^\top W_Q x_L) W_Q x_L [X^\top]_i \right]$$

$$= \widehat{\mathbb{E}} \left[ 1/2L \cdot l'(f(U; X, \widetilde{Y})) W_Q \left( X \cdot \text{diag} \left( \mathbb{1}(X^\top W_K^\top W_Q x_L) \right) x_L^\top \right)^\top \right]$$

$$[\nabla_{W_K} \widehat{L}(U_t)]_i = \widehat{\mathbb{E}} \left[ 1/2L \cdot l'(f(U_t; X, \widetilde{Y})) y_L y_i \mathbb{1}([X^\top]_i W_K^\top W_Q x_L) [W_Q]_i x_L [X^\top]_i \right]$$

$$[\nabla_{W_K} \widehat{L}(U_t)]_j = \widehat{\mathbb{E}} \left[ 1/2L \cdot l'(f(U_t; X, \widetilde{Y})) y_L y_j \mathbb{1}([X^\top]_j W_K^\top W_Q x_L) [W_Q]_j x_L [X^\top]_j \right]$$

Similarly, we have

$$\nabla_{W_Q} \widehat{L}(U) = \widehat{\mathbb{E}} \left[ l'(f(U; X, \widetilde{Y})) \nabla (y_L f(U; X, \widetilde{Y})) \right]$$

$$= \widehat{\mathbb{E}} \left[ l'(f(U; X, \widetilde{Y})) y_L \nabla \left( \widetilde{Y}/2L \cdot \text{ReLU} \left( X^\top W_K^\top W_Q x_L \right) \right) \right]$$

$$= \widehat{\mathbb{E}} \left[ 1/2L \cdot l'(f(U; X, \widetilde{Y})) y_L \sum_{i=1}^{2L} y_i \nabla \text{ReLU} \left( [X^\top]_i W_K^\top W_Q x_L \right) \right]$$

$$= \widehat{\mathbb{E}} \left[ 1/2L \cdot l'(f(U; X, \widetilde{Y})) y_L \sum_{i=1}^{2L} y_i \mathbb{1}([X^\top]_i W_K^\top W_Q x_L) W_K X_i x_L^\top \right]$$

$$= \widehat{\mathbb{E}} \left[ 1/2L \cdot l'(f(U; X, \widetilde{Y})) W_K X \cdot \text{diag} \left( \mathbb{1}(X^\top W_K^\top W_Q x_L) \right) x_L^\top \right]$$

$$[\nabla_{W_Q} \widehat{L}(U_t)]_i = \widehat{\mathbb{E}} \left[ 1/2L \cdot l'(f(U_t; X, \widetilde{Y})) y_L y_i \mathbb{1}([X^\top]_i W_K^\top W_Q x_L) [W_K]_i X_i x_L^\top \right]$$

$$[\nabla_{W_Q} \widehat{L}(U_t)]_j = \widehat{\mathbb{E}} \left[ 1/2L \cdot l'(f(U_t; X, \widetilde{Y})) y_L y_j \mathbb{1}([X^\top]_j W_K^\top W_Q x_L) [W_K]_j X_j x_L^\top \right]$$

With $l = -\log \sigma \left( y_L f(U; X, \widetilde{Y}) \right)$, we have $l' \triangleq l'(f(U; X, \widetilde{Y})) = \frac{-y_L \exp(-y_L f(U; X, \widetilde{Y}))}{1 + \exp(-y_L f(U; X, \widetilde{Y}))}$. According to $\nabla_{W_K} \widehat{L}(U)$ and $\nabla_{W_Q} \widehat{L}(U)$, let $A = \widehat{\mathbb{E}} \left[ l' X \cdot \text{diag} \left( \mathbb{1}(X^\top W_K^\top W_Q x_L) \right) x_L^\top \right] \in \mathbb{R}^{d \times d}$, then we have

$$W_{K,t+1} = W_{K,t} - \eta \nabla_{W_{K,t}} \widehat{L}(U_t) - \eta \lambda W_{K,t}$$

$$= (1 - \eta \lambda) W_{K,t} - \eta \nabla_{W_{K,t}} \widehat{L}(U_T)$$

$$= (1 - \eta \lambda) W_{K,t} - \eta/2L \cdot W_{Q,t} A_t^\top$$

Similarly,

$$W_{Q,t+1} = W_{Q,t} - \eta \nabla_{W_{Q,t}} \widehat{L}(U_t) - \eta \lambda W_{Q,t}$$

$$= (1 - \eta \lambda) W_{Q,t} - \eta \nabla_{W_{Q,t}} \widehat{L}(U_t)$$

$$= (1 - \eta \lambda) W_{Q,t} - \eta/2L \cdot W_{K,t} A_t$$

**Eigen decomposition and the gradient descent of eigenvalues.** Assume that $W_K \simeq W_Q$ and simultaneous diagonolizability,

$$W_K = M \cdot \mathrm{diag}(\sigma(W_K))\Phi^\top$$
$$W_Q = M \cdot \mathrm{diag}(\sigma(W_Q))\Phi^\top$$

Then,

$$
\begin{aligned}
W_{K,t+1} &= (1 - \eta\lambda)W_{K,t} - \eta/2L \cdot W_{Q,t}[A_t]^\top \\
&= (1 - \eta\lambda)W_{K,t} - \eta/2L \cdot M_t \cdot \mathrm{diag}(\sigma(W_{Q,t}))\Phi_t^\top[A_t]^\top \\
&= (1 - \eta\lambda)W_{K,t} - \eta/2L \cdot M_t \cdot \mathrm{diag}(\sigma(W_{Q,t}))\Phi_t^\top[A_t]^\top\Phi_t\Phi_t^\top \\
&= (1 - \eta\lambda)W_{K,t} - \eta/2L \cdot M_t \cdot \mathrm{diag}(\sigma(W_{Q,t}))\left(\Phi_t^\top A_t\Phi_t\right)^\top\Phi_t^\top \\
W_{Q,t+1} &= (1 - \eta\lambda)W_{Q,t} - \eta/2L \cdot W_{K,t}A_t \\
&= (1 - \eta\lambda)W_{Q,t} - \eta/2L \cdot M_t \cdot \mathrm{diag}(\sigma(W_{K,t}))\Phi_t^\top A_t \\
&= (1 - \eta\lambda)W_{Q,t} - \eta/2L \cdot M_t \cdot \mathrm{diag}(\sigma(W_{K,t}))\Phi_t^\top A_t\Phi_t\Phi_t^\top \\
&= (1 - \eta\lambda)W_{Q,t} - \eta/2L \cdot M_t \cdot \mathrm{diag}(\sigma(W_{K,t}))\left(\Phi_t^\top A_t\Phi_t\right)\Phi_t^\top
\end{aligned}
$$

If we have $A$ is symmetric and $\Phi^\top A\Phi$ is diagonal, then for the eigenvalues of $W_K$ and $W_Q$, i.e. $\sigma(W_K)$ and $\sigma(W_Q)$,

$$
\begin{aligned}
\sigma\left(W_{K,t+1}\right) &= (1 - \eta\lambda)\sigma\left(W_{K,t}\right) - \eta/2L \cdot \sigma(W_{Q,t}) \odot \sigma([A_t]^\top) \\
\sigma\left(W_{Q,t+1}\right) &= (1 - \eta\lambda)\sigma\left(W_{Q,t}\right) - \eta/2L \cdot \sigma(W_{K,t}) \odot \sigma(A_t)
\end{aligned}
$$

Let $\sqrt{w} = \sigma(W_K) = \sigma(W_Q) \in \mathbb{R}^d$ and $w = \sigma(U) = \sigma(W_K) \odot \sigma(W_Q) \in \mathbb{R}^d$, $\alpha = \sigma(A)$,

$$
\begin{aligned}
\sigma\left(W_{K,t+1}\right) \odot \sigma\left(W_{Q,t+1}\right) =& (1 - \eta\lambda)\left(\sigma\left(W_{K,t}\right) \odot \sigma\left(W_{Q,t}\right)\right) - \eta/2L \cdot \left(\sigma(W_{K,t})^{\odot 2}\right) \odot \sigma(A_t) \\
& - \eta/2\left(\sigma(W_{Q,t})^{\odot 2}\right) \odot \sigma([A_t]^\top) \\
=& (1 - \eta\lambda)\left(\sigma\left(W_{K,t}\right) \odot \sigma\left(W_{Q,t}\right)\right) - \eta/2L\left(\sigma(W_{K,t})^{\odot 2} + \sigma(W_{Q,t})^{\odot 2}\right) \odot \sigma(A_t)
\end{aligned}
$$

Finally, we have

$$w^{t+1} = (1 - \eta\lambda)w^t - \eta/2L \cdot 2w^t \odot \alpha^t$$

**Analysis the relationship of $\overline{\alpha} = \mathbf{Tr}(A)$ and $\overline{w} = \mathbf{Tr}(U)$.** In the following, we analysis the relationship of $\overline{\alpha}$ and $\overline{w}$. To Compute trace of matrix $A$,

$$
\begin{aligned}
\mathrm{Tr}(A) &= \mathrm{Tr}\left(\widehat{\mathbb{E}}\left[l'X \cdot \mathrm{diag}\left(\mathbb{1}(X^\top W_K^\top W_Q x_L)\right)x_L^\top\right]\right) \\
&= \widehat{\mathbb{E}}\left[\mathrm{Tr}\left(l'X \cdot \mathrm{diag}\left(\mathbb{1}(X^\top W_K^\top W_Q x_L)\right)x_L^\top\right)\right] \\
&= \widehat{\mathbb{E}}\left[l'\underbrace{\mathrm{Tr}\left(X \cdot \mathrm{diag}\left(\mathbb{1}(X^\top W_K^\top W_Q x_L)\right)x_L^\top\right)}_{M}\right]
\end{aligned}
$$

For term $M$,

$$
\begin{aligned}
M &= \mathrm{Tr}\left(X \cdot \mathrm{diag}\left(\mathbb{1}(X^\top W_K^\top W_Q x_L)\right)x_L^\top\right) \\
&= \sum_{i=1}^d\left(\sum_{j=1}^L X_{ij}\left[\mathbb{1}(X^\top W_K^\top W_Q x_L)\right]_j\right)x_{Li} \\
&\leq \max(\|x\|_2^2)\sum_{j=1}^L\underbrace{\left[\mathbb{1}(X^\top W_K^\top W_Q x_L)\right]_j}_{Z}
\end{aligned}
$$

For term $Z$,

$$W_K = M \cdot \mathrm{diag}(\sigma(W_K))\Phi^\top$$

$$W_Q = M \cdot \mathrm{diag}(\sigma(W_Q))\Phi^\top$$

$$X^\top W_K^\top W_Q x_L = X^\top \Phi \cdot \mathrm{diag}(\sqrt{w}) M^\top M \cdot \mathrm{diag}(\sqrt{w})\Phi^\top x_L$$

$$= X^\top \Phi \cdot \mathrm{diag}(w)\Phi^\top x_L$$

and then

$$\left[\mathbb{1}(X^\top W_K^\top W_Q x_L)\right]_j = \left[\mathbb{1}(X^\top \Phi \cdot \mathrm{diag}(w)\Phi^\top x_L)\right]_j$$

$$= \left[\left(\mathbb{1}(X^\top \Phi)\mathbb{1}(\mathrm{diag}(w))\mathbb{1}(\Phi^\top x_L)\right)\right]_j$$

$$= \mathbb{1}\left([X^\top]_j \Phi\right)\mathbb{1}(\mathrm{diag}(w))\mathbb{1}(\Phi^\top x_L)$$

$$= \sum_{k=1}^d \mathbb{1}([X^\top]_j\Phi)\mathbb{1}(\Phi^\top x_L)\mathbb{1}(w_k)$$

$$= \sum_{k=1}^d \mathbb{1}([X^\top]_j x_L)\mathbb{1}(w_k)$$

Combine term $M$ and term $Z$, and assume that almost $\forall w_i > 0$, then we have

$$\overline{\alpha} = \mathrm{Tr}(A)$$

$$= \widehat{\mathbb{E}}\left[l'\mathrm{Tr}\left(X \cdot \mathrm{diag}\left(\mathbb{1}(X^\top W_K^\top W_Q x_L)\right)x_L^\top\right)\right]$$

$$= p_-\mathbb{E}\left[l'_-\mathrm{Tr}\left(X \cdot \mathrm{diag}\left(\mathbb{1}(X^\top W_K^\top W_Q x_L)\right)x_L^\top\right)\right] + p_+\mathbb{E}\left[l'_+\mathrm{Tr}\left(X \cdot \mathrm{diag}\left(\mathbb{1}(X^\top W_K^\top W_Q x_L)\right)x_L^\top\right)\right]$$

$$\geq p\widehat{\mathbb{E}}\left[l'_- \max(\|x\|_2^2)\sum_{j=1}^L \left[\mathbb{1}(X^\top W_K^\top W_Q x_L)\right]_j\right] + (1-p)\mathbb{E}\left[l'_+\mathrm{Tr}\left(X \cdot \mathrm{diag}\left(\mathbb{1}(X^\top W_K^\top W_Q x_L)\right)x_L^\top\right)\right]$$

$$= p\widehat{\mathbb{E}}\left[l'_- \max(\|x\|_2^2)\sum_{j=1}^L \sum_{k=1}^d \mathbb{1}([X^\top]_j x_L)\mathbb{1}(w_k)\right]$$

$$= p\max(\|x\|_2^2)\widehat{\mathbb{E}}\left[l'_-\sum_{j=1}^L \sum_{k=1}^d \mathbb{1}([X^\top]_j x_L)\mathbb{1}(w_k)\right]$$

$$= p\max(\|x\|_2^2)\widehat{\mathbb{E}}\left[l'_-\mathbf{1}^\top\mathbb{1}(X^\top x_L)\right] \triangleq -pk$$

where $p$ is the proportion of negative logistic loss, $k = \max(\|x\|_2^2)\widehat{\mathbb{E}}\left[|l'_-|\mathbf{1}^\top\mathbb{1}(X^\top x_L)\right] > 0$. We conclude that the lower bound of $\overline{\alpha}$ is independent with $\overline{w}$, naturally,

$$\overline{w}_{t+1} \leq (1-\eta\lambda)\overline{w}_t + \eta/2L \cdot 2pk\overline{w}_t$$

**Analysis** $W_t$ **and** $V_t$. By similar proof, for $W = [W_K^1]^\top W_Q^1$, let $A^1 = \widehat{\mathbb{E}}\left[l'X_1 \cdot \mathrm{diag}\left(\mathbb{1}(X_1^\top[W_K^1]^\top W_Q^1 x_{L,1})\right)x_{L,1}^\top\right] \in \mathbb{R}^{d\times d}$, $w^1 = \sigma(W_K^1) \odot \sigma(W_Q^1) \in \mathbb{R}^d$, $\alpha^1 = \sigma(A^1)$, we also have

$$\nabla_{W_K^1}\widehat{L}(W) = \widehat{\mathbb{E}}\left[1/L \cdot l'(f(W;X_1,Y))W_Q^1\left(X_1 \cdot \mathrm{diag}\left(\mathbb{1}(X_1^\top[W_K^1]^\top W_Q^1 x_{L,1})\right)x_{L,1}^\top\right)^\top\right]$$

$$\nabla_{W_Q^1}\widehat{L}(W) = \widehat{\mathbb{E}}\left[1/L \cdot l'(f(W;X_1,Y))W_K^1 X_1 \cdot \mathrm{diag}\left(\mathbb{1}(X_1^\top[W_K^1]^\top W_Q^1 x_{L,1})\right)x_{L,1}^\top\right]$$

and $p_1$ is the proportion of the negative derivative of logistic loss $l'(f(W;X_1,Y)) < 0$

$$\overline{w}_{t+1}^1 = (1-\eta\lambda)\overline{w}_t^1 + 2p_1 k_1\eta/L \cdot \overline{w}_t^1, \quad k_1 \triangleq \max(\|x\|_2^2)\widehat{\mathbb{E}}\left[|l'_-|\mathbf{1}^\top\mathbb{1}(X_1^\top x_{L,1})\right]$$

For $V = [W_K^2]^\top W_Q^2$, let $A^2 = \widehat{\mathbb{E}}\left[l' X_2 \cdot \text{diag}\left(\mathbb{1}(X_2^\top [W_K^2]^\top W_Q^2 x_{L,2})\right) x_{L,2}^\top\right] \in \mathbb{R}^{d \times d}$, $w^2 = \sigma(W_K^2) \odot \sigma(W_Q^2) \in \mathbb{R}^d$, $\alpha^2 = \sigma(A^2)$, we have

$$\nabla_{W_K^2} \widehat{L}(V) = \widehat{\mathbb{E}}\left[1/L \cdot l'(f(V; X_2, Y)) W_Q^2 \left(X_2 \cdot \text{diag}\left(\mathbb{1}(X_2^\top [W_K^2]^\top W_Q^2 x_{L,2})\right) x_{L,2}^\top\right)^\top\right]$$

$$\nabla_{W_Q^2} \widehat{L}(V) = \widehat{\mathbb{E}}\left[1/L \cdot l'(f(V; X_2, Y)) W_K^2 X_2 \cdot \text{diag}\left(\mathbb{1}(X_2^\top [W_K^2]^\top W_Q^2 x_{L,2})\right) x_{L,2}^\top\right]$$

and $p_2$ is the proportion of the negative derivative of logistic loss $l'(f(V; X_2, Y)) < 0$

$$\overline{w}_{t+1}^2 = (1 - \eta\lambda)\overline{w}_t^2 + 2p_2 k_2 \eta/L \cdot \overline{w}_t^2, \quad k_2 \triangleq \max(\|x\|_2^2)\widehat{\mathbb{E}}\left[|l'_-|\mathbf{1}^\top \mathbb{1}(X_2^\top x_{L,2})\right]$$

**In the elementary stage.** With learning rate $\eta_1$, $\text{Tr}(W_t) \triangleq \overline{w}_t^1$, and $\text{Tr}(V_t) \triangleq \overline{w}_t^2$, we have

$$\overline{w}_{t+1}^1 = (1 - \eta_1\lambda)\overline{w}_t^1 + 2p_1 k_1 \eta_1/L \cdot \overline{w}_t^1, \quad k_1 \triangleq \max(\|x\|_2^2)\widehat{\mathbb{E}}\left[|l'_-|\mathbf{1}^\top \mathbb{1}(X_1^\top x_{L,1})\right]$$

$$\overline{w}_{t+1}^2 = (1 - \eta_1\lambda)\overline{w}_t^2 + 2p_2 k_2 \eta_1/L \cdot \overline{w}_t^2, \quad k_2 \triangleq \max(\|x\|_2^2)\widehat{\mathbb{E}}\left[|l'_-|\mathbf{1}^\top \mathbb{1}(X_2^\top x_{L,2})\right]$$

then through $t_1 \leq \frac{1}{\eta_1\lambda}$ iterations, according to the dynamic of the trace of $W$ and $V$,

$$\overline{w}_{t_1}^1 = (1 - \eta_1\lambda + 2p_1 k_1 \eta_1/L)^{t_1} \overline{w}_0^1$$

$$\overline{w}_{t_1}^2 = (1 - \eta_1\lambda + 2p_2 k_2 \eta_1/L)^{t_1} \overline{w}_0^2$$

We conclude that $\text{Tr}(W_t)$ and $\text{Tr}(V_t)$ have similar update rules where the rate of exponential growth over time mainly depends on three factors: (1) The learning rate $\eta_1$. (2) The proportion of the negative derivative of logistic loss $p$. (3) The negative derivative of the logistic loss is selected based on the similarity between query $x_L$ and sequence $X$, i.e. $\mathbb{1}(X_1^\top x_{L,1})$. Further compute $k$ with the mean absolute value of the selected negative derivative.

Combine Theorem 2 with small and random noise, $\|W_{t_1}\|_F \approx \|\overline{W}_{t_1}\|_F$ and $\|V_{t_1}\|_F \approx \|\overline{V}_{t_1}\|_F$, we conclude the following corollary that at time $t_1$,

$$\overline{w}_{t_1}^1 = \text{Tr}(W_{t_1}) \leq \sqrt{\text{Tr}(W_{t_1}^\top W_{t_1})} = \|W_{t_1}\|_F \lesssim d\log(1/\epsilon_{W,1})$$

$$\overline{w}_{t_1}^2 = \text{Tr}(V_{t_1}) \leq \sqrt{\text{Tr}(V_{t_1}^\top V_{t_1})} = \|V_{t_1}\|_F \lesssim \frac{1}{\text{Poly}(d)}$$

Finally, we have

$$\text{Tr}(W_{t_1}) > \text{Tr}(V_{t_1})$$

**In the specialized stage.** With learning rate $\eta_2$, $\text{Tr}(W_t) \triangleq \overline{w}_t^1$, and $\text{Tr}(V_t) \triangleq \overline{w}_t^2$, we have

$$\overline{w}_{t+1}^1 = (1 - \eta_2\lambda)\overline{w}_t^1 + 2p_1 k_1 \eta_2/L \cdot \overline{w}_t^1, \quad k_1 \triangleq \max(\|x\|_2^2)\widehat{\mathbb{E}}\left[|l'_-|\mathbf{1}^\top \mathbb{1}(X_1^\top x_{L,1})\right]$$

$$\overline{w}_{t+1}^2 = (1 - \eta_2\lambda)\overline{w}_t^2 + 2p_2 k_2 \eta_2/L \cdot \overline{w}_t^2, \quad k_2 \triangleq \max(\|x\|_2^2)\widehat{\mathbb{E}}\left[|l'_-|\mathbf{1}^\top \mathbb{1}(X_2^\top x_{L,2})\right]$$

Through $t_2 \leq \frac{\log^2(1/\epsilon_{V,1})}{\eta_2\lambda\epsilon_{V,1}^2}$ iterations, according to the dynamic of the trace of $W$ and $V$,

$$\overline{w}_{t_1+t_2}^1 = (1 - \eta_2\lambda + 2p_1 k_1 \eta_2/L)^{t_2} \overline{w}_{t_1}^1$$

$$\overline{w}_{t_1+t_2}^2 = (1 - \eta_2\lambda + 2p_2 k_2 \eta_2/L)^{t_2} \overline{w}_{t_1}^2$$

Similar to the elementary stage, we conclude that $\text{Tr}(W_t)$ and $\text{Tr}(V_t)$ still have similar update rules where the rate of exponential growth over time mainly depends on three factors.

Combine with Theorem 3 and 4, we have

$$\overline{w}_{t_1+t_2}^1 = \text{Tr}(W_{t_1+t_2}) \leq \sqrt{\text{Tr}(W_{t_1+t_2}^\top W_{t_1+t_2})} = \|W_{t_1+t_2}\|_F \lesssim d\log(1/\epsilon_{W,1}) + \frac{\log(1/\epsilon_{V,1})}{\lambda^{3/2}\epsilon_{V,1}^2}$$

$$\overline{w}_{t_1+t_2}^2 = \text{Tr}(V_{t_1+t_2}) \leq \sqrt{\text{Tr}(V_{t_1+t_2}^\top V_{t_1+t_2})} = \|V_{t_1+t_2}\|_F \lesssim \frac{1}{\text{Poly}(d)} + \frac{\log(1/\epsilon_{V,1})}{\epsilon_{V,1}}$$

Finally, we have

$$\text{Tr}(W_{t_1+t_2}) < \text{Tr}(V_{t_1+t_2})$$

