# OpenReview forum: "Mastering Syntax, Unlocking Semantics: A Mathematically Provable Two-stage Learning Process in Transformers"
_ICLR.cc/2025/Conference — ICLR 2025 Conference Withdrawn Submission_

### Official Review · Reviewer_PxzQ · 2024-10-28

**Soundness:** 2
**Presentation:** 1
**Contribution:** 2
**Rating:** 3
**Confidence:** 2

**Summary:**

This paper attempts to show that transformer language models learn in two stages: in the first, they learn syntax, and in the second, semantics. The authors justify this via mathematical proof. They study simple, one-layer transformer models, trained on data composed of two components—P and Q—where the former is easy and the latter difficult to learn. They show that given a limited number of timesteps, the model will only able to learn P at first; Q will be learned later. They analogize P to syntax, and Q to semantics. The authors then perform experiments on factual knowledge datasets to show that models do indeed learn syntax (word categories) first, but only acquire semantics (factual knowledge) later.

**Strengths:**

This paper engages in a seemingly rigorous theoretical analysis of learning trajectories in transformer language models, and attempts to connect this theoretical analysis to real trends in language models' learning trajectories. This is valuable, given that we still have few theoretical insights about learning in transformers that usefully transfer to real-world models.

**Weaknesses:**

- **Weak connection between proofs and core claims**: I do not claim to understand the proofs very well, but my understanding is that the proofs attempt to characterize the learning trajectory of a transformer-based model trained on data with an easy-to-fit component P and hard-to-fit component Q. This model (which is really composed of two sub-models working in parallel) is able to learn the P component quickly, while the Q component is learned later. This is all fine, but it is then claimed (186-188) that P aligns with the syntactic information in the corpus (easy to learn) and that Q aligns with the semantic information in the corpus (hard to learn). This assumption is unjustified: how do you know that syntax and semantics specifically should align with P and Q? It may well be that models learn in two stages (if their data are made up of components from two different distributions, one easy and one hard to learn), but how do you know that these two distributions of disparate difficulty exist in reality, or that syntax and semantics correspond to the easy/hard distributions? This assumption is really the crux of the paper, but goes unjustified.
- **Unclear what this paper believes syntax / semantics to be**: This paper claims to show that the models first learn syntactic and then semantic information, but what exactly syntactic and semantic information (or, alternatively "elementary" and "specialized" knowledge) are goes undiscussed.
    - **Operationalization of syntactic ability**: The authors seem to operationalize models' syntactic ability as "the model predicts precisely the right category we expect", but this operationalization is flawed. Consider one example given in the paper, "The mother tongue of Danielle Darrieux is"; based on the fact that the model outputs "the" rather than "French" or "Dutch", they infer that it lacks syntactic abilities. However, such an inference is not licensed: "the" is a perfectly valid start to many continuations. For example, the sentence could continue like "The mother tongue of Danielle Darrieux is the 6th most spoken language in the world". This sentence is syntactically (and semantically) valid, even though the next token is not a language. If the authors really want to make a claim like this, they should measure the probability assigned to tokens that cannot plausibly begin a syntactically valid completion. Moreover, prior work in the field has already come up with ways of measuring models' syntactic knowledge; see syntax benchmarks like [SyntaxGym](https://aclanthology.org/2020.acl-demos.10/) and [BLiMP](https://direct.mit.edu/tacl/article/doi/10.1162/tacl_a_00321/96452/BLiMP-The-Benchmark-of-Linguistic-Minimal-Pairs), among many others.
    - **Syntax vs. semantics, or something else?**:  This paper only considers syntax and semantics as the two components of a given token, but in its empirical evaluations (if we take them to be valid), we only see that one specific type of syntactic knowledge (next-token category constraints) emerges prior to a specific type of "semantic" knowledge (factual / world knowledge). But this doesn't license a claim about syntax and semantics more broadly; there are many different categorizations of these datapoints that could separate these two. For one, many would consider factual/world knowledge to be something distinct from other types of semantic knowledge; see e.g. [the formal-functional distinction](https://arxiv.org/abs/2301.06627). In general, to make this broad syntax vs. semantics claim, you'd need to test more types of syntactic vs. semantic abilities.
    - **No engagement with prior work from computational linguistics studying the same question**: The (computational) linguistics literature has tackled the very question that this paper purports to answer, with similar results. [This](https://direct.mit.edu/coli/article/50/1/293/118131/Language-Model-Behavior-A-Comprehensive-Survey) is a nice summary; Section 3.3, "Language Models Learn Syntactic Rules Early in Pre-training" tackles this very question. [Other work](https://proceedings.mlr.press/v235/belrose24a.html) takes a more theoretical view on these issues.
- **Little empirical evidence for claims**: The core claim of this paper, that models learn syntactic information first and semantic information second, is a plainly empirical one. The easiest thing to do would be to simply come up with ways to measure models' syntactic and semantic knowledge at each point in training, and then plot how these abilities change over time. This would involve a variety of tasks / metrics for syntax and semantics, and careful thought about which tasks/metrics make sense. This paper only measures LM loss, which doesn't tell us anything about syntactic or semantic abilities, trains on one very limited dataset, and performs limited qualitative analysis of model abilities to make claims about syntactic / semantic abilities. This is insufficient.

Ultimately, I feel that this paper could potentially have value to people in more (ML-)theoretically oriented communities for its proofs. But the connection between the mathematical results and the more linguistically oriented claims is tenuous, and the engagement with related work from the field it makes claims about is very low.

**Questions:**

- Why do you claim (186-188) "For each individual input sample $x_i^n$ in prompt $P_n$, it is composed of two types of components: $P$ component represents easy-to-fit features, aligning with syntactic information in the corpus, and $Q$ component represents hard-to-fit features, aligning with semantic information in the corpus."? Particularly, what justifies the alignment between $P$ and $Q$ with syntax and semantics specifically? Can you prove that syntactic and semantic features (however you want to define them) are distributed in such a way that aligns with your definitions of $P$ and $Q$?
- Have you considered alternative hypotheses regarding what the easy/hard-to-learn components of examples are, besides syntax and semantics? It is important to rule these out before concluding that the model is learning syntax and semantics, specifically, in that order.
- How would you position your work relative to prior work that also suggests, using stronger empirical evaluations, that models learn syntax early on? As I see it, the "models learn syntax first" angle isn't new, and this work is better positioned as an explanation for why they do so. But this would require justifying the claim that syntax and semantics actually fulfill the conditions behind P and Q in your proofs.
- You cite (Vaswani, 2017) for the transformer model, but shouldn't this be (Vaswani et al., 2017)?
- How do you train the model using which Figure 5 was made? You say that it uses the GPT-2 architecture (this should be cited) and the counterfact dataset, but do not say anything about how the model is actually trained. I guess that you do train it, though, since you have per-epoch results, and these are not typically available for GPT-2 models

---

> ### Author Response · Authors · 2024-11-26
>
> Thank you sincerely for your thoughtful comments! We understand that your primary concern lies in aligning our theoretical modeling with practical syntax-semantic analysis. We sincerely regret any confusion caused by our introduction of the syntax, semantics or easy-to-fit component $\mathcal{P}$, hard-to-fit component $\mathcal{Q}$. We respectfully argue that these theoretical modelings are indeed motivated by experimental findings and are estabilished to provide a ***rigorous mathematically analysis*** for this stage-wise learning phenomenon. In addition to the current empirical validations, as you suggested (e.g., syntax benchmarks like BLiMP), we have designed more experiments with specific quantitative metrics about the syntax and semantics to further support our theory.
>
> We strongly encourage to read ***General Response*** for detailed clarification of our motivation regarding the statements on syntax and semantics, as well as further experimental evidence (primarily addressing your concerns in Weaknesses, Question 1 and 3)! Below, we do our best to address your additional questions adequately.
>
> > **Q2:** Have you considered alternative hypotheses regarding what the easy/hard-to-learn components of examples are, besides syntax and semantics? It is important to rule these out before concluding that the model is learning syntax and semantics, specifically, in that order.
>
> **A:** Thanks for your question! As you mentioned, there are some empirical studies which have already examined various aspects of language. This provides a strong foundation to support that syntax and semantics are representative and well-understood components of linguistic features. Moreover, we would like to emphasize that the core contribution of our research lies in offering ***a theoretical understanding*** that explicitly characterizes the model's learning dynamics across features of differing complexity. Syntax and semantics serve as **proxies** for easy-to-fit (syntax) and hard-to-fit (semantics) features. The focus is not to claim these are the *only* such features but rather to demonstrate the general mechanism of how the model learns features with varying difficulty levels.
>
> In addition, we encourage to read the "Re-clarification for our motivation and theoretical establishments" part in ***General Response***, which maybe crucial for your concerns!
>
>
> > **Q4:** You cite (Vaswani, 2017) for the transformer model, but shouldn't this be (Vaswani et al., 2017)?
>
> **A:** Thank you for pointing out this citation mistake. We will make the correction in the formal version!
>
> > **Q5:** How do you train the model using which Figure 5 was made?
>
> **A:** Thanks for your question! Our transformer model is based on the GPT-2 architecture with 12 layers, 12 attention heads, and 768-dimensional embeddings. Limited by computation resources and training data, we use "AutoModelForCausalLM.from_pretrained("gpt2")" to load a pretrained version from the HuggingFace model hub, not training from scratch. In order to observe the optimization process of transformers, we train the model for 200 epochs using the AdamW optimizer with a batch size of 32, with Counterfact and HotpotQA dataset. All experiments are conducted using a single 24GB NVIDIA GeForce RTX 3090.
>
> Once again, we sincerely thank you for the detailed review and constructive feedback! We hope that our clarifications satisfactorily address your concerns and we welcome any further discussions that may be helpful to the evaluation.

---

> > ### Comment · Reviewer_PxzQ · 2024-11-26
> >
> > Thanks for this response! I'm ultimately unconvinced by it, as it doesn't really address the way in which you operationalize syntax / semantics in your empirical analyses, or many of my other points (e.g. about related work). I do see now a way in which this paper might have been able to prove its point: it could have built on prior work from computational linguistics that shows specific syntactic phenomena are learned before other specific semantic phenomena, explaining theoretically why that is the case, with empirical results to confirm. This is not, unfortunately, where the paper is now.
> >
> > More generally, I get the impression, from the way in which you discuss and study syntax and semantics, that no (computational) linguist was involved in the creation of this work; I highly recommend you involve one! Be careful with claims like this one:
> >
> > > there are some empirical studies which have already examined various aspects of language. This provides a strong foundation to support that syntax and semantics are representative and well-understood components of linguistic features.
> >
> > Words do indeed have syntactic and semantic word-level features, such as part of speech, or animacy, but no, "syntax" and "semantics" alone are not actually features of a word (and such features are often studied at the phrase / sentence level). Please talk to a linguist!
> >
> > Finally, I have read your general response as well, but found it no more convincing. I recognize that this is a theoretical work, but if you are to make theoretical claims about linguistic attributes of words / LMs' learning process, you must engage more with linguistic theory, not just mathematical / ML theory.

---

### Official Review · Reviewer_7oPM · 2024-11-03

**Soundness:** 2
**Presentation:** 3
**Contribution:** 2
**Rating:** 3
**Confidence:** 5

**Summary:**

This paper presents an innovative feature-learning-inspired approach that uses the in-context learning framework to acquire knowledge during training dynamics. The authors construct a robust mathematical framework to address the problem first, providing a rigorous and detailed analysis of training error and convergence properties. Additionally, through motivating experiments, they demonstrate a link between their theoretical findings and practical applications in NLP, showcasing the potential of their approach for real-world complex problems.

**Strengths:**

The paper is well-written, with a thorough review of related work and clear, thought-provoking remarks. The mathematical framework is robust, and the proofs are detailed and precise, enhancing the rigor and readability of the theoretical contributions.

**Weaknesses:**

While the paper presents a strong foundation, there are issues within the proofs that need further clarification. Addressing these concerns would strengthen confidence in the accuracy of the error bound analysis; without such revisions, the analysis may not be fully reliable.

Additionally, the experimental section lacks sufficient explanation, which would be essential for solidifying the connection between the empirical results and the theoretical framework.

Although the paper primarily focuses on the training perspective, it would benefit from an analysis of pre-trained model properties, such as error behavior, distribution shift, and generalization. This could offer more compelling insights into the applicability of the approach for NLP and related fields.

**Questions:**

I would appreciate and change the score if you could explain the following points.

**Theoretical Analysis and Proof:**

Your paper presents a robust theoretical framework with well-constructed proofs. However, there are a few points listed below, particularly in Theorem 2, that I found somewhat unclear.

1. ***Error Bound Analysis:*** Could you comment on the error bound for a pre-trained transformer in your framework? Specifically, how does the error behave with varying context (or prompt) lengths for such a model?

2. ***Clarification of “Easy-to-Fit” and “Hard-to-Fit” Components:*** Given the rigorous mathematical nature of your paper, a more formal explanation or intuitive insight into “easy-to-fit” versus “hard-to-fit” components would help clarify the intrinsic differences between the P and Q components. Without this, it’s challenging to grasp the core distinctions between these parts.

3. ***Explanation of Proof Steps in Theorem 2:*** Could you briefly clarify specific steps in the appendix related to Theorem 2? In particular:

   - Lines 2220 to 2222: based on your notation in line 2183 of $\epsilon_W$, and the equation in equation (21), it seems from line 2220 to line 2222, you use an inequality: $\log(Poly(d)) \le \log(d)$, please explain why.

   - Lines 2228 to 2230: with your notation in line 2211 $\epsilon_W=(Poly(d))^{2/3}$, and $L=Poly(d)$, please explain how do you get $$\frac{\sqrt{d} \log(d)}{L} \epsilon_W\le \frac{\sqrt{d}\log{d}}{Poly(d)}$$

   -  Lines 2238 to 2243: in line 2238, is the max value already greater than log(2)? $$\max(\log(1+e^{-x}), \log(1+e^{x}))>log(2),\ \text{where}\ x=\epsilon_{W,1}+\sqrt{d}/L\log(d) \epsilon_W$$ Later you mention the LHS of above inequality less than $\epsilon_{W,1}+\frac{\sqrt{d}\log(d)}{L}\epsilon_W+\frac{1}{\sqrt{\log(d)}}$, the upper bound of $K_{t_1}^1(\overline{W}_{t_1})$. If so, does the upper bound of Theorem 2 align with Theorem 1 on the same constant scale, which cannot converge to zero as the dimension $d$ goes to infinity?



**Experimental Setup:**

Your motivating experiments are thoughtfully designed to link with the theoretical work. However, I have a few questions regarding the details:

1. ***Transformer Architecture:*** You mention that the practical model is based on a GPT-2 architecture. Is this still a one-layer transformer as in your theoretical analysis? If it’s not one layer, could you briefly explain how the one-layer theoretical results extend to this more complex structure? Additionally, could you specify the dimensionality $d$ used in the experiments?

2. ***Syntactic and Semantic Evaluation:*** You state that “all predictions meet syntactic requirements.” Could you clarify how this was verified? More specifically, what distinguishes syntactic from semantic evaluation in your setup? Considering the large-scale nature of your dataset, was there any metric or human evaluation to support this claim? Did you consider including plots showing accuracy over time? This could provide additional insight into the learning process.

3. ***Transition Point Analysis (T=5):*** The transition at T=5 is intriguing and seems tightly related to your theoretical findings. Would it be possible to compute an approximation of the theoretical bound (e.g., $t_1$) and compare it with this empirically observed threshold? This could help clarify the connection between your experimental and theoretical results.

4. ***Prompt Construction Details:***

   - Is prompt length fixed during training?
   - Could you elaborate on how the prompts are constructed in practice? Are they derived from a single dataset, such as the Counteract dataset?
   - In Figure 5, do all questions use the same prompts. Besides, are these prompts changed during training?

---

> ### Comment · Reviewer_7oPM · 2024-11-25
> **Request for Clarification Regarding Critical Issue**
>
> Dear Authors,
>
> I acknowledge the significant effort you have invested in your manuscript, and I truly appreciate the dedication reflected in your work. However, I would like to ask for clarification on whether you are able to address the critical issue in your proof I have outlined above.
>
> If the issue cannot be resolved, I may need to reject your work, as this concern pertains to what I understand to be the central contribution and result of your paper.
>
> Best regards,
>
> Reviewer 7oPM

---

> > ### Author Response · Authors · 2024-11-26
> >
> > Thank you sincerely for your insightful comments! Below, we do our best to address your questions adequately.
> >
> > **Theoretical Analysis and Proof:**
> >
> > > **Q:** Error Bound Analysis: Could you comment on the error bound for a pre-trained transformer in your framework? Specifically, how does the error behave with varying context (or prompt) lengths for such a model?
> >
> > **A:** Thanks for your question! We would like to make a more detailed discussion on the error bound one by one!
> > - **In Theorem 1(a.2): cannot learn the hard-to-fit $\mathcal{Q}$**.
> >   We provide the lower bound for the training loss of hard-to-fit component $\mathcal{Q}$. The value, $\log 2 - \frac{1}{\sqrt{\log d}}-\sqrt{\frac{\log d}{N}}$, is close to $\log 2$ with high data dimension $d$ and more training prompts $N$. Perhaps $\log 2$ is small in absolute value, but in Theorem 3(c.2), we prove that with annealing the learning late, the loss of component $\mathcal{Q}$ has a great drop, whose upper bound is much smaller than $\log 2$. Overall, within $t_1$ iterations, the training loss of component $Q$ remains large and it exhibits that within $t_1$ iterations, the specialized knowledge like $\mathcal{Q}$ is not effectively learned by the network $g$.
> > - **In Theorem 2(b.2): learn the easy-to-fit $\mathcal{P}$**.
> >   We provide that within $t_1$ iterations, the training loss of easy-to-fit component $\mathcal{P}$ is upper bounded by $\epsilon_{W,1} + \frac{\sqrt{d} \log d}{L} \epsilon_W + \frac{1}{\sqrt{\log d}}$. With the definition in Equation (16) and Equation (10), we have $\epsilon_{W,1} \triangleq \tau_0 (u+\gamma_0)^2 \sqrt{\frac{d\log d}{L}}$ and $\epsilon_W = K^{4/3}\lambda^{-4/3}\tau_0^{-4/3} L^{2/3}$. It's obvious to find that longer prompt length (i.e., larger $L$), better optimimzation performance (i.e., smaller training loss). With our choices of $\lambda, \tau_0, L$ in Assumption 1, the upper bound of the training loss of component $\mathcal{P}$ has the order of $\frac{1}{Poly(d)}+\frac{1}{(Poly(d))^{1/3}}+\frac{1}{\sqrt{\log d}}$. When with data dimension going to infinity, prompt length goes greately large accordingly! The loss of easy-to-fit component $\mathcal{P}$ is upper bounded by an $o(1)$ term which converges to zero. In summary, the network $h$ learns elementary knowledge like $\mathcal{P}$, marking the so-called elementary stage.
> > - **In Theorem 3(c.2): learn the hard-to-fit $\mathcal{Q}$**.
> >   We observe that the upper bound of the training loss of hard-to-fit component $\mathcal{Q}$, given by $\epsilon_{V,1} + \frac{1}{(\log d)^{1/4}} + \frac{1}{\sqrt{\log d}}$. Similarly, with the definition in Equation (17), we have $\epsilon_{V,1} \triangleq \tau_0 (u+r)^2 \sqrt{\frac{d\log d}{L}}$. It presents that with longer prompt length, we can achieve lower optimization error. With our choices of $\lambda, \tau_0, L$ in Assumption 1, the upper bound has the order of $\frac{1}{\text{Poly}(d)} + \frac{1}{(\log d)^{1/4}} + \frac{1}{\sqrt{\log d}}$. It converges to zero as dimension $d$ goes to infinity. Compared to Theorem 1 with constant lower bound, we conclude that with a small learning rate, the network $g$ learns specialized knowledge, marking the so-called specialized stage.
> > - **In Theorem 4(d.2): continue to preserve the knowledge $\mathcal{P}$**.
> >   We prove that the small changes in loss is upper bounded by $\frac{\epsilon_{V,1}^2}{\log^2(1/\epsilon_{V,1})\sqrt{\log d}}$. With the definition in Equation (17), we have $\epsilon_{V,1} \triangleq \tau_0 (u+r)^2 \sqrt{\frac{d\log d}{L}}$. Then the upper bound represented with $L$ is $\frac{\tau_0^2 (u+r)^4 d \sqrt{\log d}}{L \log^2(\tau_0 (u+r)^2 \sqrt{d\log d /L})}$, thus with larger prompt length $L$, this change is smaller. It indicates that longer prompt provides a good chance for network $h$ to better preserve the knowledge $\mathcal{P}$ acquired during the elementary stage.

---

> > > ### Author Response · Authors · 2024-11-26
> > >
> > > > **Q:** Clarification of “Easy-to-Fit” and “Hard-to-Fit” Components: Given the rigorous mathematical nature of your paper, a more formal explanation or intuitive insight into “easy-to-fit” versus “hard-to-fit” components would help clarify the intrinsic differences between the P and Q components. Without this, it’s challenging to grasp the core distinctions between these parts.
> > >
> > > **A:** Thank you for suggestion! We sincerely regret any confusion caused by our introduction of the syntax, semantics or easy-to-fit component $\mathcal{P}$, hard-to-fit component $\mathcal{Q}$. We respectfully argue that these theoretical modelings are indeed motivated by experimental findings and are estabilished to provide a rigorous mathematically analysis for this stage-wise learning phenomenon. We strongly encourage to read ***General Response*** for further clarification of our motivation regarding the statements on syntax and semantics, as well as further experimental evidence!
> > >
> > > > **Q:** **Explanation of Proof Steps in Theorem 2.**
> > >
> > > **A:** Thank you for carefully reviewing our proof and giving us the opportunnity to further examine the long-form proof! We are greatly delighted to provide additional clarification regarding your questions! For Lines 2220 to line 2222, in our analysis, let $\text{Poly}(d)$ represent a polynomial in $d$, i.e., $\text{Poly}(d) = d^k$ for some $k > 0$. Then $\log(\text{Poly}(d)) = k \log d$. Consequently in this case, we use the expression $\log(\text{Poly}(d)) \lesssim \log d$ to indicate that $\log(\text{Poly}(d))$ grows at the same order as $\log d$. For Lines 2228 to 2230, we sincerely regret for this typo in the proof. As you mentioned, it's indeed more accurate to write $
> > > \frac{\sqrt{d}\log d}{L} \epsilon_W \lesssim \frac{\sqrt{d}\log d}{(Poly(d))^{1/3}}$ rather than $\frac{\sqrt{d}\log d}{L} \epsilon_W \lesssim \frac{\sqrt{d}\log d}{Poly(d)}$. Yet, this does not affect our final result, as we do not substitute the order of $\epsilon_W$ and $L$ into $K^1_{t_1}(\overline{W}_{t_1}) \lesssim \epsilon_{W,1}+\frac{\sqrt{d}\log d}{L}\epsilon_W)+\frac{1}{\sqrt{\log d}}$. Furthermore, in the original version, we provide the right analysis in "Messages Behind Theorem 2", as noted in Line 394.
> > >
> > > **Experimental Setup:**
> > >
> > > > **Q:** Transformer Architecture: You mention that the practical model is based on a GPT-2 architecture. Is this still a one-layer transformer as in your theoretical analysis? If it’s not one layer, could you briefly explain how the one-layer theoretical results extend to this more complex structure?
> > >
> > > **A:** Thanks for your question! As you mentioned, our theoretical results are established on a single self-attention layer to simplify the mathematical analysis. This controlled setup allows for rigorous proofs of the characteristics of stage-wise training dynamics in transformers. We consider that a multi-layer transformer can be seen as a stack of single-layer transformers, where each layer further refines the representations learned by the previous layer. In this structure, the stage-wise learning dynamics proved in single-layer transformers persist and may even be amplified through feature interactions across layers. **Empirically**, we use the GPT2 model with 12 layers, 12 attention heads, and 768-dimensional embeddings. The experimental results show the same training dynamics as our theory indicated, suggesting that the underlying theoretical insights generalize to more complex architectures.
> > >
> > > In summary, by combining theoretical results and empirical evidence, the transition from single-layer to multi-layer transformers is justified and the observed phenomenon is consistent across these settings.

---

> > > > ### Author Response · Authors · 2024-11-26
> > > >
> > > > > **Q:** Syntactic and Semantic Evaluation: You state that “all predictions meet syntactic requirements.” Could you clarify how this was verified? More specifically, what distinguishes syntactic from semantic evaluation in your setup? Considering the large-scale nature of your dataset, was there any metric or human evaluation to support this claim?
> > > >
> > > > **A:** This is an excellent question! We do not spend much human evaluation to verify our results due to our selection of datasets. Currently, in our datasets, the syntactic errors are limited to tokens like "of", "the", "The" and "a", etc. Therefore, we only need to check if these words appear in the predicted answers and manually determine whether they comply with the syntactic rules. **It is not necessary to check every sample**, as this is difficult to achieve on large-scale datasets. Furthermore, what you suggested, the metric to evaluate syntax and semantics, **indeed motivates us to conduct new experiments!** We found that there are some syntax benchmarks like BLiMP, which is generally used to detect syntax in the model's output. Therefore, we are considering training the model simultaneously on multiple syntax datasets (e.g., BLiMP) and semantic datasets (e.g., SICK) of similar scale, and observing whether the model shows stage-wise preferences for different tasks. In this case, we can use the individual average loss on the syntax dataset and semantic dataset as metrics to evaluate the model's capabilities, achieving a human-free evaluation.
> > > >
> > > > We strongly encourage to read ***General Response*** for detailed clarification of our motivation regarding the statements on syntax and semantics, as well as further experimental evidence!
> > > >
> > > > **Prompt Construction Details.**
> > > > > **Q:** Is prompt length fixed during training? Could you elaborate on how the prompts are constructed in practice? Are they derived from a single dataset, such as the Counteract dataset? In Figure 5, do all questions use the same prompts. Besides, are these prompts changed during training?
> > > >
> > > > **A:** Thanks for your question! We would like to provide more prompt construction details. For CounterFact dataset:
> > > > - It is a question-answering dataset consisting of knowledge tuples in the form of (subject, relation, answer). There are three paraphrased prompts for each question. The main prompt is generated by replacing '{subject}' in the 'requested_rewrite["prompt"]' field with the specific entity. For example, "The mother tongue of {} is" is processed to "The mother tongue of Danielle Darrieux is."
> > > > - Based on the analysis above, the prompt length is not fixed, as the prompt is dynamically generated based on the data points. The length of the 'subject' affects the final prompt length, and the paraphrases may also vary in length due to differences in semantic expression.
> > > > - For each data point, the main prompt and the two paraphrases are fixed after construction. That is, the prompt does not change dynamically during training. However, since each data point has multiple paraphrases, the model switches between different paraphrases during training, thereby indirectly introducing prompt diversity.
> > > >
> > > > For HotpotQA dataset:
> > > > - We convert the original question to the prompt '<question> The answer is' or '<question>? The answer is' (if the question ends with ? or .), where the prompt variable '<question>' is replaced by the original question.
> > > > - Based on the above construction, the length of the prompt is not fixed; it depends on the specific question. The original question will be dynamically inserted into the prompt template.
> > > > - The prompts do not change during training. Once the question is formatted according to the specified template, the prompt remains fixed for each question throughout the training process.
> > > >
> > > > Once again, we sincerely thank you for the detailed review and constructive feedback! We hope that our clarifications satisfactorily address your concerns and we welcome any further discussions that may be helpful to the evaluation.

---

> ### Comment · Reviewer_7oPM · 2024-11-26
> **Further clarification from line 2238 to line 2243**
>
> Thank you for your feedback, before I go to the rest of the feedback, I still want you to further clarify this process, since I think it is wrong. To avoid any misunderstanding, let me explain my question in detail:
>
> From lines 2238 to 2243, you want to derive the key components to get the upper bound for $K_{t_1}^1(\overline{W}_{t_1})$.
>
> Here you first derive the upper bound for $K_{t}^1(W^*)$:
> $$
> K_{t}^1(W^*) \le \max(\log(1+e^{-x}), \log(1+e^{x})) \le \epsilon_{W,1} + \frac{\sqrt{d}\log(d)}{L} \epsilon_W, \ \text{where}\ x=\epsilon_{W,1}+\sqrt{d}/L\log(d) \epsilon_W
> $$
> (Pay Attention, the middle part has a lower bound $log(2)$!!! you can more clearly find that in line 2241)
>
> Later based on this property, you goes to line 2288 and have:
>
> $$K_{t_1}^1(\overline{W}_{t_1})\le K_{t_1}^1(W^*) + C$$
>
> and then based on the result from line 2238 to line 2243, you have:
>
> $$
> K_{t_1}^1(W^*) + C \le \max(\log(1+e^{-x}), \log(1+e^{x})) + C \le  \epsilon_{W,1} + \frac{\sqrt{d}\log(d)}{L} \epsilon_W +\frac{1}{\sqrt{\log(d)}}
> $$
> which gives you the upper bound for  $K_{t_1}^1(\overline{W}_{t_1})$.
>
> Based on this, you write your argument in  Line 394 is "upper bounded by an o(1) term which converges to zero as the dimension d goes to infinity" where the upper bound refer to  $  \epsilon_{W,1} + \frac{\sqrt{d}\log(d)}{L} \epsilon_W +\frac{1}{\sqrt{\log(d)}}$.
>
> However, as I explained above, the upper bound already has a lower bound which is shown in your process "line 2238 to line 2243"
> $$
> \epsilon_{W,1} + \frac{\sqrt{d}\log(d)}{L} \epsilon_W +\frac{1}{\sqrt{\log(d)}} \ge \max(\log(1+e^{-x}), \log(1+e^{x})) +\frac{1}{\sqrt{\log(d)}} \ge \log(2) +\frac{1}{\sqrt{\log(d)}}
> $$
> and as $d$ goes to infty, this cannot go to zero!!!
>
> More precisely, this implies the upper bound of Theorem 2 aligns with Theorem 1 on the same constant scale, which cannot converge to zero as the dimension $d$ goes to infinity. Can you provide more explanation on this. Thank you!
>
> Moreover, I will check the rest of the comprehensive feedback in detail, and generate the feedback more carefully for them later. Thank you for your detailed feedback.

---

> ### Comment · Reviewer_7oPM · 2024-11-26
> **Follow-up on "Further clarification from line 2238 to line 2243"**
>
> In fact, your proof from line 2241 to line 2243 is wrong. You want to show:
>
> $$\max\left(\log(2) - \frac{1}{2} ( \epsilon_{W,1} + \frac{\sqrt{d}\log(d)}{L} \epsilon_W), \log(2) + \frac{1}{2}( \epsilon_{W,1} + \frac{\sqrt{d}\log(d)}{L} \epsilon_W) \right) \le \epsilon_{W,1} + \frac{\sqrt{d}\log(d)}{L} \epsilon_W $$
>
> It would be much clear to see the problem if simply denote $S = \epsilon_{W,1} + \frac{\sqrt{d}\log(d)}{L} \epsilon_W$ and the above process can be rewritten as:
>
> $$
> \max\left(\log(2) - \frac{1}{2} S,\ \log(2) + \frac{1}{2} S \right) \le S
> $$
> which is WRONG based on your current conditions (especially, when $d$ goes infty where S goes to zero, this inequality is wrong). It might be correct for small $d$ where $S$ has a lower bound of $cons. \times \log(2)$, but it will make your argument in Theorem 2 become wrong, since you can not make $d$ sufficiently large.
>
> In a summary, from line 2238 to line 2243, you first find a larger upper bound (the max thing) and make a mistake on the following inequality which gives your current upper bound. The intermediate step is wrong, and that makes your argument for theorem 2 incorrect.

---

> > ### Author Response · Authors · 2024-11-26
> > **Reply to Remaining Concern**
> >
> > Thanks for your careful examination of our proof! We are greatly delighted by your interest in our research. We sincerely regret that we did not fully understand your concerns regarding this part of proof earlier. After carefully reconsidering your points and checking the proof, wholeheartedly agree with your observations. What you have highlighted is an excellent question, and we recognize that the original statements contained some inaccuracies.
> >
> > In our proof logic, we first establish that $\overline{W}\_{t_1}$ approaches to $W^\star$ and then the training loss satisfies $K^1\_{t_1}(\overline{W}\_{t_1}) \leq K^1_{t_1}(W^\star) + C$. **Notably, a constant $\log (1+e^{-1})$ (corrected from $\log 2$), appears in the upper bound in Theorem 2, truly due to its inherent presence in the optimal achievable loss $K^1_{t_1}(W^\star)$.**
> >
> > Let us make a correction as follows.
> > - **Regarding Theorem 2:**
> > We first know that the ground truth label $y^n_L \in \\{-1, +1\\}$. Then, in Line 2227, we show that model output is upper bounded by $\epsilon_{W,1} + \frac{\sqrt{d} \log d}{L} \epsilon_W$ with $\epsilon_{W,1} = 1/Poly(d)$ and $\epsilon_W = {(Poly(d))}^{2/3}$. We can also find that $|f_t^1(W^\star; X^n_1, Y^n)|$ satisfies the above upper bound. When with choosable prompt length $L$,
> > $$|f_t^1(W^\star; X^n_1, Y^n)| \lesssim \epsilon_{W,1} + \frac{\sqrt{d}\log d}{L} \epsilon_W \rightarrow 1$$
> > Thus the logistic loss $\log (1+e^{-z})$ should be expanded as a Taylor series at 1, leading to the $\log (1+e^{-1})$ term in Theorem 2, which was mistakenly omitted in the original version. Specifically, we have
> > $$
> > K_t^1(W^\star)=\frac{1}{N}\sum_{n=1}^N l\left(f_t^1(W^\star; X_1^n, Y^n) \right) =\frac{1}{N}\sum_{n=1}^N \log\left(1+\exp(- y^n_Lf^1_t)\right)  \lesssim \log(1+e^{-1}) + \epsilon_{W,1} + \frac{\sqrt{d}\log d}{L} \epsilon_W
> > $$
> > Therefore, at iteration $t_1$, the loss is upper bounded by
> > $$
> > \log(1+e^{-1}) + \epsilon_{W,1} + \frac{\sqrt{d}\log d}{L} \epsilon_W + \frac{1}{\sqrt{\log d}}
> > $$
> > **Although it does not converge to zero as dimension goes to infinity, it indeed approaches to the minimal achievable loss.**
> >
> > - **Regarding Theorem 1:**
> > In Line 2045-2051, we demonstrate that at iteration $t_1$, the model predictions for positive and negative samples are close to zero with dimension $d$ goes to infinity (different from Theorem 2, where with choosable prompt length $L$, it can tend to $1$).
> > $$
> > \max\\{|g_{t_1}(X_2,z)|, |g_{t_1}(X_2,z-\zeta)|,  |g_{t_1}(X_2,z+\zeta)|\\} \lesssim \frac{1}{(\log d)^{1/4}}
> > $$
> > Given these predictions, the logistic loss $\log (1+e^{-z})$ can be expanded as a Taylor series at 0, leading to the appearance of the $\log 2$ term in Theorem 1. Whether the model outputs or the lower bound of loss, can reveal that at iteration $t_1$, the hard-to-fit component is not effectively learned by network $g$.
> >
> >
> > Once again, thank you for your efforts in reviewing our proof! This has been incredibly valuable in refining our work. We sincerely hope these corrections effectively address your concerns!

---

> ### Comment · Reviewer_7oPM · 2024-11-26
> **Deep concern with your argument**
>
> Thank you so much for your further clarification. Meanwhile, I believe you should first modify your manuscript, since if you change your bound, the **"theorem 2"** and **"messages behind theorem 2"** should be rewritten and some attractive result becomes invalid, such as **"the network $h$ learns elementary knowledge like $\mathcal{P}$, marking the so-called elementary stage"**. I think all the reviewer should be notified by this key modification.
>
> Moreover, if you wish to use a Taylor series expansion at 1 to derive the upper bound, similar to the process outlined in Theorem 1, it should also be possible to determine a lower bound for $K_t^1(W^*)$. Based on your recent feedback, I believe the lower bound would be:
> $$
> K_t^1(W^*) = \frac{1}{N}\sum_{n=1}^N \log(1+\exp(-y_L^n f_t^1)) \gtrsim \log(1+e^{-1}) - (\epsilon_{W,1} + \frac{\sqrt{d}\log(d)}{L} \epsilon_W).
> $$
> as $d$ goes to infty, this can give you a lower bound for $K_t^1(W^*)$, which is $\log(1+e^{-1})$. Similar to your argument in Theorem 1, network $h$ shouldn't be assumed to effectively learn the elementary knowledge $\mathcal{P}$. I was wondering how would you interpret this?

---

### Official Review · Reviewer_cfMD · 2024-11-04

**Soundness:** 2
**Presentation:** 3
**Contribution:** 2
**Rating:** 3
**Confidence:** 2

**Summary:**

This work presents a theoretical proof on the learning process of Transformer models can be split into two stages: a first stage focusing on learning elementary knowledge (or syntax), and a second stage focusing on learning specialized knowledge (or semantics). The theoretical proof is done in a single-layer Transformer with certain simplifications and assumptions. Empirical verification is also done on two datasets: Counterfact and HotpotQA. This work also shows an interesting spectral characteristic of the attention weights as a corollary of the proof.

**Strengths:**

1. This work provides a rigorous proof of the training dynamics on the single-layer transformer model.
2. Empirical verification is done on two real NLP datasets. The spectral characteristics analysis is interesting and insightful.

**Weaknesses:**

1. The difference between syntax and semantics in this work is basically one is easier to learn than the other as a feature. Consequently, what this paper proofs is basically Transformer models learn easier features first, and learn more complex features later. While this finding is true according to the analysis done in this paper, this proof does not lead to much new sights about Transformers, syntax/semantics or ICL. Right now, even though the proof is done on Transformers, it really feels like a general conclusion that should be held on a much broader class of models. It might be useful to add a discussion section highlighting what properties of Transformer or ICL are critical for this proof. I would also suggest shifting the focus of this paper away from the syntax/semantics contrast. The syntax/semantics aspect is more suitable to be one of the additional findings in the empirical verification part.
2. The empirical verification part of this work should be more detailed. Right now, there are no quantitative metrics about the syntax or semantics accuracy after each stage. There is also no study focusing on the actual length of the first learning stage even though the theory seems to suggest a certain length of the first learning stage.

**Questions:**

1. In the analysis, one key difference between the two learning stage is the constant learning rate and the annealing learning rate. Is that crucial to the proof and do you observe the same effects in practice?
2. L2 normalization is hardly used in the actual training of Transformers. How do you view this difference between the proof and the practice?

---

> ### Author Response · Authors · 2024-11-26
>
> Thank you sincerely for your thoughtful comments! We are delighted that you found our paper provides rigorous proof and insightful spectral characteristics analysis. To further address your concerns regarding the transformer-specific theory, ***we have made more clarifications that our theoretical modeling and proof are both specifically tailored to transformers.*** We strongly encourage to read ***General Response*** for further clarification of our motivation regarding syntax and semantics, as well as additional experiments! Below, we do our best to address your questions adequately.
>
>
> > **Q1:** The difference between syntax and semantics in this work is basically one is easier to learn than the other as a feature [...] Right now, even though the proof is done on Transformers, it really feels like a general conclusion that should be held on a much broader class of models.
>
> **A:** Thanks for your question! We would like to make more clarifications that our theoretical modeling and proof are specifically tailored to transformers. In more detail:
> - **Training Prompt Structure.** As suggested by the general ICL regime, for the $n$-th prompt, input samples $x_1^n, \cdots, x_{L-1}^n$ and query $x_L^n$ are drawn randomly and independently from the same data distribution. The input-label pairs are stacked to form a training prompt $P^n=\left(x_1^n, y_1^n, \cdots, x_{L-1}^n, y_{L-1}^n, x_L^n\right)$, with prompt length $L$. **In summary,** we explore the training dynamics of transformers, considering training them to hold ICL abilities using $N$ training prompts as introduced above. This prompt structure is specifically designed to align with the properties of attention layers in transformers, yet is largely irrelevant to other model architectures.
> - **Proof Specific to Transformers.** We investigate how model weights are optimized and updated, as well as how the training loss evolves during the training process. Theoretically, we primarily adopt the signal-noise decomposition technique to analyze the characteristics of differences in activations and network output under various activation and weight schemes (such as using noise weight to compute activation and signal weight to compute attention score, which is a crucial aspect). Further details are provided in the *Proof sketches* in Appendix Remark 6-9. **In summary**, the stage-wise training dynamics and the signal-noise decomposition technique are highly inspiring for other models, yet variations in training data, activation modes and model architectures pose significant challenges in characterizing the changes in activations and model outputs, and further in the final finite-time convergence results.
>
> > **Q2, Q3:** I would also suggest shifting the focus of this paper away from the syntax/semantics contrast. The syntax/semantics aspect is more suitable to be one of the additional findings in the empirical verification part [...] The empirical verification part of this work should be more detailed. Right now, there are no quantitative metrics about the syntax or semantics accuracy after each stage.
>
> **A:** Thank you for suggestion! We sincerely regret any confusion caused by our introduction of the syntax, semantics or easy-to-fit component $\mathcal{P}$, hard-to-fit component $\mathcal{Q}$. We respectfully argue that these theoretical modelings are indeed motivated by experimental findings and are established to provide a rigorous mathematically analysis for this stage-wise learning phenomenon. In addition to the current empirical validations, we have designed more experiments with specific quantitative metrics about the syntax and semantics to further support our theory. We strongly encourage to read ***General Response*** for further clarification!

---

> ### Author Response · Authors · 2024-11-26
>
> > **Q4:** In the analysis, one key difference between the two learning stage is the constant learning rate and the annealing learning rate. Is that crucial to the proof and do you observe the same effects in practice?
>
> **A:** Thanks for your question! Annealing the learning rate is indeed crucial to enable the model to acquire specialized knowledge in the later stage without undermining the elementary knowledge. We would like to analysis the learning rate annealing strategy theoretically and practically!
>
> - **Theoretically,** in deep learning models, the complexity of knowledge required varies at different stages. A larger learning rate enables the model to quickly learn the simple characteristics of the data distribution. And then a smaller learning rate helps fine-tune the model to capture fine-grained specialized features without disrupting elementary knowledge. **This stage-wise learning has been evaluated in our theory and empirical validations!** Furthermore, a high learning rate during the initial phase helps the model escape shallow suboptimal solutions, while a smaller learning rate in later phases allows for more precise parameter adjustments, increasing the likelihood of finding better solutions.
>
> - **Practically,** a large learning rate in later stages would result in excessive parameter updates, disrupting learned patterns. Annealing mitigates this issue, making the training process smoother. In addition, learning rate annealing works well with **various optimizers** (e.g., Adam, SGD) and delivers significant improvements across different tasks, including image classification, natural language processing, and reinforcement learning.
>
>
> > **Q5:** L2 normalization is hardly used in the actual training of Transformers. How do you view this difference between the proof and the practice?
>
> **A:** Thanks for your question! We acknowledge that there is a certain gap between theory and practice, including the use of a simplied transformer architecture and L2 normalization. **In theoretical analysis**, L2 normalization plays a crucial role as it limits the unbounded growth of parameters, reduces model complexity and enhances the controllability of finite-time convergence analysis.
>
> Moreover, in the **practical training** of transformers, L2 regularization (typically implemented as the 'weight decay' parameter in optimizers like AdamW) remains a commonly used technique. Its value is generally set within a relatively low range (such as $10^{-5}$ to $10^{-2}$), aligning with our assumptions on hyperparameters, where $1/\lambda = \mathcal{O}(\sqrt{\log d})$, as stated in Assumption 1.
>
> **In total**, incorporating L2 normalization provides significant benefits for theoretical analysis, while its small magnitude in practical training ensures minimal divergence between our theory and real-world implementations.
>
>
> Once again, we sincerely thank you for the detailed review and constructive feedback! We hope that our clarifications satisfactorily address your concerns and we welcome any further discussions that may be helpful to the evaluation.

---

> > ### Comment · Reviewer_cfMD · 2024-12-02
> >
> > Thank the authors for the explanations and it indeed resolved some of my concerns! However, I'll remain my original rating due to (1) while I totally understand the motivation part behind syntax/semantics, it is just not very rigorous to include that as the main claim of the paper (concern also shared by Reviewer PxzQ), and I still believe that shifting the focus away from syntax/semantics and only use that as one empirical experiment section will improve the quality of this draft, making the claim more accurate without touching any technical part of this work; and (2) technical problems pointed out by Reviewer 7oPM.

---

### Official Review · Reviewer_nsMi · 2024-11-06

**Soundness:** 3
**Presentation:** 3
**Contribution:** 3
**Rating:** 6
**Confidence:** 2

**Summary:**

This theoretical paper examines a scenario where there are two sets of features: "easy-to-learn" features, which are learned quickly, and "hard-to-learn" features, which require more time and effort to acquire. The authors demonstrate, using a simplified transformer model with a specific learning rate schedule resembling typical approaches, that the model first learns easy-to-learn features before moving on to hard-to-learn features. They argue that easy-to-learn features correspond to general syntactic knowledge in language models, while hard-to-learn features relate to semantics, defined as specialized or domain-specific knowledge.  The empirical part is very limited, the focus is on the theory.

**Strengths:**

This paper presents an impressive theoretical approach to explaining an intriguing empirical observation: that learnng various phenomena often progresses through distinct stages. To my knowledge, there is limited theoretical work that delves into the mechanisms underlying this 'staged' progression, making this paper a valuable contribution to the field. The observation that easy-to-learn features are leveraged early in learning, followed by a shift to harder-to-learn features, is supported here with rigorous analysis (see my 'confusions' below though), supprting the intuition that has lacked formal backing.

**Weaknesses:**

1. The generative process description for the data is somewhat unclear. Based on the current explanation, it seems that the Bayes optimal classifier might not need to rely on semantic features; syntactic features appear sufficient to solve the task in an asymptotic setup. This would havew undermined the notion that x2 captures specialized knowledge; specialized knowledge should provide benefits on a limited set of specific tasks, rather than being redundant to general knowledge. Figure 3 and the reference to Li et al. (2019) suggest that my interpretation is incorrect. E.g., in Li et al there’s a stochastic choice in generating features for each example (would 'easy' or 'hard' for this case)., but it is not what is happening here. This needs to be explained better.

2. I’m unclear on the rationale for assuming a block-diagonal structure (Eq. 2). In realistic settings, one wouldn't typically know which features are hard or easy in advance.

3. We normally use adaptive gradient methods rather than SGD. Would such a method, which rescales gradient components, affect the findings? For instance, might it amplify updates for weights associated with hard features (i.e., x2)?

4. What if t the initial learning phase is skipped and we started annealing immediately? How would it affect learning dynamics?

5. The decision to link optimization parameters (such as the learning rate in the second stage) to the data generation procedure seems unrealistic. Is the idea that this selection approximates parameters typically chosen through hyperparameter tuning? This paper demonstrates that specific parameters enable the model to acquire hard-to-learn knowledge in the later stage without undermining the easy-to-learn knowledge, but it doesn’t show that this outcome holds across a wide range of parameters or those commonly used in practice.

6. I would discourage the authors from using terms semantics and syntax as these seem to be misleading.

More broadly, it is unclear to me if the two stages are results of the explicit choices in defining the generative process and the stage of learning, or would emerge under more broad range of settings.

Overall, to fully grasp the importance of the assumptions, a careful reading of the proofs is necessary, which was challenging for  me and would probably be challenging for conference reviewers in general. Given the paper's length (56 pages!), it seems more appropriate for a journal (e.g., JMLR).

**Questions:**

I posed questions in the paragraphs describing weaknesses.

---

> ### Author Response · Authors · 2024-11-26
>
> Thank you sincerely for your support and insightful comments! We strongly encourage to read the ***General Response*** first and return here for further clarifications! Below, we do our best to address your questions adequately.
>
> > **Q1:** The generative process description for the data is somewhat unclear. Based on the current explanation, it seems that the Bayes optimal classifier might not need to rely on semantic features; syntactic features appear sufficient to solve the task in an asymptotic setup [...] Figure 3 and the reference to Li et al. (2019) suggest that my interpretation is incorrect. E.g., in Li et al there’s a stochastic choice in generating features for each example (would 'easy' or 'hard' for this case)., but it is not what is happening here. This needs to be explained better.
>
> **A:** Thank you for your thoughtful consideration! As you mentioned, it is indeed natural to further consider the existence of stochastic noise in generating samples. This implies that for certain samples, the prediction task depends solely on features from $x_1$ (aligning with your statement that "the Bayes optimal classifier might not need to rely on semantic features,", yet this case is beyond our current scope). For other samples, the prediction task depends solely on features from $x_2$. **The remaining cases, which we really focus on, are those where features from both $x_1$ and $x_2$ are simultaneously present.**
>
> Our theoretical modeling of the data structure, the case where both $x_1$ and $x_2$ are present in a sample, ***better benefits our goal*** of exploring how transformers acquire different types of knowledge during the dynamics of training. Specifically, completing the composite nonlinear classification task requires the model to grasp knowledge related to both components $\mathcal{P}$ and $\mathcal{Q}$. This further accelerates our conclusion: when features of varying difficulty coexist in the training data, the model exhibits a clear preference in learning, showing a stage-wise learning process for easier and harder features.
>
> We sincerely hope the above analysis satisfactorily address your concerns!
>
> > **Q2:** I’m unclear on the rationale for assuming a block-diagonal structure (Eq. 2). In realistic settings, one wouldn't typically know which features are hard or easy in advance.
>
> **A:** Thanks for your question! In deep learning models, the actual structure of weights and the interactions between features are highly complex, making it nearly impossible to directly analyze the characteristics of a fully developed practical model. The assumption of block-diagonal structure is not a strict description of the actual training process but rather a ***theoretical abstraction*** used to study ***whether the model can effectively decompose tasks within different feature subspaces***.
>
> Referring to **feature learning theory**, we indeed do not need to know in advance which features are simple and which are complex. The distinction between $X_1$ and $X_2$ lies in theoretical abstraction: for a $2d$-dimensional feature space, simple feature dimensions are grouped into $X_1$, and complex ones are grouped into $X_2$. After such *abstraction and reordering*, the features can theoretically form a block structure (where features exhibit a certain degree of independence in the two subspaces). Correspondingly, we assume the weight matrix is block-diagonal, which allows us to more clearly observe how the network learns the distributions of feature subspaces $X_1$ and $X_2$ through distinct weights $W$ and $V$, respectively. This makes theoretical analysis feasible and helps us better understand the behavior of the model in separating feature learning processes.
>
> In total, we model the learning process of different types of features based on assumptions with certain practical foundations, and we acknowledge that this simplification represents a general limitation for the theoretical analysis of weight optimization. Meanwhile, it is indeed a promising direction to explore more practical weight structures, and we hope to provide theoretical techniques and insights to inspire future work!

---

> > ### Author Response · Authors · 2024-11-26
> >
> > >**Q3:** We normally use adaptive gradient methods rather than SGD. Would such a method, which rescales gradient components, affect the findings? For instance, might it amplify updates for weights associated with hard features (i.e., x2)?
> >
> > **A:** Thanks for your question! Adaptive gradient methods (such as Adam and AdaGrad) adjust the learning rate of each parameter based on historical gradients. This adjustment dynamically slows down parameter updates over the course of training (for most gradient directions), effectively achieving an "automatic annealing" effect. Similarly, our learning rate annealing strategy reduces the initial large learning rate, allowing the model to focus on learning more complex features in the later stages of training. Additionally, in our experiments using Adam for optimization, we observed a similar phenomenon of stage-wise learning. The core purpose of both approaches is similar: (1) **Accelerating early-stage learning**: A larger learning rate in the early stages of training helps to quickly capture easy-to-fit patterns (e.g., syntactic features). (2) **Refining late-stage learning**: A reduced learning rate in the later stages gradually optimizes hard-to-fit patterns (e.g., semantic features).
> >
> >
> > > **Q4:** What if the initial learning phase is skipped and we started annealing immediately? How would it affect learning dynamics?
> >
> > **A:** Thanks for your question! With the fundamental settings introduced at the beginning of Section 4, we have $\eta_1 = \Theta(1)$ and $\eta_2 = \eta_1 \lambda^2 \epsilon_{V,1}^2 r < \eta_1$. In the scenario you mentioned, if we started annealing immediately, this would mean setting a relatively small learning rate $\eta_1^\prime$ (as the order of $\eta_2$). Specifically, with $\eta_1^\prime = \eta_1 \lambda^2 \epsilon_{V,1}^2 r$, the corresponding $t_1^\prime \triangleq \frac{1}{\eta_1^\prime \lambda}$ would be $\mathcal{O}((\log d)^{3/2} (\text{Poly}(d))^2)$, which is significantly longer than the original $t_1 = \mathcal{O}(\sqrt{\log d})$. This results in a very low training process even for the easy-to-fit component $\mathcal{P}$, and similarly a far longer $t_2^\prime$ for the hard-to-fit component $\mathcal{Q}$.

---

> ### Author Response · Authors · 2024-11-26
>
> > **Q5:** The decision to link optimization parameters (such as the learning rate in the second stage) to the data generation procedure seems unrealistic. Is the idea that this selection approximates parameters typically chosen through hyperparameter tuning? This paper demonstrates that specific parameters enable the model to acquire hard-to-learn knowledge in the later stage without undermining the easy-to-learn knowledge, but it doesn’t show that this outcome holds across a wide range of parameters or those commonly used in practice.
>
> **A:** Thanks for your question! We acknowledge that our theoretical results are established on a relatively restricted setting (with annealing the learning rate and some hyperparameter order requirements), yet ***this is required for rigorous mathematical proof***.
>
> **(1)** We would like to point that ***our theory aligns well with empirical observations in practice***. For example, models tend to learn easy-to-fit features (such as syntactic information) in the early stages of training, while hard-to-fit features (such as semantic information) are gradually learned in later stages. This dynamic is consistent with actual training processes and empirical intuition, which validates the relevance of our theoretical analysis.
>
> **(2)** Furthermore, our research provides new insights into the model’s learning process beyond the rigorous stage-wise learning theory. The choices of learning rate and hyperparameters suggest that
>   - Learning rate scheduling: Use dynamic adjustment strategies, such as a larger initial learning rate for the learning of easy-to-fit features and then a smaller learning rate for hard-to-fit features.
>   - Initialization: Use stable small-variance initialization schemes based on the data dimension.
>   - Prompt optimization: Adjust the input sequence length, especially in high-dimensional scenarios where sufficient context is necessary.
>
> In summary, while our work involves certain assumptions, its core value lies in offering a theoretical perspective on understanding the dynamics of model learning. It also provides a theoretical basis for optimizing practical strategies, such as hyperparameter tuning and learning rate scheduling.
>
>
> > **Q6, Q7:** I would discourage the authors from using terms semantics and syntax as these seem to be misleading [...] More broadly, it is unclear to me if the two stages are results of the explicit choices in defining the generative process and the stage of learning, or would emerge under more broader range of settings.
>
> **A:** Thank you for suggestion! We sincerely regret that there is some confusion for the introduction about syntax, semantics or easy-to-fit component $\mathcal{P}$, hard-to-fit component $\mathcal{Q}$. We respectfully argue that these theoretical establishments are indeed insightful and reasonable, rather than purely constructive. We strongly encourage to read ***General Response*** for further clarification!
>
> Once again, we sincerely thank you for the detailed review and constructive feedback! We hope that our clarifications satisfactorily address your concerns and we welcome any further discussions that may be helpful to the evaluation.

---

> > ### Comment · Reviewer_nsMi · 2024-12-02
> > **Thanks for the clarification**
> >
> > Thank you for the clarifications. After reading the other reviewers' comments, I tend to agree that the paper would benefit from another substantial round of revisions. This includes refining the framing of the work (e.g.,  the notions of syntax and semantics asraised by Reviewer PxzQ) and strengthening the theoretical results (see Reviewer 7oPM).
> >
> > Regarding your response to my comments, I understand the reasoning behind some of the simplifications you made. However, I remain concerned that these simplifications may weaken the paper's claims. For example, aspects like the learning rate schedule are not purely technical but could potentially impact the dynamics of learning semantics and syntax, and hence influence the conclsoins drawn from your work.
> >
> > That said, I acknowledge the ambitious scope of this project and appreciate the effort involved. I hope to see it revised and eventually published. As indicated in the initial review, given its scale and depth, I believe a journal venue, such as JMLR, might be more suitable for its final version.

---

### Author Response · Authors · 2024-11-26
**General Response to All Reviewers (1/2)**

We would like to express our gratitude to all the reviewers for their insightful and constructive suggestions which have helped us to improve our manuscript significantly. We are delighted to find that the reviewers agree that ***our paper is praised for*** its valuable theoretical contribution to the mechanisms underlying the stage-wise training dynamics, well-written with a thorough review of related work and clear, thought-provoking remarks. However, they also raised ***primary concerns*** regarding: (a) the motivation for the theoretical data structure to study syntax and semantics; (b) the limitation of experiments to verify syntax-then-semantics learning. In the following, we make more clarifications to all reviewers!

**Regarding (a): Re-clarification for our motivation and theoretical establishments.** We aim to further clarify the flow of our study in detail. Through the preliminary experiments shown in Figure 1, we focus on two critical characteristics of linguistic features: syntax and semantics. Experimentally, we observed that the model demonstrates a stage-wise preference for learning different types of features (syntax and semantics) during training. **Compared to most of empirical analyses**, our goal is to theoretically demonstrate how transformers acquire different types of knowledge during training. With this research motivation, we first study the properties of these two types of features--syntax and semantics in practice, enabling us to develop rigorous theoretical analyses ***using specified data structures that exhibit the same characteristics as syntax and semantics.***

**Step1: Observed characteristics of syntax and semantics.**
From the preliminary experiments shown in Figure 1, we focus on syntax and semantics as the two key linguistic features. For two similar questions, "The mother tongue of Thomas Joannes Stieltjes is" and "The mother tongue of Danielle Darrieux is," we specifically analyzed the characteristics of syntax and semantics:

- For syntax, although "Thomas Joannes Stieltjes" is replaced with "Danielle Darrieux", the syntactic structure remains unchanged. The model could easily infer that the answer would be a noun, demonstrating the ***smoothness*** of syntactic information: even with small changes, syntax remains recognizable.
- For semantics, when "Thomas Joannes Stieltjes" is replaced with "Danielle Darrieux", the answers to two questions are quite different, which shows that semantic understanding is highly sensitive to changes. Correctly understanding and distinguishing subtle semantic variations require complex reasoning by the model during training. As a result, semantic structures are ***non-robust***.

**Step2: Construction of theoretical data structures $P$ and $Q$.**
Based on the observed smoothness of syntactic information and non-robustness of semantic information, we specially design two data structures $P$ and $Q$ which exhibit the same characteristics, enabling us to develop theoretical analyses.

- For data structure $P$: $x_{i,1}^n$ is generated by combining $w^\star$ (a fixed direction) with noise $e$. Despite the presence of noise, the data remains linearly separable. This suggests that the data with distribution $P$ are **smooth**, as a linear classifier can handle it well and accurately separate the data points.
- For data structure $Q$: Positive class samples are $\alpha z$, and negative class samples are $\alpha (z \pm \zeta)$. $\zeta$ is small making them harder to separate linearly. This indicates that even minor perturbations can lead to label prediction changes, resulting in **non-robust** features.
- In total, through this careful construction, $P$ has smooth features, aligning with syntactic information, while $Q$ has non-robust features, aligning with semantic information.

**Step3: $P$ is easy-to-fit component and $Q$ is hard-to-fit component.** In addition to their smooth and non-robust characteristics, $P$ and $Q$ exhibit different levels of fitting difficulty. $P$ is easy-to-fit, since even with noise, the data remains linearly separable. The presence of a large margin makes it easier to classify the data using a simple linear model. $Q$ is hard-to-fit, since the boundary between data points is highly complex and a more complex nonlinear model is required to fit data.

In summary, we sincerely regret some confusion about the introduction about syntax, semantics or easy-to-fit component $\mathcal{P}$, hard-to-fit component $\mathcal{Q}$. We respectfully argue that these theoretical establishments are indeed insightful and reasonable, rather than purely constructive.

---

> ### Author Response · Authors · 2024-11-26
> **General Response to All Reviewers (2/2)**
>
> **Regarding (b): More experiments with quantitive metrics of syntax and semantics.** In our current experiments, due to the selection of datasets, the syntactic errors are limited to tokens like "of", "the", "The" and "a", etc. Thus, we only need to check if these words appear in the predicted answers and manually determine whether they comply with the syntactic rules. It is not necessary to check every sample, as this is difficult to achieve on large-scale datasets. Furthermore, as suggested by reviewers, the metric to evaluate syntax and semantics, **indeed motivates us to conduct new experiments!** We found that there are some syntax benchmarks like BLiMP, which is generally used to detect syntax in the model's output. Therefore, we are considering training the model simultaneously on multiple syntax datasets (e.g., BLiMP) and semantic datasets (e.g., SICK) of similar scale, and observing whether the model shows stage-wise preferences for different tasks. In this case, we can use the individual average loss on the syntax dataset and semantic dataset as metrics to evaluate the model's capabilities, achieving a human-free evaluation.
>
> Once again, we sincerely thank all the reviewers for the detailed review and constructive feedback! We hope that our clarifications satisfactorily address the concerns and we welcome any further discussions that may be helpful to the evaluation.

---

### Note · Authors · 2024-12-09

**Comment:**

We sincerely thank the reviewers for taking the time to provide valuable feedback. We plan to revise and polish our paper further based on their insightful recommendations. We have decided to withdraw the manuscript at this stage and will work on improving it for future submission!

**Withdrawal Confirmation:**

I have read and agree with the venue's withdrawal policy on behalf of myself and my co-authors.